# FlowNet: A Generic Independent and Interactive Model for Streamflow Forecasting

## Abstract

Streamflow forecasting plays a crucial role in water research for flood prevention, water resource management, or climate resilience. However, it is a challenging task due to complex hydrological system interactions, human interventions and global climate change. In this paper, we introduce FlowNet, a *unique local global interactive modeling* framework, which is capable of effectively predicting multiple hydrology stations with varied input climate features and data availability at the same time. The key idea of FlowNet is to contruct *independent* prediction models for each station from its local data and from its adjacent neighbors via a hydrological-related directed graph before letting these models to *iteratively* and *interactively* adjust each other to maximize their prediction agreements. This helps to reduce uncertainty, thus improving their accuracy. Additionally, FlowNet dynamically captures inter-station relationships via its directional and delay-aware graph reconstruction method. As a generic framework, FlowNet can be used with any existing Deep Learning (DL) backbone models such as RLinear, PatchTST or iTransformer. However, we also introduce another backbone, called Disentangled Multiscale Cross-attention Transformer (DMCT), to capture the multiscale seasonality-trend information for further performance boost. Extensive experiments on 3 large datasets, including LamaH (with 425 hydrology stations in Europe), CAMELS (672 stations in USA) and MRB (with 26 gauge stations in the Mekong River Basin), show that FlowNet significantly outperforms 18 state-of-the-art (SOTA) prediction methods in terms of MAE, RMSE, and NSE.

## 1 Introduction

River flow forecasting, which aims to accurately predict future flow conditions using historical hydrological data, is a critical research area with wide-ranging impacts such as flood management, water resource optimization, infrastructure protection, and climate resilience (Feng et al., 2021; Zhou et al., 2025; Jiang et al., 2024). However, despite many research efforts (Giladi et al., 2021; Najafi et al., 2024), it remains a challenging task, as hydrological systems involve complex and nonlinear interactions among many geographic and climate factors such as topography, rainfall, river discharge and soil texture as well as human interventions such as dam constructions (Haddeland et al., 2014).

Traditional streamflow forecasting methods typically rely on physical simulations (Vreugdenhil, 2013) such as MODFLOW (Harbaugh et al., 2000) and SWAT (Gassman et al., 2007), or statistical techniques such as ARIMA (Wang et al., 2018) and BJP (Robertson & Wang, 2012). However, these approaches struggle to effectively capture these intricate spatial-temporal dynamics, particularly under conditions of sparse or irregularly sampled data (Brunner et al., 2021). For example, simulation models often require a large amount of specific data such as soil type, land use, or digital elevation models (DEM), which are very difficult to collect and require significant effort, expertise, and computational power to set up (Giladi et al., 2021; Brunner et al., 2021). Moreover, global warming causes complex changes in climate patterns, and consequently breaks stationarity, a key assumption of most traditional models, (possibly) affecting their effectiveness (Milly et al., 2008).

Recently, deep learning has emerged as a powerful approach for modeling long-term complex temporal dynamics in streamflow forecasting tasks with various employed architectures such as MLPs (Sivakumar et al., 2002), GRUs (Farfan et al. 2024), CNNs (Ghimire et al., 2021), Transformer (Castangia et al., 2023), and especially LSTMs (Hu et al., 2020). These methods do not require

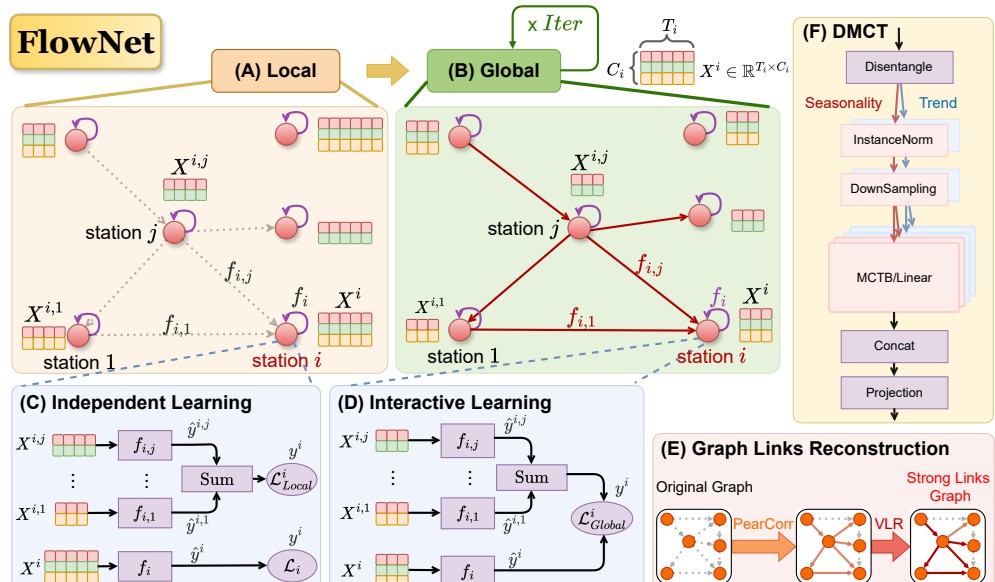

Figure 1: The overall pipeline of *FlowNet* with a barebone downstream graph $\mathcal{G}$. In the *Local phase* (A), for each station $i$, we train an independent per-station model $f_i$ and cross-station models $f_{i,j}$, where $j \in parents(i)$ (i.e. $j$ is parent of $i$), to learn the flow of station $i$ via the two independent loss functions $\mathcal{L}_i$ and $\mathcal{L}_{Local}^i$ described in (C) (cf. Section 2.2). In the *Global phase* (B), the per-station model $f_i$ and cross-station models $f_{i,j}$ *interactively* adjust each other in multiple iterations to maximize their prediction agreements via two global losses $\mathcal{L}_{Global}^i$ and $\mathcal{L}_{Global}^{i,j}$ described in (D) (cf. Section 2.2). In (E), the original downstream flow graph is refined by using our proposed methods PearCorr to search the correlated links and then VLR to reconstruct strong links graph (cf. Section 2.4). (F) presents our proposed architecture *Disentangled Multiscale Cross-attention Transformer (DMCT)* to capture multiscale seasonality-trend information.

wide ranges of complex data like traditional techniques, while acquiring impresive performances in major cases (Vizi et al., 2023). However, most of them often treat monitoring stations independently, neglecting the essential spatial relationships and dependencies of directional water flow between stations (Ding et al., 2020). To address these dependencies, Kratzert et al. (Kratzert et al., 2018) use a single LSTM model to jointly predict multiple stations, implicitly exploiting their underlying physics. Other works such as (Kirschtein et al. 2024) use graphs to explicitly capture relationships among stations via different hydrology and topography aspects and employ GNN-based models to forecast all stations. However, these methods typically rely on restrictive assumptions such as uniform input feature sets, similar training data periods or static adjacency graphs (Zhao et al., 2020; Zhou et al., 2025). They also focus on propagating features among stations, neglecting explicit relationships among predicted outcomes of different stations.

**Our contributions.** In this paper, we introduce *FlowNet*, a *flexible* and *generalizable* framework specifically designed for multivariate spatio-temporal streamflow forecasting on multiple gauge stations jointly. Compared to existing works, FlowNet has the following key differences.

First, FlowNet also aims to predict all stations jointly like other above mentioned methods. However, it follows an entirely different concept, called *interactive local global modeling strategy* (cf. Section 2.2 for details). Concretely, given a relationship graph of all stations as a backbone, rather than using a single large model like (Kratzert et al., 2018), each station is first represented by a set of smaller arbitrary models, including a *independently* customized *per-station* model and different *independent cross-station* models to predict its own streamflow from its local data and from its neighbors's local data in the *local stage* of FlowNet, respectively. Hence, these models can flexibly exploit all varying available local training data and input features to maximize their learning outcomes. In the *global stage*, all models *iteratively* and *interactively* adjust each other to maximize their prediction agreements by exchanging and fusing their prediction outcomes via the backbone graph, thus reducing prediction uncertainty and further enhancing their accuracy. This interaction scheme is the central of FlowNet and makes it highly *data* and *model flexibility*, i.e., it does not require uniform inputs like existing works such as (Kirschtein et al. 2024) and can be used with any existing DL models.

Table 1: Summary of notations in the paper.

| Notation | Definition |
|----------|------------|
| $X \in \mathbb{R}^{T \times C}$ | multivariate time series with time steps $T$ and channel $C$ |
| $S = \{S_1, \ldots, S_N\}$ | the set of $N$ hydrology stations in river networks |
| $A \in \mathbb{B}^{N \times N}$ | the adjacent matrix of stations |
| $\mathcal{G} = (S, A)$ | a directed graph of stations |
| $x_t^i = X_{t-L:t,:}^i \in \mathbb{R}^{L \times C_i}$ | the historical data at station $S_i$ from $t - L$ to $t$ |
| $x^{i,j} = X_{t-L:t,:}^j \oplus X_{t-L:t,c}^i$ | a concatenated historical data of $X^j$ and historical flow of $X^i$ |
| $F = \{f_k \| k = 1..M\}$ | a set of $M$ small independent models |
| $f_i$ | per-station model of station $S_i$ |
| $f_{i,j}$ | cross-station model of stations $S_i, S_j$ |
| $y_t^i = X_{t+1:t+H,c}^i \in \mathbb{R}^{H \times 1}$ | the future data of $c$-th channel associate from historical data $x_t^i$ |
| $\hat{y}^i = f_i(x^i)$ | a prediction outcome of station $S_i$ |
| $\hat{y}^{i,j} = f_{i,j}(x^{i,j})$ | the prediction outcome of $f_{i,j}$ |
| $\hat{y}_{inflow}^i$ | the predicted inflow of $S_i$ |
| $\hat{y}_{outflow}^i$ | the predicted outflow of $S_i$ |
| $\mathbf{s}, \mathbf{t}$ | the seasonality/trend components of time series |
| $\mathbf{h}$ | the multiscale latent sequences of Transformer block |
| $\Omega$ | the maximum lag threshold for Pearson Correlation Analysis |
| $\lambda$ | the maximum correlations under $\Omega$ lags |

Second, FlowNet captures inter-station relationships by proposing a directional and delay-aware graph reconstruction method, enabling robust *interactive learning* across spatially distributed stations dynamically as described above (cf. Section 2.4 for details).

Third, since FlowNet is a generic framework, it can employ diverse DL models such as MLPs, RNNs, and Transformers for different stations. However, we also propose a specific backbone model for FlowNet, called Disentangled Multiscale Cross-attention Transformer (DMCT), explicitly designed to decompose time series into distinct seasonal and trend components, effectively processing each at multiple temporal scales to improve the prediction accuracy (cf. Section 2.3 for details).

Fourth, we demonstrate the performance of FlowNet in (i) predicting daily water discharges for 425 hydrology stations in the Central Europe from the LamaH-CE dataset (Klingler et al., 2021) and 672 stations in the USA from the CAMELS dataset (Newman et al., 2015) and (ii) predicting water levels for 26 stations in the Mekong River Basin (MRB) collected from the Mekong River Commission (MRC). FlowNet acquires significantly better prediction accuracy in terms of NSE, MAE, and RMSE compared to 18 state-of-the-art (SOTA) prediction models with diverse DL architectures such as Transfomer-based models like CATS and iTransformer, MLP-based models like DLinear and RLinear, CNN-based models like MICN, GNN-based models like GCN, ResGCN and ResGAT, RNN-based models like LSTM and GRU, and hybrid models like AGCLSTM. Many of these models are specifically designed or widely used for streamflow forecasting tasks.

## 2 Our Proposed Method FlowNet

Let $X \in \mathbb{R}^{T \times C}$ be a multivariate time series, where $T$ is the number of time steps and $C$ is the number of channels. Additionally, we denote the $c$-th channel in the $t$-th time step as $X_{t,c} \in \mathbb{R}$, the time series of $c$-th channel as $X_{:,c} \in \mathbb{R}^T$, and the multivariate data in $t$-th time step as $X_{t,:} \in \mathbb{R}^C$.

**Problem formulation.** Let $S = \{S_1, \ldots, S_N\}$ be the set of $N$ hydrology stations in river networks. Let $\mathcal{G}$ be a directed graph that connects these stations via their direct flow relationships as follows.

**Definition 1** (Downstream flow graph $\mathcal{G}$). *Let $\mathcal{G} = (S, A)$ be a directed graph, where $S \in \mathbb{R}^N$ is a set of $N$ stations and $A \in \mathbb{B}^{N \times N}$ is the binary adjacency matrix. Two stations $S_i$ and $S_j$ are linked together, i.e., $A_{i,j} = 1$, if the water flows from $S_i$ to $S_j$ directly. Otherwise, $A_{i,j} = 0$.*

For each station $S_i$, let $X^i \in \mathbb{R}^{T_i \times C_i}$ be the multivariate time series data associated with it, *where the c-th channel contains streamflow values* and the remaining channels are exogenous variables such as climate. Notably, the number of time steps $T_i$ and the number of channels $C_i$ are station-specific. We aim to predict future flow values (i.e. the $c$-th channel) of all stations jointly.

**Key concepts of FlowNet.** Though all existing works, such as (Zhao et al., 2020), focus on a single large joint prediction model $f$ for all stations, we follow an entirely different approach that constructs a set of $M$ small independent models $F = \{f_k | k = 1..M\}$ and train them in two different phases: the *local* and *global* ones. Initially, in the *local* phase, each $f_k$ predicts the flow value of a single station independently in the beginning and belongs to one of the 3 categories including *per-station* (i.e. measured flow at a station), *inflow* (i.e. water flow into a station) and *outflow* (i.e. water flow out a station) forecasting as described in Section 2.1. After that, in the *global* phase, these models *iteratively interact* with others to adjust themselves via the flow graph $\mathcal{G}$ to maximize their prediction agreements, thus reducing uncertainty and increasing their accuracy (cf. Section 2.2). This setting is *data flexible*, since in many river basins like the MRB, the time series data lengths and collected hydrology features can be very different at different stations. Rather than choosing only a uniform subset of data for all stations like most existing works such as (Kirschtein et al. 2024), FlowNet can effectively utilize all of them due to its *independent* learning model to maximize learning generability. Concretely, each model can be trained in the local phase using all available data locally. During the global phase, the data is limited to common data of participated models. Note that, some recent HGNN-based methods such as (Jiang et al., 2024) can deal with data heterogeneous but still require uniform data for nodes with the same type. Moreover, FlowNet is also *model flexible*, any existing deep learning model can be independently used as $f_i$, including lightweight models that are computationally efficient and less prone to overfitting, especially when training data are limited. The interaction and ensemble fusion concepts of FlowNet among *per-station*, *inflow*, and *outflow* models also help it to produce more stable results than existing works.

## 2.1 Overview of FlowNet

In FlowNet, we have two main kinds of prediction models including: *per-station* and *cross-station* ones. The per-station model uses local data at each station to predict its future streamflow.

**Definition 2** (Per-station forecasting). *For each station $S_i$ and an arbitrary time step $t$, our objective is to predict future data $y_t^i = X_{t+1:t+H,c}^i \in \mathbb{R}^{H \times 1}$ of c-th channel water flow from multivariate historical data $x_t^i = X_{t-L:t,:}^i \in \mathbb{R}^{L \times C_i}$ from a model $f_i$, where $L$ is the lookback window length and $H$ is the future horizon length. For simplicity, we drop the term $t$ out whenever it is clear from the context and let $y^i$ be the ground truth and $\hat{y}^i$ be the prediction at $S_i$. We learn, $y^i \approx \hat{y}^i = f_i(x^i)$.*

In a river network, stations have physical relationships. Particularly, the water flows from upperstream stations to lower-stream stations. These flow relationships are exploited in FlowNet via its *local-global interaction scheme* shown in Figure 1 using a flow graph $\mathcal{G}$ to guide the model interactions. Given a station $S_i \in S$, let $parents(S_i)$ be the set of stations $S_j$ where $A_{j,i} = 1$ and $childs(S_i)$ be the set of stations $S_j$ where $A_{i,j} = 1$. At a time $t$, we expect that $X_{t,c}^i = \sum_{j \in parents(S_i)} inflow(X_{t-L:t,c}^j) + \epsilon_t$, where $inflow$ is the flow contribution of a parent station $S_j$ within a lookback window $L$ to station $S_i$ and $\epsilon$ is a noisy factor. Similarly, we have the *outflow* relationship from $S_i$ to its child stations. They are exploited to build cross-station models in FlowNet.

**Definition 3** (Cross-station prediction). *Wlog., let $S_i$ and $S_j$ be two adjacent stations with corresponding data $X^i \in \mathbb{R}^{T_i \times C_i}$ and $X^j \in \mathbb{R}^{T_j \times C_j}$ and $A_{j,i} = 1$. Let $f_{i,j}$ be a model to predict the inflow contribution from station $S_j$ to $S_i$. At an abitrary time $t$, let $\hat{y}_t^{i,j}$ be the prediction outcome of $f_{i,j}$, we have $\hat{y}_t^{i,j} = f_{i,j}(x_t^{i,j})$, where $x_t^{i,j} = X_{t-L:t,:}^j \oplus X_{t-L:t,c}^i$ be a concatenated historical data of $X^j$ and historical flow of $X^i$. However, unlike the per-station forecasting in Definition 2, we do not have a ground truth $y_t^{i,j}$. Instead, it is a learnable latent variable that can be inferred from the* inflow

*relationship discussed above during the training process. Similarly, we drop the term $t$ out for simplicity. Let $\hat{y}^i$ be a prediction outcome of station $S_i$, we learn $y^i \approx \hat{y}^i = \sum_{j \in parents(S_i)} f_{i,j}(x^{i,j})$. The outflow cross-station prediction is defined similarly. We use $\hat{y}^i_{inflow}$ and $\hat{y}^i_{outflow}$ to denote predicted inflow and outflow of $S_i$, respectively.*

**Overall structure of FlowNet.** Figure 1 shows the overall pipeline of our method. Specifically, we first refine the downstream flow graph $\mathcal{G}$ via the graph link reconstruction module (E) to avoid weak relationships, which can happen due to factors such as long-distance stations or human interventions (cf. Section 2.4). Then we train all *per-station* and *cross-station* models independently using their local-specific data as much as possible for better performance in the local phase (A). In this way, even if some stations do not have data on some channels or having different channel lengths, they will not affect other stations like existing works such as (Kirschstein & Sun, 2024) (cf. Section 2.2). In the global phase (B), the central of FlowNet, all models iteratively and interactively exchange their outcomes and adjust themselves using common data among them to further reducing uncertainty via their flow relationships in $\mathcal{G}$ in $Iter$ iterations, where $Iter$ is a predefined parameter (cf. Section 2.2). Additionally, we propose a Disentangled Multiscale Cross-attention Transformer (DMCT) model to capture the multiscale seasonality and trend information and use it as a base model for FlowNet (cf. Section 2.3). Details for each part are described below, and pseudocodes can be found in Appendix F.

## 2.2 THE LOCAL AND GLOBAL LEARNING PHASES

**Local phase - independent learning.** As described above, the local phase aims to train a set $F$ of all per-station and cross-station models that are capable of independently predicting flow outcomes for each station. These models provide multiple diverse views on future flow values for each station.

First, for each station $S_i$, we construct a per-station model $f_i$ and train it independently using all of it available local time series data $X^i \in \mathbb{R}^{T_i \times C_i}$. Intuitively, having more related historical hydrology data can help the model to have better generalization, thus effectively coping with climate changes as pointed out in (Milly et al., 2008). FlowNet, with its independent learning scheme, can help us to do so without having to reduce data to match other stations like (Kirschstein & Sun, 2024; Zhou et al., 2025). Each $f_i$ is trained using the loss function $\mathcal{L}^i = Loss(\hat{y}^i, y^i)$, where $Loss$ is the MAE loss.

Second, we train two sets of cross-station models for each station $S_i$, including the *inflow* and *outflow* models. For each station $S_j \in parents(S_i)$, we create a model $f_{i,j}$. However, we cannot train them independently like the per-station ones, but in groups due to their latent outputs. That also means the training data period will now be restricted to $K^i_{inflow} = min_{S_j \in parents(S_i) \cup S_i}(T_j)$, i.e. nearest $K^i_{inflow}$ time points. Following Definition 3, the local inflow loss will be defined as $\mathcal{L}^i_{Local} = Loss(\hat{y}^i_{inflow}, y^i)$, where $\hat{y}^i_{inflow} = \sum_{S_j \in parents(S_i)} \hat{y}^{i,j}$. The *outflow* cross-station models are trained similarly.

**Global phase - interactive learning.** In the global phase, all models will interact with others to adjust their prediction outcomes, thus maximizing their agreements. That can incoporate diverse but consistent views into each station, thus leading to performance improvements as shown in Section 3. Here, we limit the interaction to nearest adjacent neighbors in the graph $\mathcal{G}$ only to reduce computation overhead. For each station $S_i$, we have a set of 3 prediction outcomes: $\hat{y}^i$ from the per-station model, $\hat{y}^i_{inflow}$ from inflow cross-station models and $\hat{y}^i_{outflow}$ from outflow cross-station models. The unified/ensembled prediction outcome of station $S_i$ will be:

$$\hat{y}^i_{Global} = mean(\hat{y}^i_{inflow}, \hat{y}^i_{outflow}) \tag{1}$$

We then define a global losses between the ground truth $y^i$, local prediction $\hat{y}^i$ and global prediction $\hat{y}^i_{Global}$ to update the per-station model $f_i$ as following:

$$\mathcal{L}^i_{Global} = \alpha \cdot Loss(\hat{y}^i, y^i) + (1 - \alpha) \cdot Loss(\hat{y}^i, \hat{y}^i_{Global}) \tag{2}$$

where the first term denotes the difference between the ground truth and the prediction result of the model, and the second term denotes the difference between the predicted results of the model and the global result. The purpose is to minimize their differences, thus balancing final prediction outcomes from diverse views. Note that the update process will be restricted to the training period of $K^i_{Global} = min(K^i_{inflow}, K^i_{outflow})$ nearest time points, and if both inflow and outflow models do not exist, $\mathcal{L}^i_{Global}$ will be equivalent to the per-station loss $\mathcal{L}^i$.

Similarly, we do the global update for all cross-station models via the ground truth $y^i$, their own aggregated result $\hat{y}^i_{inflow}$ or $\hat{y}^i_{outflow}$, and the global result $\hat{y}^i_{Global}$.

## 2.3 DISENTANGLED MULTISCALE CROSS-ATTENTION TRANSFORMER

We propose the **D**isentangled **M**ultiscale **C**ross-attention **T**ransformer (DMCT) as the backbone model to extract the seasonality-trend temporal features efficiently in streamflow prediction. Recent studies, such as DLinear (Zeng et al., 2023), utilizes the seasonality-trend decomposition (STD) method to disentangle the original time series $x^i$ of $S^i$ into seasonality $\mathbf{s}^i$ and trend $\mathbf{t}^i$ independently. For simplicity, we omit the station notation $i$ in the following contents unless otherwise stated.

$$\mathbf{s}, \mathbf{t} = \text{SeriesDecomp}(x) \tag{3}$$

Next, we utilize a stacked Multiscale Cross-attention Transformer Block (MCTB) to capture the temporal features with multiscale information which has been proven to be efficient (Wang et al., 2024b; Zhang et al., 2025). Firstly, we decompose the original sequences seasonality $\mathbf{s}$ and trend $\mathbf{t}$ into multiscale subsequences: $\{\mathbf{s}^0, \dots, \mathbf{s}^l\}$ and $\{\mathbf{t}^0, \dots, \mathbf{t}^l\}$ by down sampling method from TimeMixer (Wang et al., 2024b). Then we based on the distinct properties of seasonality and trend, we apply an independent Cross-attention Transformer block from CATS (Kim et al., 2024) to seasonality and a Linear layer to trend separately. And to balance the efficiency and performance, we apply a Linear layer to extract the multiscale information. At last, we use the concatenate operation to mix the multiscale latent sequences and project to the future sequence length by one Linear layer. Overall, the process of MCTB with $l$ levels is as follows.

$$\{\mathbf{s}^0, \dots, \mathbf{s}^l\} = \text{DownSampling}(\mathbf{s}), \ \{\mathbf{t}^0, \dots, \mathbf{t}^l\} = \text{DownSampling}(\mathbf{t}) \tag{4}$$

$$\mathbf{h} = \text{Concat}(\text{MCTB}(\mathbf{h}_\mathbf{s}^\ell) + \text{Linear}(\mathbf{h}_\mathbf{t}^\ell)), \ \ell = 0, \dots, l \tag{5}$$

$$\hat{y} = \text{Projection}(\mathbf{h}) \tag{6}$$

Additionally, we adapt the instance normalization (Kim et al., 2022) to FlowNet when models learn interactively. Concretely, we use the mean and standard deviation values from the target station $S_i$ to replace the ones of the input time series from the cross-station series $x^{i,j}$ and predicted output $\hat{y}^{i,j}$ as follows:

$$x^{i,j} = \frac{x^{i,j} - \mu_i}{\sqrt{\sigma_i + \epsilon}}, \ \hat{y}^{i,j} = \hat{y}^{i,j} \times \sqrt{\sigma_i + \epsilon} + \mu_i \tag{7}$$

where $\mu_i$ and $\sigma_i$ denote the mean and variance of the target station time series $x^i$, respectively, and $\epsilon$ is a small constant for numerical stability.

## 2.4 GRAPH LINKS RECONSTRUCTION MODULE

As described above, the downstream flow graph $\mathcal{G}$ provides direct relationships among stations. However, not all connected stations contribute effectively to predictive performance due to factors such as long geographical distances or human interventions (e.g., dams or irrigation systems). For example, the two linked stations, Kontum and Stung Treng, in the MRB are nearly 400km away from each other. Thus, their relationship is weaker. To address this, we propose a two-step scheme that detects and retains only strongly correlated and beneficial links to improve predictive accuracy.

**Pearson Correlation Analysis (PearCorr).** We utilize the Pearson correlation coefficient to analyze the lag correlations between two adjacent stations $S_i$ and $S_j$ in the graph $\mathcal{G}$. We repeatedly shift the related time series $\omega$ time steps and calculate the Pearson correlation coefficient until reaching the maximum lag threshold $\Omega$. Let $\lambda_{i,j}$ be the maximum correlations under $\Omega$ lags (default $\Omega = L$).

$$\lambda_{i,j} = \max(\text{PearCorr}(\text{Shift}(X^i_{:,c}, \omega), X^j_{:,c})), \ \omega = 0, \dots, \Omega \tag{8}$$

**Validation-based Links Reconstruction (VLR).** Additionally, we propose a validation-based links reconstruction scheme to refine the adjacent matrix $A \in \mathbb{B}^{N \times N}$. Given two adjacent stations $S_i$ and $S_j$ with $A_{j,i} = 1$, we construct two model $f_{i,j}$ that use data from $S_j$ to predict $S_i$ and $f_i$ that predict $S_i$ via its local data. We keep $A_{j,i} = 1$ if $S_i$ and $S_j$ have strong enough correlation and $f_{i,j}$ has close performance to $f_i$ on the validation set (indicated by the loss $\mathcal{L}_{\text{vali}}(.)$).

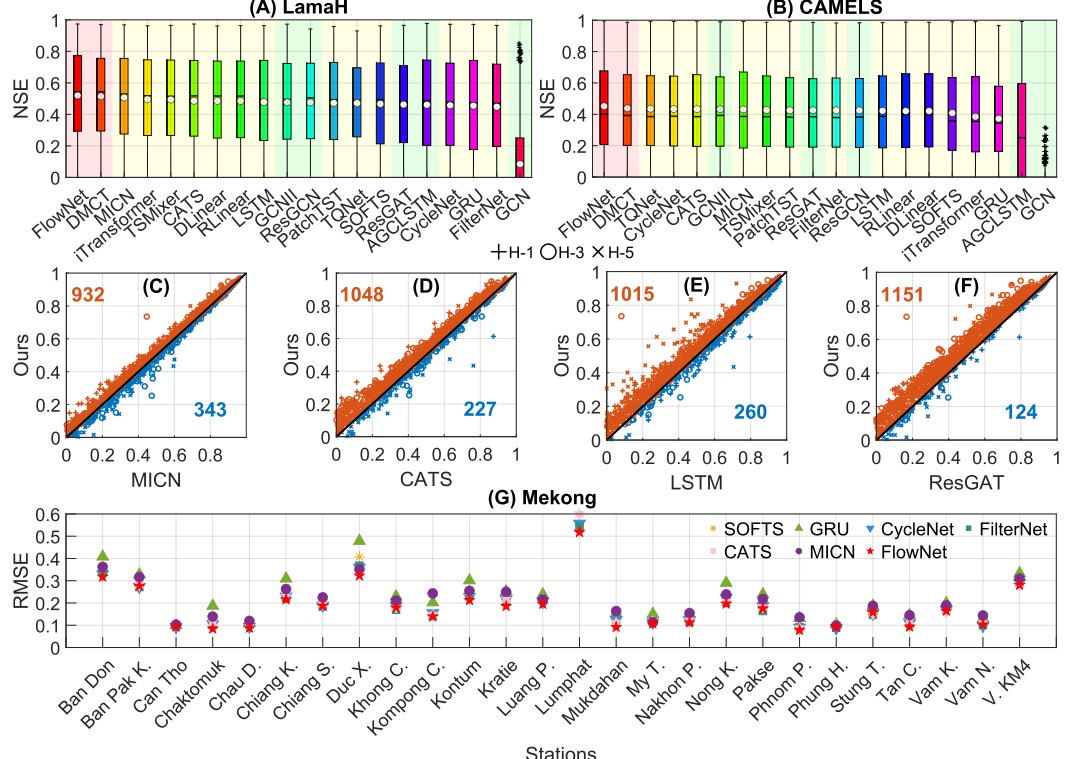

Figure 2: Overall performance of FlowNet for $H \in \{1, 3, 5\}$. (A) Full NSE results of all studied models on CAMELS (sorted in descending order of mean values). (B) Full NSE results of all studied models on LamaH (sorted). (C-F) NSE comparisons between FlowNet and 4 selected baselines on LamaH. A point over the diagonal line indicates that FlowNet is better. (G) RMSEs of FlowNet compared to some baselines on 26 stations of the Mekong dataset ($H = 1$).

$$A_{i,j} = \begin{cases} 1 & \text{if } \mathcal{L}_{\text{vali}}(f_{i,j}) < \gamma \cdot \mathcal{L}_{\text{vali}}(f_i) \text{ and } \lambda_{i,j} > \phi, \\ 0 & \text{otherwise} \end{cases} \tag{9}$$

where $\phi$ (default 0.7) and $\gamma$ (default 1.2) are two predefined thresholds.

## 3  EXPERIMENTS

**Benchmarks.** We demonstrate the performance of FlowNet for two main tasks: (i) predicting daily water discharges for 425 hydrology stations in the Central Europe from the LamaH-CE dataset (Klingler et al., 2021) and 671 stations in the USA from the CAMELS dataset (Newman et al., 2015) and (ii) predicting daily water levels for 26 stations in the Mekong River Basin (MRB) collected from the Mekong River Commission (MRC). Please refer to Appendix A for detail descriptions.

**Baselines.** We compare our method with 18 state-of-the-art (SOTA) baselines including: Transformer-based models (CATS (Kim et al., 2024), iTransformer (Liu et al., 2023), PatchTST (Nie et al., 2023)), MLP-based models (DLinear (Zeng et al., 2023), RLinear (Li et al., 2023), CycleNet (Lin et al., 2024), TQNet (Lin et al., 2025), FilterNet (Yi et al., 2024), SOFTS (Han et al., 2024), TSMixer (Chen et al., 2023)), CNN-based model (MICN (Wang et al., 2023)), GNN-based models (GCN, GCNII, ResGCN, ResGAT (Kirschstein & Sun, 2024)), and RNN-based models (LSTM (Kratzert et al., 2018), GRU (Chung et al., 2014)) and a hybrid model AGCLSTM (Feng et al., 2022). Among them, GNN, GCNII, ResGCN, ResGAT, LSTM and especially AGCLSTM are specifically tailored for streamflow forecasting in (Kratzert et al., 2018; Kirschstein & Sun, 2024; Feng et al., 2022).

**Evaluation metrics and experimental settings.** We use the Nash–Sutcliffe model efficiency coefficient (NSE), the mean absolute error (MAE) and the root mean squared error (RMSE) as metrics to evaluate the performance for all baselines (c.f. Appendix B for details).

Table 2: Mean NSE results of 18 selected SOTA baselines compared to our methods DMCT and FlowNet on 3 datasets with 3 different prediction horizon settings $H \in \{1, 3, 5\}$. Best results are highlighted in bold and second best results are underlined.

| Dataset | CAMELS | | | LamaH | | | Mekong | | |
|---|---|---|---|---|---|---|---|---|---|
| Horizon | 1 | 3 | 5 | 1 | 3 | 5 | 1 | 3 | 5 |
| CycleNet | 0.5587 | 0.4058 | 0.3361 | 0.6012 | 0.4276 | 0.3433 | 0.9268 | 0.8818 | 0.8476 |
| TQNet | 0.5629 | 0.4070 | 0.3381 | 0.5686 | 0.4686 | 0.3779 | 0.9309 | 0.8862 | 0.8530 |
| DLinear | 0.5330 | 0.3955 | 0.3342 | 0.6100 | 0.4587 | 0.3871 | 0.9315 | 0.8889 | 0.8545 |
| RLinear | 0.5322 | 0.3944 | 0.3362 | 0.6100 | 0.4549 | 0.3906 | 0.9269 | 0.8825 | 0.8472 |
| FilterNet | 0.5409 | 0.3935 | 0.3418 | 0.5922 | 0.4153 | 0.3385 | 0.9248 | 0.8815 | 0.8483 |
| iTransformer | 0.4857 | 0.3840 | 0.2861 | 0.6349 | 0.4698 | 0.3805 | 0.9217 | 0.8822 | 0.8431 |
| PatchTST | 0.5574 | 0.4048 | 0.3150 | 0.6121 | 0.4379 | 0.3672 | 0.9322 | 0.8884 | 0.8427 |
| CATS | 0.5565 | 0.4087 | 0.3339 | 0.6315 | 0.4376 | 0.3903 | 0.9206 | 0.8773 | 0.8359 |
| LSTM | 0.5368 | 0.3978 | 0.3338 | 0.6442 | 0.4563 | 0.3356 | 0.8708 | 0.8401 | 0.8130 |
| GRU | 0.4733 | 0.3453 | 0.2984 | 0.6427 | 0.4606 | 0.2624 | 0.8862 | 0.8479 | 0.8230 |
| MICN | 0.5493 | 0.4071 | 0.3372 | 0.6370 | 0.4837 | 0.3992 | 0.9111 | 0.8766 | 0.8431 |
| SOFTS | 0.5291 | 0.3772 | 0.3187 | 0.5675 | 0.4557 | 0.3769 | 0.9198 | 0.8774 | 0.8420 |
| TSMixer | 0.5657 | 0.3949 | 0.3224 | 0.6407 | 0.4577 | 0.3841 | 0.9287 | 0.8838 | 0.8487 |
| GCN | -0.069 | -0.071 | -0.072 | 0.1159 | 0.0715 | 0.0657 | 0.8151 | 0.7815 | 0.7577 |
| GCNII | 0.5565 | 0.4064 | 0.3345 | 0.5975 | 0.4463 | 0.3863 | 0.8813 | 0.8320 | 0.8029 |
| ResGCN | 0.5426 | 0.4005 | 0.3331 | 0.5965 | 0.4387 | 0.3917 | 0.8829 | 0.8284 | 0.7979 |
| ResGAT | 0.5414 | 0.4019 | 0.3335 | 0.6089 | 0.4062 | 0.3721 | 0.8776 | 0.8207 | 0.7891 |
| AGCLSTM | 0.1956 | -0.143 | -0.069 | 0.5966 | 0.4408 | 0.3458 | 0.8876 | 0.8585 | 0.7727 |
| DMCT | 0.5631 | 0.4093 | 0.3422 | 0.6503 | 0.4876 | 0.4060 | 0.9309 | 0.8889 | 0.8530 |
| FlowNet | **0.5784** | **0.4228** | **0.3540** | **0.6598** | **0.4928** | **0.4067** | **0.9323** | **0.8908** | **0.8555** |

## 3.1 Main Results

Figure 2 (A) show NSE values of all studied methods over all 425 stations and 3 prediction horizons for LamaH. FlowNet acquires the best overall performances compared to other baselines, following by DMCT. The same results can be observed in (B) for CAMELS. In (C, D, E, F), we directly compare NSEs of FlowNet with top baselines for all 425 stations of LamaH over 3 prediction horizons. FlowNet outperforms ResGAT, a recent GNN-based method, on 1,151/1,275 cases (90.2%) and LSTM, the most success model for streamflow forecasting, on 1,015/1,275 (79.6%). Moreover, major points are far from the diagonal line, indicating that FlowNet has much better performances than others. (G) shows RMSEs of FlowNet compared to selected baselines for all 26 stations of the Mekong dataset with $H = 1$. FlowNet consistently outperforms others in major cases.

Additionally, as shown in Table 2, FlowNet has mean NSEs of (0.932, 0.890, 0.855) for $H = (1, 3, 5)$ over 26 stations in the Mekong dataset, which are much better than (0.882, 0.828, 0.797) of ResGCN, the best performed GNN-based approach in (Kirschtein et al. 2024). It also outperforms all baselines in terms of mean NSEs over 9 combination cases of datasets and prediction horizons. Full results can be found in Tables 8, 9, and 10 in Appendix C. For RMSE, FlowNet is the best method on 6 and second best method on 2 cases. For MAE, it acquires 4 best and 4 second best over 9 cases.

## 3.2 Ablation Studies and Analyses

**Effects of the local global interaction scheme.** To understand effects of FlowNet's interaction scheme (c.f. Section 2.2), we conduct ablation experiments on 6 stations of the Mekong dataset with significant correlation links from other stations. As shown in Figure 3 (A), the backbone model can benefit significantly from FlowNet (with both local and global phases) with most of the selected stations having the best NSE values, compared to the sole use of the local or global phase. This means interactions among stations help to reduce uncertainty, thus increasing performances as we discussed in Section 2.1.

Table 3: Effects of different components of DMCT on the Mekong dataset, where w/o D, w/o M, w/o IN denote without Disentangled, Multiscale and Instance Normalization modules, respectively.

| | DMCT | w/o D | w/o M | w/o IN |
|---|---|---|---|---|
| H=1 | **0.930** | 0.920 | 0.927 | -0.18 |
| H=3 | **0.888** | 0.877 | 0.887 | -1.81 |
| H=5 | **0.853** | 0.843 | 0.852 | -2.45 |

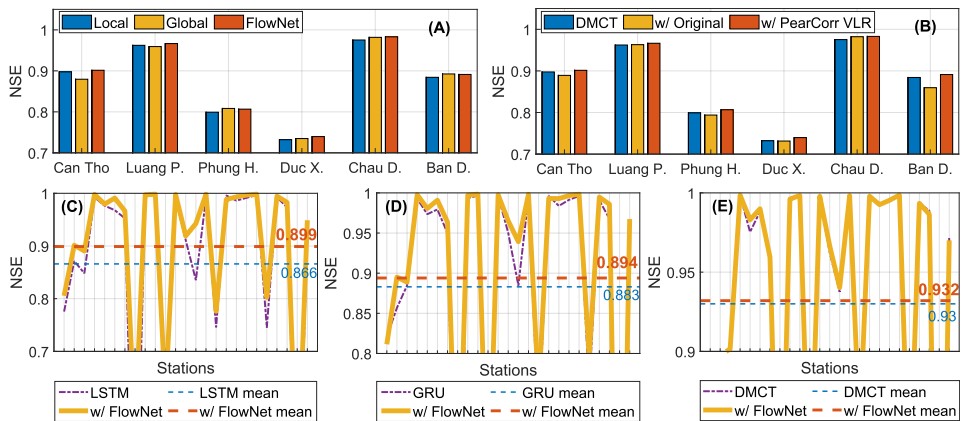

Figure 3: Ablation studies on the Mekong data. (A) Performances of FlowNet when using with Local only, Global only, and Local-Global scheme on 6 selected stations. (B) Performances of DMCT, FlowNet with the original graph $\mathcal{G}$ and FlowNet with our graph reconstruction strategy on 6 selected stations. (C-E) Performances of FlowNet on all 26 stations when being used with different barebones including LSTM, GRU and our proposed model DMCT.

**Effects of the graph link reconstruction.** In Figure 3 (B), we study deeper on our proposed graph link reconstruction approach, which contains two steps: PearCorr and VLR (c.f. Section 2.4). While using the original graph $\mathcal{G}$ does not lead to clear performance improvements compared to DMCT since stations in the Mekong dataset are far from others (c.f. Figure 5 in Appendix E), thus weakening their relationships. The graph refinement module helps to boost performances in all cases compared to DMCT since weak links are removed, thus making the interactions among models more effective (c.f. refined graphs in Figures 9 and 10 in Appendix E).

**Effects of different backbone models.** We employ different DL architectures as backbone models for FlowNet, including LSTM, GRU, and our DMCT backbone. Figure 3 (C) shows NSEs for all 26 stations for LSTM and LSTM with FlowNet. As we see, FlowNet significantly boosts the NSE scores of most stations compared to those of LSTM. The mean NSE of FlowNet thus increases to 0.899 from 0.866 of LSTM. The same results can be observed on GRU and DMCT in (D, E), respectively. These demonstrates the generality and flexibility of FlowNet for boosting performances of its employed barebones via its model interactive scheme.

**Effects of different components of DMCT.** Table 3 demonstrates that Disentangled module, Multi-scale module and Instance Normalization are necessary for accurate prediction performance. Notably, without Instance Normalization the model cannot capture the accurate temporal features due to temporal distribution shift (Kim et al., 2022).

**Computational analysis.** Table 4 shows computational costs of FlowNet and other baselines. FlowNet requires additional cross-station models and interaction overheads. Hence, it is more computationally expensive than its barebone DMCT and other lightweight models lie DLinear. But it is less costly than SOFTS or MICN in terms of memory and parameters. Nevertheless, in exchange for its high computation cost, FlowNet significantly outperforms others in terms of prediction accuracy as shown above.

**Hyperparameter sensitivity.** We study the sensitivity of all hyperparameters for FlowNet and DMCT in Figure 7 in Appendix D such as the learning rate, the dimension of the hidden layer, the number of multiscale levels, the regulation parameter $\alpha$, number of global iterations ($Iter$), and look back window size ($L$).

Table 4: Memory consumptions, times per epoch, and parameters for different methods on Can Tho station of the Mekong datasets with lookback length $L = 32$ and predict horizon $H = 5$.

| Method | Memory(MB) | Time(s) | Parameters |
|---|---|---|---|
| DLinear | 262 | 0.56 | 0.3K |
| LSTM | 374 | 0.71 | 68K |
| GRU | 358 | 0.64 | 51K |
| iTransformer | 332 | 1.04 | 0.4M |
| PatchTST | 368 | 1.13 | 0.4M |
| MICN | 404 | 1.14 | 1.2M |
| TSMixer | 262 | 0.78 | 9.2K |
| SOFTS | 436 | 0.88 | 0.9K |
| FilterNet | 292 | 0.62 | 77K |
| CATS | 268 | 0.86 | 0.2M |
| DMCT | 364 | 1.08 | 0.5M |
| FlowNet | 424 | 1.80 | 1.1M |

## 4 RELATED WORKS

River flow forecasting plays a critical role in flood management, water resource optimization, infrastructure protection, and climate resilience and has attracted many research efforts such as (Feng et al., 2021; Kratzert et al., 2019; Zhou et al., 2025; Bindas et al., 2024; Song et al., 2025; Eddin et al., 2025; Wang et al., 2024a; Kratzert et al., 2024).

Traditional streamflow forecasting methods such as ARIMA, Multiple Linear Regression (MLR), and Moving Average (MA), which assume linearity and thus often underperform in capturing the nonlinear and chaotic behavior of hydrological systems (Sivakumar, 2009). Recently, RNN-based models like LSTM and GRU have demonstrated superior performance in capturing nonlinear temporal dependencies in flood forecasting tasks (Kratzert et al., 2018). Enhancements such as attention-based LSTMs further improve accuracy by focusing on influential time steps (Ding et al., 2020). However, these models typically treat stations as independent time series and often ignore the spatial interactions between them, which are critical in hydrological systems with directional flow structures. FlowNet, in contrast, aims to predict all stations jointly and use their spatial interaction to improve results.

GNNs are well-suited for hydrological networks, where stations can be represented as nodes and river flows as directed edges. Foundational models such as GCN (Kipf & Welling, 2017), ChebNet (Defferrard et al., 2016), and GraphSAGE (Hamilton et al., 2017) provided mechanisms for learning spatial representations. These approaches have inspired applications in traffic prediction (Song et al., 2020) and later, hydrology. Recent models such as ST-GCN (Feng et al., 2021) integrates GCNs with LSTM and attention mechanisms to capture spatiotemporal dependencies. (Zhou et al., 2025) propose HCGCN, introducing flow direction and time delays directly into a time-delayed directed graph structure, enhancing realism and performance. But they typically require all stations to have identical features and sequence lengths, and rely on static adjacency matrices that fail to reflect dynamic hydrological influences, limiting their effectiveness across heterogeneous and dynamically evolving hydrological networks. HGCN-based models such as (Jiang et al., 2024) are more data flexible than GNNs-based ones but nodes of the same types still require unified inputs. They also focus on feature propagation among nodes rather than outcomes. In contrast, FlowNet uniquely uses independent models to interact with others for better utilizing arbitrary existing data and boosting performance via exchanging and adjusting prediction outcomes iteratively. It also dynamically refine the underlying graph for more effective interactions among stations.

Recently, many SOTA methods in general time series forecasting have been introduced in the literature and archived SOTA performances on various time series benchmarks such as iTransformer (Liu et al., 2023), FilterNet (Yi et al., 2024), TQNet (Lin et al., 2025), CycleNet (Lin et al., 2024), Informer (Zhou et al., 2021), AutoFormer (Wu et al., 2021), FEDFormer (Zhou et al., 2022), PatchTST (Nie et al., 2023), CAT (Kim et al., 2024), DLinear (Zeng et al., 2023) or RLinear (Li et al., 2023). These methods can also be applied for streamflow forecasting. Hence, we include them into our study to ensure our comparison is not limited only to models specifically designed for hydrology.

## 5 CONCLUSION

We introduce a *first independent and interactive modeling framework*, called FlowNet, for streamflow prediction. FlowNet utilizes a unique local-global scheme that establishes individualized models and introduces interactive mechanisms for multiple heterogeneous station data, and establishes synchronous spatio-temporal dependencies for all stations through iterative interactive learning with a dynamic refined graph and a proposed DMCT basebone. FlowNet acquires superior performances compared to 18 SOTA baselines on 3 large scale benchmarks. It is also highly flexible and can be used with any exiting DL models and available data for each station.

## 6 REPRODUCIBILITY STATEMENT

The LamaH-CE and CAMELS datasets are publicly available, with sources cited in Section 3. The Mekong Water Level dataset and the source code used for all experiments are included in the supplementary material.

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

# Appendices

Our appendices are outlined as follows. We describe the details of the dataset that we used to evaluate our method in A. The experiment setups, including the metrics selection, parameter settings and implementation details are described in B. In C, we first show the results with standard deviation that run on multiple different random seeds to demonstrate the stability of FlowNet compared to other baselines. Then we show the full performance comparison on three datasets (CAMELS, LamaH and Mekong) on all evaluation metrics (NSE, RMSE and MAE) and future prediction horizons $H$. The hyperparameter sensitivity for FlowNet and DMCT is studied in D. We present the refined graphs in E. In F, we provide the pseudo-code for our algorithms. In G, we describe the limitations of our work and we present the broader impact in H. LLM declaration can be found in I.

## Contents

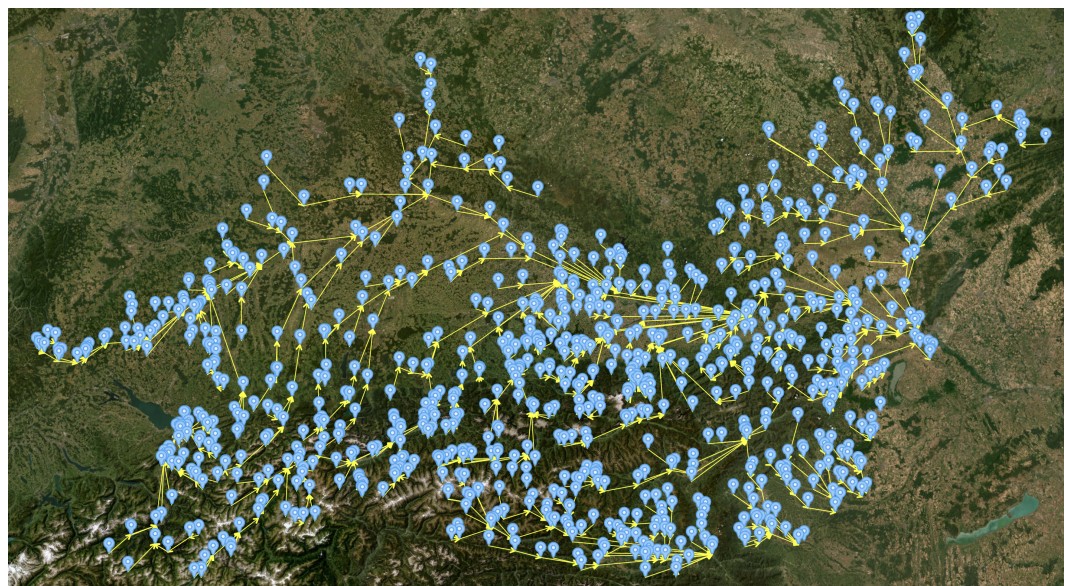

Figure 4: Visualization of the LamaH river network. Each blue marker corresponds to a gauging station from the LamaH dataset, while yellow lines with arrowheads indicate the directed river connections from upstream to downstream.

# A    DATASET DETAILS

## A.1    LAMAH DATASET

The LamaH-CE (Large-Sample Data for Hydrology and Environmental Sciences for Central Europe) dataset, LamaH for short, serves as a critical benchmark for large-scale hydrological modeling, providing standardized data from 859 gauged catchments across Central Europe with daily and hourly resolutions spanning over 35 years of records (Klingler et al., 2021). This dataset consist of long-term hydrological time series, including streamflow, precipitation, and meteorological features with a comprehensive suite of over 60 static catchment attributes, containing topographic indices, climatic statistics, land cover classifications, soil properties, and geological characteristics. Such diversity allows robust exploration of watershed behaviors under varying environmental conditions, particularly supporting machine learning applications in predicting complex hydrological processes.

In our experiments, we extract an 18-year daily subset (2000–2017) from 425 high-quality catchments to ensure temporal consistency and spatial representativeness. The data is partitioned into training (2000–2013), validation (2014–2015), and test sets (2016–2017). We focus on the target variable of daily streamflow, together with four dynamic features including precipitation, topsoil moisture, air temperature, and surface pressure. These features capture key hydrological drivers, with precipitation and moisture informing infiltration-runoff dynamics, while temperature and pressure influence evapotranspiration and atmospheric interactions.

The LamaH dataset provides the streamflow graph inside. Figure 4 shows the station locations and the downstream flow graph of LamaH.

## A.2    CAMELS DATASET

The CAMELS dataset (Catchment Attributes and Meteorology for Large-sample Studies) is a benchmark resource for large-sample hydrological modeling, providing standardized data for 671 catchments in the contiguous United States (CONUS) over the period 1980–2014 and is publicly available (Addor et al., 2017). This dataset contains daily time series of meteorological data (e.g., precipitation, temperature) and streamflow measurements with a comprehensive suite of static catchment attributes, thus allowing the exploration of watershed behavior across diverse climatic and topographic conditions.

Table 5: The summary of Mekong dataset.

| Validation Start Date | Testing Start Date | Validation Data Points | Testing Data Points | Frequency | Target |
|---|---|---|---|---|---|
| 2019/11/1 | 2021/11/1 | 731 | 730 | Daily | Water.Level |

The static attributes contains 35 variables spanning multiple categories: climate indices (e.g., mean precipitation, aridity index, snow fraction), topographic features (e.g., mean elevation, slope area), vegetation characteristics (e.g., forest fraction, leaf area index), soil properties (e.g., soil depth, hydraulic conductivity), and geological traits (e.g., carbonate rock proportion, permeability). These attributes are derived from spatially aggregated data and serve as critical inputs for characterizing catchment heterogeneity. Dynamic features include meteorological variables such as precipitation, temperature, and potential evapotranspiration, which are provided at daily resolutions to capture temporal dynamics. The streamflow data, measured at catchment outlets, are used as the target variable for predictive modeling.

We partition the dataset into a 10-year subset, aligning with common practices in hydrological model evaluation. The training set spans October 1, 1997, to September 30, 2004 (7 years), the validation set covers October 1, 2004, to September 30, 2005 (1 year), and the test set extends from October 1, 2005, to September 30, 2007 (2 years).

Preprocessing steps are applied to address scale disparities across catchments. Following Cai et al. (2024), we normalize streamflow values using catchment area and mean annual precipitation to account for volumetric differences. Additionally, a logarithmic transformation for streamflow varibale $v$ is employed to stabilize variance and handle zero-inflation in flow data:

$$v_o = \log_{10}(\sqrt{v + 0.1}) \tag{10}$$

where $v_o$ represents the transformed streamflow feature, and the constant $0.1$ prevents numerical instability for zero or near-zero values. This transformation enhances model training by improving the homogeneity of input distributions across catchments of varying sizes and regimes.

There is no streamflow graph provided in CAMELS. Hence we use a KNN graph (with $K = 2$) to simulate the flow relationships. The intution is that if two stations are close, they are more likely affecting others.

### A.3 MEKONG WATER LEVEL DATASET

We summarize the overall structure of the Mekong dataset in Table 5, which reports the start dates of the validation and testing sets, the number of data points in each split, the temporal resolution, and the prediction target. Specifically, the dataset adopts a daily frequency, with the validation period starting from November 1, 2019 and the testing period from November 1, 2021, corresponding to 731 and 730 samples, respectively. The forecasting target across all stations is the daily water level.

Table 6 provides fine-grained information for each individual hydrological station. These stations differ substantially in both temporal coverage and sensing modalities: the training periods range from as early as January 1910 (Stung Treng) to as recent as September 2007 (Vam Nao), and the number of training samples varies dramatically from fewer than 5,000 records (Kontum, Vam Nao) to more than 40,000 records (Stung Treng). Furthermore, the input channels are heterogeneous across stations, consisting of one to three variables drawn from water level (WL), water discharge (WD), and rainfall (RF). For example, Chiang Saen provides the richest set of three channels (WL, WD, RF), while many downstream stations such as Kompong Cham or Kratie only include WL.

This variability highlights a key characteristic of the Mekong dataset: it is highly irregular both temporally and spatially. Unlike standardized benchmark datasets with uniform lengths and modalities, the Mekong data reflects the real-world complexities of hydrological monitoring systems, where station deployments differ in historical coverage, measurement availability, and environmental context. Such irregularities make modeling challenging, as methods must handle unbalanced input lengths, heterogeneous channel configurations, and station-specific dynamics. We intentionally include this dataset in our evaluation because it offers a stringent testbed for assessing the flexibility and robustness of different methods. Our proposed method FlowNet that generalizes well across the Mekong dataset

Table 6: The details of each station dataset. WL denotes water level, WD denotes water discharge, and RF denotes rainfall.

| Station | Training Start Date | Training Data Points | Channels | Channel Types |
|---|---|---|---|---|
| Ban Don | 1992/1/1 | 10166 | 2 | WL, RF |
| Ban Pak Kanhoung | 1989/1/1 | 11261 | 1 | WL |
| Can Tho | 1979/4/1 | 14824 | 2 | WL, RF |
| Chaktomuk | 1980/7/1 | 14367 | 1 | WL |
| Chau Doc | 1960/6/29 | 21674 | 2 | WL, RF |
| Chiang Khan | 1967/1/1 | 19297 | 2 | WL, WD |
| Chiang Saen | 1965/12/31 | 19663 | 3 | WL, WD, RF |
| Duc Xuyen | 1985/1/1 | 12722 | 1 | WL |
| Khong Chiam | 1966/1/1 | 19662 | 2 | WL, WD |
| Kompong Cham | 1930/1/1 | 32811 | 1 | WL |
| Kontum | 2007/2/22 | 4635 | 2 | WL, RF |
| Kratie | 1933/1/8 | 31708 | 1 | WL |
| Luang Prabang | 1960/1/1 | 21854 | 1 | WL |
| Lumphat | 2000/5/7 | 7117 | 2 | WL, WD |
| Mukdahan | 1960/1/1 | 21854 | 2 | WL, WD |
| My Thuan | 2006/8/1 | 4840 | 2 | WL, RF |
| Nakhon Phanom | 1972/4/1 | 17380 | 2 | WL, WD |
| Nong Khai | 1969/1/1 | 18566 | 2 | WL, WD |
| Pakse | 1960/1/1 | 21854 | 2 | WL, WD |
| Phnom Penh Port | 1960/1/1 | 21854 | 1 | WL |
| Phung Hiep | 1985/1/1 | 12722 | 1 | WL |
| Stung Treng | 1910/1/1 | 40116 | 2 | WL, WD |
| Tan Chau | 1997/12/31 | 7975 | 2 | WL, RF |
| Vam Kenh | 1992/1/1 | 10166 | 1 | WL |
| Vam Nao | 2007/9/2 | 4443 | 2 | WL, RF |
| Vientiane KM4 | 1923/1/1 | 35368 | 1 | WL |

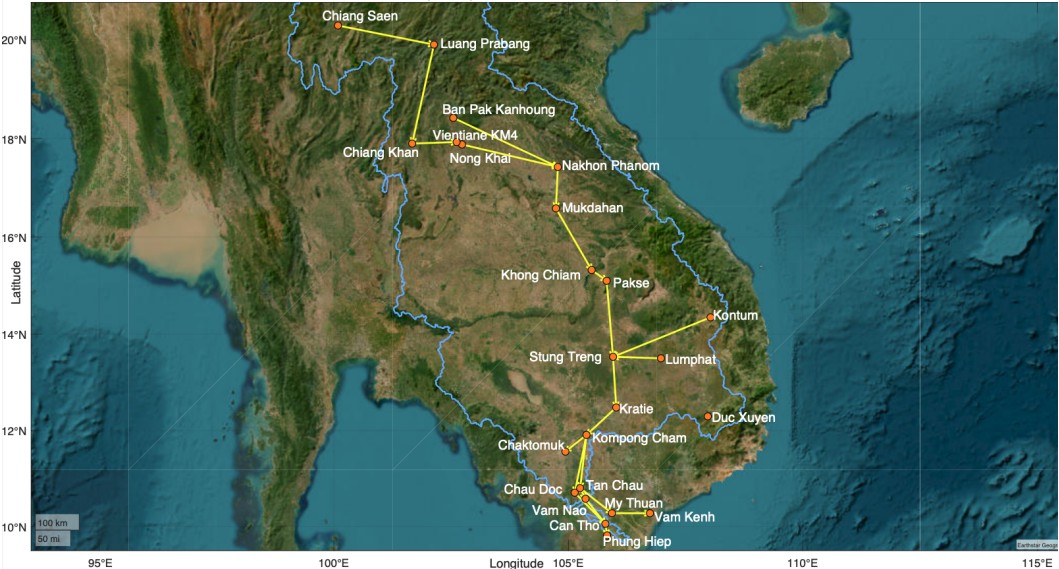

Figure 5: The original streamflow graph for the Mekong datasets. Each dot presents a station, while each arrow indicates the water flow direction.

is more likely to be applicable to other real-world spatiotemporal forecasting problems that share similar irregularities. Figure 5 shows the streamflow graph of the Mekong dataset.

## B  EXPERIMENTS DETAILS

## B.1 EXPERIMENTS SETUP

We evaluate all models using three standard hydrological and machine learning metrics: the Nash–Sutcliffe efficiency coefficient (NSE), root mean squared error (RMSE), and mean absolute error (MAE). These metrics are defined as:

$$\text{NSE} = 1 - \frac{\sum_{t=1}^{H}(y_t - \hat{y}_t)^2}{\sum_{t=1}^{H}(y_t - \bar{y})^2}, \ \text{RMSE} = \sqrt{\sum_{t=1}^{H}\frac{(y_t - \hat{y}_t)^2}{H}} \ , \ \text{MAE} = \frac{1}{H}\sum_{t=1}^{H}|y_t - \hat{y}_t| \quad (11)$$

where $y_t$ denotes the ground truth at time step $t$, $\hat{y}_t$ is the model prediction, and $\bar{y}$ is the temporal mean of $y_t$. Lower RMSE and MAE values indicate better predictive performance, whereas higher NSE values (closer to 1) indicate higher efficiency in reproducing observed dynamics.

All datasets are preprocessed using a z-score normalization,

$$x' = \frac{x - \mu}{\sigma}, \quad (12)$$

where $\mu$ and $\sigma$ denote the mean and standard deviation of the training set, respectively. This ensures that different stations and variables are placed on a comparable scale, which is particularly important for handling heterogeneous hydrological signals.

All experiments are implemented in PyTorch Paszke et al. (2019) and executed on a cluster equipped with 4 NVIDIA L4 GPUs (24GB memory each). We adopt a batch size of 32 for all GNN-based models, while for other baselines we use 128 for the Mekong dataset and 1024 for LamaH and 256 for CAMELS, following their relative dataset sizes. The number of training epochs is fixed to 100. Optimization is performed using the ADAM optimizer Kingma & Ba (2015) with the mean squared error (MSE) loss function.

The input sequence length is set to $L = 32$ for Mekong and $L = 360$ for LamaH and CAMELS, with prediction horizons $H \in \{1, 3, 5\}$ across all baselines. We conduct a systematic hyperparameter search over: hidden dimension in $\{32, 64, 128, 256\}$, number of layers in $\{1, 2, 3\}$, learning rate in $\{10^{-1}, 10^{-2}, 10^{-3}, 10^{-4}\}$, loss regularization factor $\alpha \in [0.1, 0.95]$, number of global iterations Iter $\in [1, 5]$, and global learning rate factor $\beta \in \{0.01, 0.1, 0.2\}$. The best configuration is selected via validation performance. We use the similar search strategies for baselines, number of heads in $\{2, 4, 8\}$, patch and stride length in $\{16, 48, 96, 160, 320\}$, forward function dimension $\{128, 256, 512, 1024\}$, and cycle length in $\{12, 36, 60, 120\}$.

Here we provide the hyperparameter settings for reproducibility. DMCT: learning rate: 0.001, hidden dimension: 256, number of layers: 1. FlowNet: loss regularization factor $\alpha$: 0.95, global learning rate: 0.001, global learning rate factor $\beta$: 0.01, number of global iterations Iter: 5.

## B.2 BASELINES DETAILS

**Transformer-based Models (PatchTST, iTransformer, CATS).** These models represent the most influential architectural developments for long-sequence forecasting. PatchTST introduced patching and channel-independent attention, enabling efficient modeling of long temporal contexts. iTransformer inverted the attention dimension to emphasize cross-variable dependencies, advancing multivariate modeling. CATS further demonstrated that cross-attention only architecture for Transformer enhances inter-series interactions. Collectively, these models define the current SOTA direction for scalable Transformer-based forecasting.

**MLP-based Models (DLinear, RLinear, CycleNet, TQNet, FilterNet, SOFTS, TSMixer).** Recent work has shown that well-designed MLP architectures can match or surpass complex Transformer models while being highly efficient. DLinear, RLinear, Cycle, TQNet and FilterNet revealed the surprising strength of decomposition-based linear layers, reshaping the community's understanding of linear models. Mixer-style models such as TSMixer and SOFTS introduced mixing for channel–temporal interactions, providing strong MLP-based baselines. These models collectively represent the competitive methods with lightweight design in modern time-series forecasting.

**CNN-based Model (MICN).** CNN-based architectures remain highly effective for extracting local temporal patterns and short-term dependencies. MICN exemplifies this line of work by leveraging

convolutional inductive biases to achieve strong performance with high computational efficiency. This class of models forms a crucial comparison point against Transformer- and MLP-based models.

**GNN-based Models (GCN, GCNII, ResGCN, ResGAT).** Including these baselines is important for streamflow tasks where temporal series are coupled through spatial and graph structures. GCN/GCNII enable stable deep graph feature extraction, while ResGCN and ResGAT incorporate residual learning and attention to enhance long-range spatial dependency modeling.

**RNN-based Models (LSTM, GRU).** LSTM and GRU remain widely used benchmarks due to their strong performance on moderate-length dependencies and robustness under limited data. They are extensively adopted in hydrology and environmental modeling, serving as standard baselines for streamflow forecasting. Their inclusion ensures comparability with long-standing literature.

**Hybrid Model (AGCLSTM).** AGCLSTM combines graph convolutions with recurrent dynamics and was specifically designed for streamflow prediction. It jointly models spatial topology and temporal evolution, reflecting domain-specific state-of-the-art modeling practice. This baseline is critical because it targets the same task domain as ours and represents one of the strongest specialized competitors.

## C  FULL RESULTS

### C.1  RESULTS WITH DIFFERENT RANDOM SEEDS

To comprehensively evaluate the robustness and performance of the baseline models, we conducted multiple runs using random seeds $\{43, 44, 45, 46, 47\}$ on the Mekong dataset under a forecasting horizon of $H = 1$. The mean and standard deviation results across 26 stations are summarized in Table 7. Notably, our proposed models, DMCT and FlowNet, exhibit exceptional performance consistency and competitive accuracy compared to other baselines. Specifically, FlowNet achieves the highest Nash-Sutcliffe Efficiency (NSE) of $0.932\pm0.002$, alongside the lowest RMSE ($0.180\pm0.004$) and MAE ($0.114 \pm 0.003$), indicating superior predictive precision with minimal variability across random initializations. Similarly, DMCT attains an NSE of $0.930 \pm 0.002$, with RMSE and MAE values of $0.185 \pm 0.005$ and $0.119\pm0.004$, respectively, demonstrating strong stability and reliability. In contrast, models such as LSTM and GRU show larger standard deviations (e.g., LSTM's NSE std: 0.018), highlighting sensitivity to random seeds, while graph-based approaches like GCN exhibit zero variance due to deterministic architectures but underperform in accuracy. The low standard deviations of DMCT and FlowNet underscore their insensitivity to initialization, a critical attribute for reproducible and deployable time-series forecasting models in practical scenarios.

Table 7: The full results on baselines with different random seed $\{43, 44, 45, 46, 47\}$, results averaged from 26 stations, shown in mean $\pm$ std, with forecasting length $H = 1$.

| Method | NSE | RMSE | MAE |
|---|---|---|---|
| CycleNet | 0.927±0.003 | 0.190±0.004 | 0.124±0.004 |
| TQNet | 0.930±0.002 | 0.182±0.004 | 0.116±0.003 |
| DLinear | 0.931±0.001 | 0.181±0.003 | 0.115±0.002 |
| RLinear | 0.926±0.005 | 0.184±0.005 | 0.117±0.005 |
| FilterNet | 0.926±0.003 | 0.186±0.004 | 0.121±0.004 |
| iTransformer | 0.920±0.003 | 0.195±0.005 | 0.131±0.004 |
| PatchTST | 0.929±0.005 | 0.189±0.006 | 0.124±0.006 |
| CATS | 0.919±0.003 | 0.195±0.005 | 0.128±0.005 |
| LSTM | 0.886±0.018 | 0.252±0.028 | 0.170±0.021 |
| GRU | 0.887±0.015 | 0.249±0.024 | 0.168±0.017 |
| MICN | 0.901±0.016 | 0.228±0.016 | 0.154±0.012 |
| SOFTS | 0.922±0.004 | 0.194±0.006 | 0.130±0.005 |
| TSMixer | 0.926±0.004 | 0.185±0.005 | 0.120±0.004 |
| GCN | 0.815±0.000 | 0.466±0.000 | 0.313±0.000 |
| GCNII | 0.881±0.000 | 0.363±0.000 | 0.241±0.000 |
| ResGCN | 0.883±0.000 | 0.328±0.000 | 0.221±0.000 |
| ResGAT | 0.878±0.000 | 0.384±0.000 | 0.257±0.000 |
| DMCT | 0.930±0.002 | 0.185±0.005 | 0.119±0.004 |
| FlowNet | 0.932±0.002 | 0.180±0.004 | 0.114±0.003 |

Table 8: Forecasting performance on the CAMELS dataset. Our proposed model, FlowNet, demonstrates unequivocal dominance on this benchmark. It achieves the best performance in 8 out of 9 metrics and consistently outperforms the baseline model, DMCT, in all categories. This state-of-the-art performance across all prediction horizons (1, 3, and 5) highlights its superior forecasting capabilities. The best results are marked in red and the second-best in blue.

| Method | Horizon 1 | | | Horizon 3 | | | Horizon 5 | | |
|---|---|---|---|---|---|---|---|---|---|
| Metric | NSE ↑ | RMSE ↓ | MAE ↓ | NSE ↑ | RMSE ↓ | MAE ↓ | NSE ↑ | RMSE ↓ | MAE ↓ |
| CycleNet | 0.5587 | 261.5 | 69.39 | 0.4058 | 323 | 101.4 | 0.3361 | 350.6 | 113.4 |
| TQNet | 0.5629 | 257.6 | 64.85 | 0.407 | 322.6 | 96.21 | 0.3381 | 349.6 | 114.3 |
| DLinear | 0.533 | 258.1 | 68.23 | 0.3955 | 321.8 | 96.97 | 0.3342 | 347.8 | 111.6 |
| RLinear | 0.5322 | 258.2 | 68.34 | 0.3944 | 321 | 97.36 | 0.3362 | 346.3 | 112 |
| FilterNet | 0.5409 | 268.4 | 77.48 | 0.3935 | 331.3 | 102.9 | 0.3418 | 345.9 | 111.6 |
| iTransformer | 0.4857 | 275.4 | 70.99 | 0.384 | 328.5 | 99.82 | 0.2861 | 369.4 | 119 |
| PatchTST | 0.5574 | 260.5 | 67.84 | 0.4048 | 324.5 | 95.39 | 0.315 | 359.2 | 114.5 |
| CATS | 0.5565 | 260.7 | 66.18 | 0.4087 | 321.5 | 99.3 | 0.3339 | 352.1 | 110.6 |
| LSTM | 0.5368 | 270.3 | 70.29 | 0.3978 | 324 | 96.81 | 0.3338 | 347 | 111.1 |
| GRU | 0.4733 | 289.7 | 83.03 | 0.3453 | 337.7 | 108.7 | 0.2984 | 361.6 | 121.8 |
| MICN | 0.5493 | 257.6 | 66.66 | 0.4071 | 320.4 | 94 | 0.3372 | 348.4 | **109.4** |
| SOFTS | 0.5291 | 268.9 | 73.9 | 0.3772 | 329.7 | 99.7 | 0.3187 | 353.6 | 113.7 |
| TSMixer | 0.5657 | 259.5 | 72.72 | 0.3949 | 325.3 | 95.49 | 0.3224 | 356 | 117.1 |
| GCN | -0.0695 | 472.1 | 214 | -0.071 | 472.6 | 214.3 | -0.07241 | 472.9 | 214.6 |
| GCNII | 0.5565 | 259.7 | 70.15 | 0.4064 | 323.4 | 99.01 | 0.3345 | 351.3 | 114.8 |
| ResGCN | 0.5426 | 269.4 | 77.28 | 0.4005 | 325.5 | 100.5 | 0.3331 | 351.3 | 115.2 |
| ResGAT | 0.5414 | 271.9 | 73.81 | 0.4019 | 325.3 | 99.99 | 0.3335 | 350.5 | 115.1 |
| AGCLSTM | 0.1956 | 331 | 76.76 | -0.143 | 391.6 | 105.3 | -0.06911 | 407.8 | 120.1 |
| DMCT | 0.5631 | 257.8 | 68.6 | 0.4093 | 321 | 96.33 | 0.3422 | 348.5 | 112.3 |
| FlowNet | **0.5784** | **250.7** | **64.01** | **0.4228** | **316.2** | **93.97** | **0.354** | **342.8** | 109.9 |

## C.2 Full Results of CAMELS

Table 8 reports the forecasting performance on the CAMELS dataset across three prediction horizons ($H \in \{1, 3, 5\}$) and three evaluation metrics (NSE, RMSE, and MAE). Overall, our proposed FlowNet establishes a new state of the art on this benchmark, achieving the best performance in eight out of nine cases, while consistently outperforming the strongest baseline methods.

At the short-term horizon ($H = 1$), FlowNet delivers the highest accuracy across all metrics, with an NSE of 0.5784, RMSE of 250.7, and MAE of 64.01. These improvements over other baselines demonstrate that FlowNet captures fine-scale hydrological dynamics more effectively than existing temporal or graph-based approaches.

For the medium-term horizon ($H = 3$), FlowNet achieves the best NSE (0.4228) and the lowest RMSE (316.2) and MAE (93.97), establishing clear dominance over other baselines. This result highlights the robustness of FlowNet in modeling non-stationary streamflow patterns over longer lead times.

At the long-term horizon ($H = 5$), FlowNet still achieves the state-of-the-art performance. FlowNet again leads in NSE (0.354) and RMSE (342.8), while ranking second in MAE (109.9), just behind MICN (109.4).

Meanwhile, our backbone model, DMCT, also demonstrates competitive results. It achieves second-best overall performance, outperforming classical sequence models (e.g., LSTM, GRU) and modern Transformer-based architectures (e.g., PatchTST, iTransformer), confirming the effectiveness of its design for hydrological forecasting.

In summary, these results highlight two important findings: (i) DMCT already sets a strong baseline by surpassing a broad range of existing deep learning methods, and (ii) FlowNet further advances the state of the art by achieving the best or second-best results across all horizons and metrics. This demonstrates not only the scalability of our framework but also its capacity to generalize across large-scale and challenging hydrological dataset.

Table 9: Forecasting performance on the LamaH dataset. On this more challenging dataset, FlowNet demonstrates its robust and superior capabilities. It achieves the best performance in 5 out of 9 categories and secures the second-best position in another 3 categories. Furthermore, FlowNet consistently outperforms the baseline model, DMCT, across all metrics and prediction horizons. The best results are highlighted in red and the second-best in blue.

| Method | Horizon 1 | | | Horizon 3 | | | Horizon 5 | | |
|---|---|---|---|---|---|---|---|---|---|
| Metric | NSE ↑ | RMSE ↓ | MAE ↓ | NSE ↑ | RMSE ↓ | MAE ↓ | NSE ↑ | RMSE ↓ | MAE ↓ |
| CycleNet | 0.6012 | 9.116 | 5.047 | 0.4276 | 12.62 | 7.473 | 0.3433 | 14.16 | 8.491 |
| TQNet | 0.5686 | 10.63 | 6.373 | 0.4686 | 11.72 | 6.889 | 0.3779 | 13.7 | 8.378 |
| DLinear | 0.61 | 8.747 | 4.868 | 0.4587 | 11.65 | 6.848 | 0.3871 | 12.93 | 7.875 |
| RLinear | 0.61 | 8.712 | 4.773 | 0.4549 | 11.67 | 6.858 | 0.3906 | 12.86 | 7.813 |
| FilterNet | 0.5922 | 9.547 | 5.659 | 0.4153 | 12.49 | 7.562 | 0.3385 | 13.75 | 8.665 |
| iTransformer | 0.6349 | 8.567 | 4.679 | 0.4698 | 11.76 | 6.631 | 0.3805 | 13.17 | 7.718 |
| PatchTST | 0.6121 | 8.938 | 4.664 | 0.4379 | 12.22 | 6.678 | 0.3672 | 13.29 | 7.846 |
| CATS | 0.6315 | 8.515 | 4.399 | 0.4376 | 12.18 | 6.641 | 0.3903 | 13.05 | 7.8 |
| LSTM | 0.6442 | 8.078 | 4.395 | 0.4563 | 11.55 | 6.841 | 0.3356 | 13.58 | 8.738 |
| GRU | 0.6427 | **7.741** | 4.261 | 0.4606 | 11.63 | 7 | 0.2624 | 14.77 | 9.663 |
| MICN | 0.637 | 8.309 | 4.395 | 0.4837 | 11.42 | 6.573 | 0.3992 | 12.95 | 7.597 |
| SOFTS | 0.5675 | 9.587 | 5.674 | 0.4557 | 11.88 | 6.728 | 0.3769 | 13.21 | 8.197 |
| TSMixer | 0.6407 | 8.322 | 4.35 | 0.4577 | 11.75 | 6.553 | 0.3841 | 13.16 | 7.672 |
| GCN | 0.1159 | 19.21 | 13.61 | 0.07155 | 19.57 | 13.9 | 0.06577 | 19.58 | 13.82 |
| GCNII | 0.5975 | 9.088 | 5.202 | 0.4463 | 12.19 | 7.33 | 0.3863 | 13.12 | 8.127 |
| ResGCN | 0.5965 | 9.323 | 5.41 | 0.4387 | 12.2 | 7.519 | 0.3917 | 12.92 | 7.883 |
| ResGAT | 0.6089 | 8.723 | 4.847 | 0.4062 | 12.92 | 8.299 | 0.3721 | 13.52 | 8.591 |
| AGCLSTM | 0.5966 | 7.918 | **4.101** | 0.4408 | 11.36 | **5.966** | 0.3458 | 13.09 | **7.252** |
| DMCT | 0.6503 | 8.002 | 4.389 | 0.4876 | 11.34 | 6.723 | 0.406 | 12.73 | 7.613 |
| FlowNet | **0.6598** | 7.792 | 4.385 | **0.4928** | **11.25** | 6.514 | **0.4067** | **12.7** | 7.536 |

## C.3 FULL RESULTS OF LAMAH

Table 9 reports the forecasting performance on the LamaH dataset, which is more challenging due to its complex hydrological dynamics and diverse catchment characteristics. Several important trends can be observed.

First, our backbone model DMCT provides a consistently strong benchmark across horizons. It secures the second-best NSE at all horizons (0.6503, 0.4876, 0.406 for $H = 1, 3, 5$), and also ranks among the top models in terms of RMSE and MAE. These results confirm DMCT's ability to effectively capture temporal dependencies and spatial heterogeneity in river basins.

Second, our proposed FlowNet achieves state-of-the-art performance across the majority of metrics. Specifically, FlowNet attains the best NSE at all horizons (0.6598, 0.4928, 0.4067), highlighting its ability to capture flow dynamics more accurately than all competitors. In terms of error-based metrics, FlowNet also secures the best RMSE at horizon 3 (11.25) and horizon 5 (12.7), while maintaining second-best performance at several other positions (e.g., RMSE 7.792 at horizon 1, MAE 6.514 at horizon 3, and MAE 7.536 at horizon 5). Notably, FlowNet consistently outperforms DMCT across all metrics and horizons, confirming robustness and superior generalization ability of the local-global framework of FlowNet.

In summary, FlowNet demonstrates clear dominance on LamaH, achieving the best results in 5 out of 9 categories and ranking second-best in another 3. This consistent advantage over both DMCT and advanced baselines highlights FlowNet's effectiveness in tackling the challenges posed by large-scale and heterogeneous hydrological forecasting tasks.

## C.4 FULL RESULTS OF MEKONG

Table 10 reports the forecasting results on the Mekong Water Level dataset. The results consistently highlight the superiority of our proposed FlowNet across different prediction horizons. FlowNet attains the best performance in 6 out of 9 evaluation categories and ranks second in an additional 2 cases, demonstrating both robustness and generalization across metrics.

Table 10: Forecasting performance on the Mekong Water Level dataset. The results clearly demonstrate the dominance of our proposed model, FlowNet. It achieves the best performance in a remarkable 6 out of 9 categories and secures a second-best position in 2 other categories. Furthermore, FlowNet consistently outperforms the baseline model, DMCT, across all metrics and prediction horizons, establishing a new state-of-the-art on this dataset. The best results are highlighted in red and the second-best in blue.

| Method | Horizon 1 | | | Horizon 3 | | | Horizon 5 | | |
|---|---|---|---|---|---|---|---|---|---|
| Metric | NSE ↑ | RMSE ↓ | MAE ↓ | NSE ↑ | RMSE ↓ | MAE ↓ | NSE ↑ | RMSE ↓ | MAE ↓ |
| CycleNet | 0.9268 | 0.1916 | 0.1245 | 0.8818 | 0.3052 | 0.1937 | 0.8476 | 0.384 | 0.2429 |
| TQNet | 0.9309 | 0.1831 | 0.1168 | 0.8862 | 0.2952 | 0.1848 | 0.853 | 0.3786 | 0.2378 |
| DLinear | 0.9315 | 0.1821 | 0.1153 | 0.8889 | 0.2965 | 0.1839 | 0.8545 | 0.3776 | **0.2345** |
| RLinear | 0.9269 | 0.1837 | 0.118 | 0.8825 | 0.2991 | 0.1876 | 0.8472 | 0.3817 | 0.2389 |
| FilterNet | 0.9248 | 0.1886 | 0.1225 | 0.8815 | 0.2988 | 0.1894 | 0.8483 | 0.3812 | 0.2414 |
| iTransformer | 0.9217 | 0.1955 | 0.1314 | 0.8822 | 0.3009 | 0.1948 | 0.8431 | 0.3783 | 0.2456 |
| PatchTST | 0.9322 | 0.1876 | 0.1222 | 0.8884 | 0.3003 | 0.191 | 0.8427 | 0.3858 | 0.2465 |
| CATS | 0.9206 | 0.1962 | 0.1301 | 0.8773 | 0.3045 | 0.1932 | 0.8359 | 0.3911 | 0.2498 |
| LSTM | 0.8708 | 0.2612 | 0.1775 | 0.8401 | 0.3541 | 0.2337 | 0.813 | 0.4224 | 0.2778 |
| GRU | 0.8862 | 0.2416 | 0.1631 | 0.8479 | 0.3498 | 0.231 | 0.823 | 0.4156 | 0.2725 |
| MICN | 0.9111 | 0.2236 | 0.1505 | 0.8766 | 0.3188 | 0.2063 | 0.8431 | 0.3953 | 0.2547 |
| SOFTS | 0.9198 | 0.1938 | 0.1311 | 0.8774 | 0.3056 | 0.1977 | 0.842 | 0.3814 | 0.247 |
| TSMixer | 0.9287 | 0.1836 | 0.1185 | 0.8838 | 0.3007 | 0.1908 | 0.8487 | 0.3835 | 0.2418 |
| GCN | 0.8151 | 0.3134 | 0.4656 | 0.7815 | 0.384 | 0.562 | 0.7577 | 0.3862 | 0.5778 |
| GCNII | 0.8813 | 0.2407 | 0.3633 | 0.832 | 0.3126 | 0.4667 | 0.8029 | 0.3493 | 0.5257 |
| ResGCN | 0.8829 | 0.2211 | 0.3285 | 0.8284 | **0.291** | 0.4388 | 0.7979 | **0.3287** | 0.4999 |
| ResGAT | 0.8776 | 0.2566 | 0.3836 | 0.8207 | 0.3289 | 0.489 | 0.7891 | 0.3701 | 0.5499 |
| AGCLSTM | 0.8876 | 0.2315 | 0.1318 | 0.8585 | 0.3276 | 0.2024 | 0.7727 | 0.4253 | 0.2591 |
| DMCT | 0.9309 | 0.1853 | 0.1198 | 0.8889 | 0.2985 | 0.1868 | 0.853 | 0.3802 | 0.2391 |
| FlowNet | **0.9323** | **0.1796** | **0.1144** | **0.8908** | 0.2945 | **0.1835** | **0.8555** | 0.3757 | 0.2354 |

At the short-term horizon (Horizon 1), FlowNet achieves the highest NSE (0.9323), the lowest RMSE (0.1796), and the lowest MAE (0.1144), surpassing all competing methods and setting a new benchmark for near-future water level forecasting. For medium-term prediction (Horizon 3), FlowNet continues to lead with the best NSE (0.8908) and MAE (0.1835), while also securing the second-best RMSE (0.2945). Even at the long-term horizon (Horizon 5), which poses greater forecasting challenges, FlowNet maintains its advantage with the best NSE (0.8555) and competitive error values (RMSE: 0.3757, MAE: 0.2354).

Importantly, FlowNet consistently outperforms the backbone model DMCT across all horizons, underscoring its robustness and adaptability. While other models occasionally attain competitive results in isolated metrics (e.g., DLinear in MAE at Horizon 5 or ResGCN in RMSE at longer horizons), they fail to exhibit the same level of stability across horizons. In contrast, FlowNet's dominance across both accuracy (NSE) and error-based metrics (RMSE, MAE) highlights its capacity to provide reliable predictions under varying forecasting horizon settings.

Overall, these results establish FlowNet as the state of the art for hydrological forecasting on the Mekong Water Level dataset, combining short-term precision with long-term stability.

## C.5    RESULTS COMPARISON WITH EALSTM

Table 11 reports the forecasting results of EALSTM Kratzert et al. (2019) and FlowNet on CAMELS dataset. The results show the superior performance of FlowNet compared to EALSTM over most of cases 7/9. At the horizon 1, FlowNet achieves better results over all the metrics of NSE, RMSE and MAE. At horizon 3 and horizon 5, FlowNet attains best performance of NSE and RMSE.

Table 11: Mean results of FlowNet compared to EALSTM on CAMELS with 3 different prediction horizon settings $H \in \{1, 3, 5\}$. Best results are highlighted in bold.

| Method | Horizon 1 | | | Horizon 3 | | | Horizon 5 | | |
|---|---|---|---|---|---|---|---|---|---|
| Metric | NSE ↑ | RMSE ↓ | MAE ↓ | NSE ↑ | RMSE ↓ | MAE ↓ | NSE ↑ | RMSE ↓ | MAE ↓ |
| EALSTM | 0.564 | 256.6 | 64.18 | 0.4199 | 316.6 | **93.2** | 0.3531 | **342.8** | **109.14** |
| FlowNet | **0.5784** | **250.7** | **64.01** | **0.4228** | **316.2** | 93.97 | **0.354** | **342.8** | 109.9 |

Table 12: Mean results of FlowNet compared to TimesFM on CAMELS with 3 different prediction horizon settings $H \in \{1, 3, 5\}$. Best results are highlighted in bold.

| Method | Horizon 1 | | | Horizon 3 | | | Horizon 5 | | |
|---|---|---|---|---|---|---|---|---|---|
| Metric | NSE ↑ | RMSE ↓ | MAE ↓ | NSE ↑ | RMSE ↓ | MAE ↓ | NSE ↑ | RMSE ↓ | MAE ↓ |
| TimesFM | 0.5577 | 255.7 | **61.77** | 0.3746 | 327.2 | **91.99** | 0.2952 | 358.5 | **109.5** |
| FlowNet | **0.5784** | **250.7** | 64.01 | **0.4228** | **316.2** | 93.97 | **0.354** | **342.8** | 109.9 |

## C.6 RESULTS COMPARISON WITH TIMESFM

Tables 12 and 13 report the forecasting performance of TimesFM Das et al. (2024) and FlowNet on CAMELS and LamaH datasets. These tables show that FlowNet achieves better performance across the majority of metrics on NSE and RMSE.

## C.7 ABLATION RESULTS OF LOCAL GLOBAL

Table 14 reports the comparison results of local, global phases and FlowNet on Mekong dataset. FlowNet using both local and global phases shows the better performance compared to using local or global phases only.

## C.8 ABLATION RESULTS OF BACKBONE MODELS

Table 15 reports the ablation results of 3 different backbone models LSTM, GRU and DMCT with and without FlowNet on Mekong dataset. The results show that using FlowNet consistently improves the performance compared with the original models, which demonstrate the generalization of FlowNet.

## C.9 ABLATION RESULTS OF GRAPHS

Table 16 reports the ablation results of DMCT, with Original graph and with PearCorr VLR graph. The results show that the model learns the incorrect relationships from the original graph and the performance decreased, however, when apply the PearCorr VLR graph, the performance boosted, which demonstrates that the model can get benefit from a good graph.

## C.10 ABLATION RESULTS OF CAMELS KNN SETTINGS

We conduct experiments for the ablation of KNN settings on CAMELS dataset. Table 17 shows that the best performance is using KNN=2 case.

Table 13: Mean results of FlowNet compare to TimesFM on LamaH with 3 different prediction horizon settings $H \in \{1, 3, 5\}$. Best results are highlighted in bold.

| Method | Horizon 1 | | | Horizon 3 | | | Horizon 5 | | |
|---|---|---|---|---|---|---|---|---|---|
| Metric | NSE ↑ | RMSE ↓ | MAE ↓ | NSE ↑ | RMSE ↓ | MAE ↓ | NSE ↑ | RMSE ↓ | MAE ↓ |
| TimesFM | 0.63 | 8.304 | **4.002** | 0.4376 | 11.97 | **6.266** | 0.3517 | 13.5 | **7.491** |
| FlowNet | **0.6598** | **7.792** | 4.385 | **0.4928** | **11.25** | 6.514 | **0.4067** | **12.7** | 7.536 |

Table 14: Mean ablation results of local and global phases compare to FlowNet on 3 datasets with prediction horizon setting $H = 1$. Best results are highlighted in bold.

| Dataset | CAMELS | | | LamaH | | | Mekong | | |
|---|---|---|---|---|---|---|---|---|---|
| Metric | NSE ↑ | RMSE ↓ | MAE ↓ | NSE ↑ | RMSE ↓ | MAE ↓ | NSE ↑ | RMSE ↓ | MAE ↓ |
| Local | 0.5631 | 257.8 | 68.6 | 0.6503 | 8.002 | 4.389 | 0.9309 | 0.1853 | 0.1198 |
| Global | 0.517 | 285.8 | 86.68 | 0.5963 | 8.701 | 5.259 | 0.9296 | 0.1875 | 0.1214 |
| FlowNet | **0.5784** | **250.7** | **64.01** | **0.6598** | **7.792** | **4.385** | **0.9323** | **0.1796** | **0.1144** |

### C.11 RUNTIMES OF FLOWNET

Runtimes of FlowNet depend on the number of edges in the flow graph due to its inflow and outflow models. The flow graphs in an area normally follow forest styles (i.e. a collection of tree-based components) due to the flow natures of rivers (circle connections may still happen but not very common) as we can see from Figure 4 for LamaH and Figure 5 for MRB data. Hence, the numbers of edges are typically smaller than the numbers of stations. For example, in LamaH, we have 452 stations but there are only 372 edges. For MRB data, we have 26 stations and 24 edges. Even for CAMEL-US, due to the absence of flow graphs, we use KNN graphs as a replacement, the number of edges is bounded by $K \cdot 671$ stations. Hence, in real-world settings, the number of edges is $O(N)$, where $N$ is the number of stations. Hence the number of models is also $O(N)$ (actually around $3 \cdot N$).

As shown in Table 4, FlowNet (with DMCT basebone) is among the most expensive methods together with SOFT, iTransformer, PatchTST and MICN. However, its slower training time results in significantly better prediction accuracy than other methods as demonstrated in Table 2 and Figure 2 in the revised paper. And in our paper, we aim at enhancing prediction accuracy rather than training speeds. Moreover, in terms of inference times, FlowNet with DMCT is only slightly slower than ordinary approaches as we demonstrated in the below Table 18. FlowNet takes 7.208s for inference while DMCT and MICN take 5.358s and 4.102s, respectively. FlowNet uses an acceptable extra inference time to get a significant performance boost. When using the much faster basebone like RLinear, the runtime of FlowNet will be much smaller.

To reduce runtimes of FlowNet, rather than predicting a large area, we can divide it into smaller catchment areas and perform FlowNet on these sub-areas instead. For example, CAMEL-US covers the whole USA. In such an enormous area, the geographics, topography, and climate conditions will be significantly different for different locations, leading to significantly different flow characteristics in different stations at different subareas. For example, stations in mountain areas tend to have faster and unstable flows than in plain areas. Trying to jointly predict these stations can be ineffective. Hence, by focusing on smaller areas, we can reduce the number of graph links and models, thus significantly enhancing the overall performance.

Moreover, in terms of algorithmics, FlowNet is a highly parallelable method. For example, during the local phase, per-station, inflow and outflow models can all be trained in parallel due to their independences. Similarly, the interaction during the global phase can also be parallelized quite straightforwardly.

Table 15: Mean ablation results of 3 different backbone models LSTM, GRU and DMCT with and without FlowNet on Mekong dataset with prediction horizon setting $H = 1$. Best results are highlighted in bold.

| Dataset | LSTM | | | GRU | | | DMCT | | |
|---|---|---|---|---|---|---|---|---|---|
| Metric | NSE $\uparrow$ | RMSE $\downarrow$ | MAE $\downarrow$ | NSE $\uparrow$ | RMSE $\downarrow$ | MAE $\downarrow$ | NSE $\uparrow$ | RMSE $\downarrow$ | MAE $\downarrow$ |
| w/o FlowNet | 0.8708 | 0.2612 | 0.1775 | 0.8862 | 0.2416 | 0.1631 | 0.9309 | 0.1853 | 0.1198 |
| w/ FlowNet | **0.899** | **0.2017** | **0.1344** | **0.894** | **0.1969** | **0.1285** | **0.9323** | **0.1796** | **0.1144** |

Table 16: Mean ablation results of DMCT, with Original graph and with PearCorr VLR on Mekong dataset with prediction horizon $H \in \{1, 3, 5\}$. Best results are highlighted in bold.

| Horizons | Horizon 1 | | | Horizon 3 | | | Horizon 5 | | |
|---|---|---|---|---|---|---|---|---|---|
| Metric | NSE $\uparrow$ | RMSE $\downarrow$ | MAE $\downarrow$ | NSE $\uparrow$ | RMSE $\downarrow$ | MAE $\downarrow$ | NSE $\uparrow$ | RMSE $\downarrow$ | MAE $\downarrow$ |
| DMCT | 0.9309 | 0.1853 | 0.1198 | 0.8889 | 0.2985 | 0.1868 | 0.853 | 0.3802 | 0.2391 |
| w/ Original | 0.8583 | 0.1918 | 0.2772 | 0.8012 | 0.3966 | 0.2673 | 0.7607 | 0.4762 | 0.3174 |
| w/ PearCorr VLR | **0.9323** | **0.1796** | **0.1144** | **0.8908** | **0.2945** | **0.1835** | **0.8555** | **0.3757** | **0.2354** |

## C.12 STATISTICAL SIGNIFICANCE COMPARISON

Figure 6 shows the statistical significance comparison of methods across stations where FlowNet is significant better than all other methods (with $p - value$ threshold $\alpha = 0.05$). Taking the comparison of FlowNet and PatchTST as an example, the $p - value$ is $1.086e^{-25} \ll 0.05$, and looking deeper to each station, FlowNet has better NSE in $468/671$ cases($69.7\%$) compared to PatchTST with max NSE differences of $0.5556$, while PatchTST only has better NSE in $203/671$ cases ($30.2\%$). And taking the comparison of FlowNet and SOFTS as an another example, the $p - value$ is $3.132e^{-51} \ll 0.05$, and looking deeper to each station, FlowNet has better NSE in $533/671$ cases($79.4\%$) compared to SOFTS with max NSE differences of $0.6618$, while SOFTS only has better NSE in $138/671$ cases ($20.5\%$).

## D DETAILS OF THE HYPERPARAMETER SENSITIVITY

We have conducted an extensive experiment for the hyperparameter sensitivity of FlowNet and DMCT, the results are shown in Figure 7. The hyperparameters of FlowNet, including the loss regulation factor $\alpha$, the initial learning rate of the global phase, the global learning rate factor, and the number of global loops. The hyperparameters of DMCT including the lookback window length, the learning rate, the number of multiscale level, and the dimension of hidden layer.

**Loss Regulation Factor $\alpha$.** The parameter $\alpha$ is used to regulate the loss function in local-global scheme, to balance the loss between the target node and the link node models. As the results of Figure 7 (A) show, the performance of model on NSE increases with the increasing $\alpha$ until around 0.7 to 0.9. In addition, we performed another study on LamaH, which is much larger than MRB. The results are shown in Figure 8. For $H = 1$, $H = 3$ and $H = 5$, the best values for $\alpha$ are 0.9, 0.5 and 0.3, respectively. These show the effectiveness of the global interaction phase and the consistency loss.

**Global Phase Initial Learning Rate.** This hyperparameter is used to set the initial learning rate for the global phase. Due to the local-global scheme, after the local phase, we need to reset the initial learning rate to a small value to avoid the issue of learning. The result of this hyperparameter sensitivity is shown in Figure 7 (B). This shows that when the initial learning rate in global phase is too small, e.g., 1e-4, the performance is not good compared to a larger value of setting.

**Global Learning Rate Factor $\beta$.** This factor is used to regulate the learning rate in global phase with the number of global loops. As the results of Figure 7 (C) show, the model has stable performance with the different settings of this hyperparameter $\beta$.

**Number of Global Iterations.** In the local-global scheme, the global phase is iteratively repeated to converge the model. We evaluated the number of iterations in the global phase, which is shown in

Table 17: The results of different KNN settings on CAMELS of FlowNet with different horizon length $H \in \{1, 3, 5\}$. Best results are highlighted in bold.

| Horizons | Horizon 1 | | | Horizon 3 | | | Horizon 5 | | |
|---|---|---|---|---|---|---|---|---|---|
| Metric | NSE↑ | RMSE↓ | MAE↓ | NSE↑ | RMSE↓ | MAE↓ | NSE↑ | RMSE↓ | MAE↓ |
| KNN=2 | **0.5784** | **250.7** | **64.01** | **0.4228** | **316.2** | **93.97** | **0.354** | **342.8** | **109.9** |
| KNN=5 | 0.5442 | 267.3 | 75.76 | 0.4106 | 323.2 | 101.2 | 0.3427 | 349 | 117.1 |
| KNN=10 | 0.3784 | 323 | 102.4 | 0.3906 | 328.6 | 105.6 | 0.3333 | 352.2 | 121.5 |

Table 18: Inference time for different methods on CAMELS dataset with horizon length $H = 1$.

| Methods | Inference Time (s) |
|---|---|
| FlowNet (DMCT) | 7.208 |
| FlowNet (RLinear) | 1.717 |
| DMCT | 5.358 |
| DLinear | 1.144 |
| RLinear | 1.234 |
| CycleNet | 3.018 |
| TQNet | 3.611 |
| FilterNet | 1.649 |
| iTransformer | 2.999 |
| SOFTS | 2.577 |
| CATS | 3.147 |
| TSMixer | 1.891 |
| LSTM | 1.134 |
| GRU | 1.001 |
| PatchTST | 3.2 |
| MICN | 4.102 |
| GCN | 0.325 |
| GCNII | 0.35 |
| ResGCN | 0.37 |
| ResGAT | 0.394 |
| AGCLSTM | 0.693 |

Figure 7 (D). The results show that with the global phased repeated, the model will converge and have stable performance.

**Look-back Window Length** $L$**.** In Figure 7 (E), we evaluate the sensitivity of the look-back window length for DMCT. We set 4 different lengths of the look-back window $L \in \{8, 16, 32, 64\}$ and evaluate the performance. The results show that the model performs the best when the look-back window length $L$ is around 32.

**Learning Rate.** We evaluate 4 different initial learning rate for DMCT and the results are shown in Figure 7 (F), which demonstrates that the best initial learning rate for DMCT is around 0.01.

**Number of Multiscale Levels.** We study the hyperparameter sensitivity of the number of multiscale level for DMCT. As the results in Figure 7 (G) show, increasing the multiscale levels, the model performs better. This shows that the model can extract more accurate temporal features from multiscale information with higher multiscale levels.

**Dimension of Hidden Layers.** As the results in Figure 7 (H) show, we have evaluated the sensitivity of the dimension of the hidden layers. These results show that the performance of the model is stable with different settings of the dimensions of the hidden layers.

# E  GRAPHS

We visualize the graphs in Figures 5, 9, and 10, including the original graph, the graph with Pearson Correlation Analysis (PearCorr) and the graph with Validation-based Links Reconstruction (VLR).

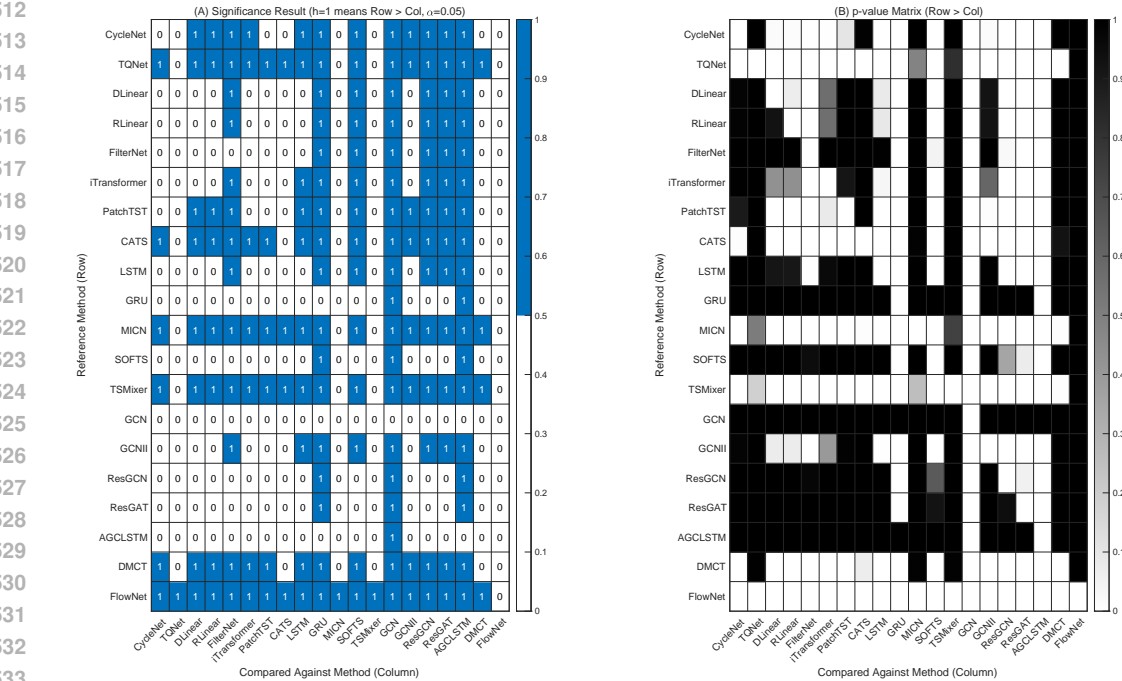

Figure 6: Statistical significance comparison of methods across stations (Paired Wilcoxon Signed-Rank Test) on CAMELS dataset with horizon length $H = 1$. (A) Significance result with $p - value$ threshold $\alpha = 0.05$ and $h = 1$ means the reference method (row) significant better than the compared against method (column), (B) $p - value$ matrix.

# F PSEUDO-CODE OF ALGORITHMS

## F.1 LOCAL PHASE

---

**Algorithm 1** Local Phase - Independent Learning

---

**Input**: Station $i \in V$, cross stations set $S := [1, \ldots, n]$, train set $\mathbf{X} := [X^i_{:,1:t}, X^{i,1}_{:,1:t}, \ldots, X^{i,n}_{:,1:t}]$, model list $M := [f_i, f_{i,1}, \ldots, f_{i,n}]$.
**Output**: model list $M$.

1: **for** each $model$ in $M$ **do**
2:      $X \leftarrow select(\mathbf{X})$             ▷ Select the corresponding set
3:      **for** each epoch **do**
4:          $x, y \leftarrow batch(X)$
5:          $\{\hat{y}^i, \hat{y}^{i,1}, \ldots, \hat{y}^{i,n}\} \leftarrow M(x)$
6:          $\hat{y}^i_{flow} = \sum_{j \in S} \hat{y}^{i,j}$             ▷ Inflow or outflow depends on the input
7:          **if** $model$ is $f_i$ **then**
8:              $loss \leftarrow \mathcal{L}^i(\hat{y}, y)$             ▷ Per-station loss
9:          **else if** otherwise **then**
10:            $loss \leftarrow \mathcal{L}^i_{Local}(\hat{y}^i_{flow}, y)$        ▷ Cross-station loss
11:          **end if**
12:          $loss.backward()$
13:          $update(model.params)$
14:      **end for**
15: **end for**
16: **return** $M$

---

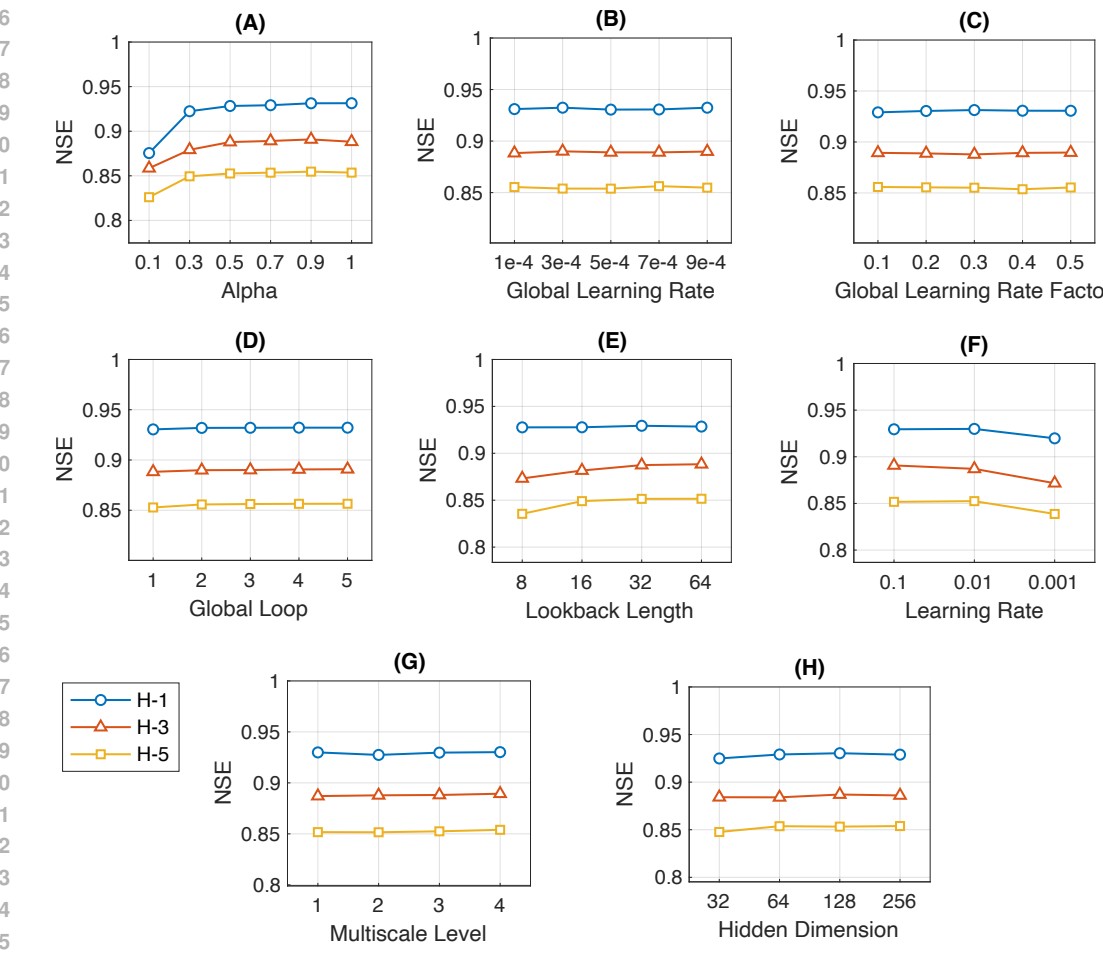

Figure 7: The hyperparameter sensitivity of FlowNet and DMCT.

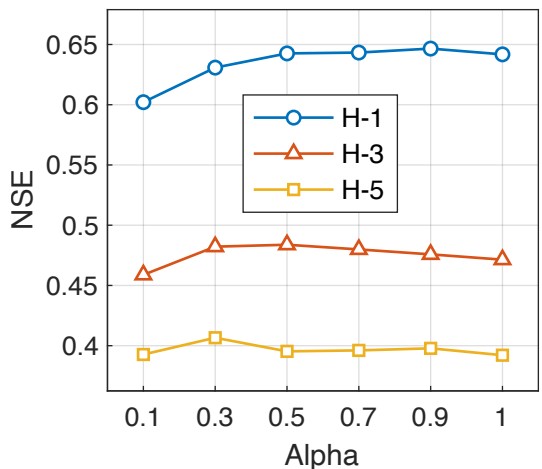

Figure 8: The Mean NSE results of the ablation study of alpha on LamaH dataset with alpha $\alpha \in \{0.1, 0.3, 0.5, 0.7, 0.9, 1.0\}$, $Iter = 3$ and forecasting length $H \in \{1, 3, 5\}$.

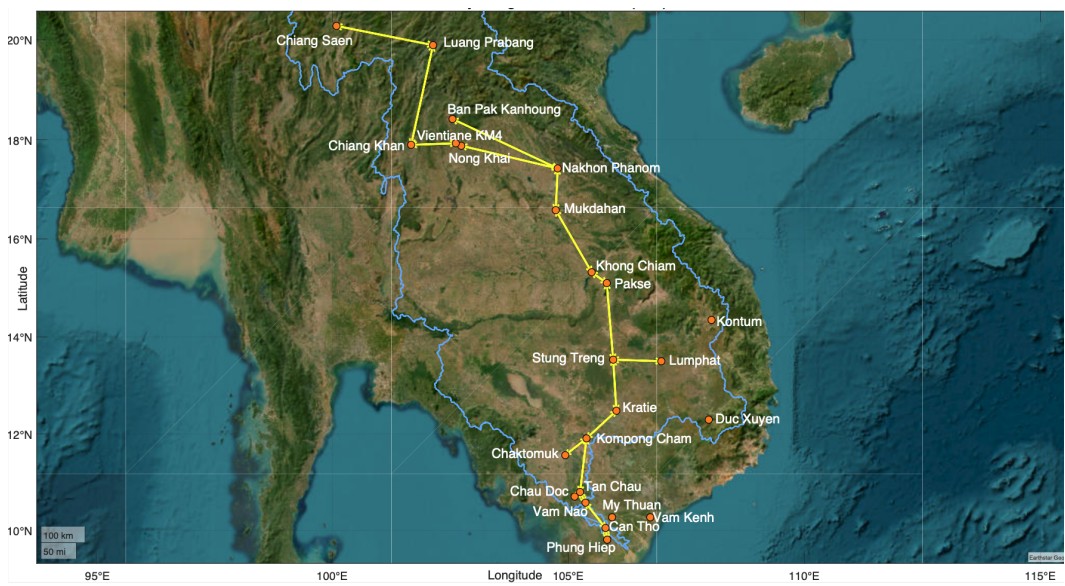

Figure 9: The graph of Pearson Correlation Analysis(PearCorr). We build the links when the links in the original graph have a significant Pearson correlation coefficient.

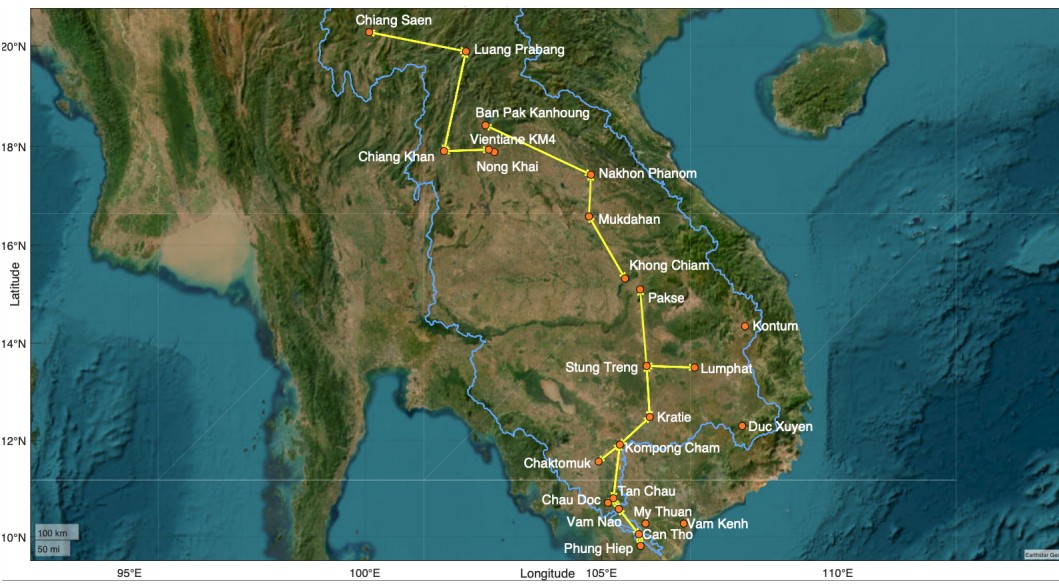

Figure 10: The graph of Validation-based Links Reconstruction (VLR). We reconstruct the links in the original graph that satisfy the PearCorr and VLR requirement.

## F.2 GLOBAL PHASE

## F.3 LOCAL GLOBAL TRAINING SCHEME

We combine the Local Phase 1 and Global Phase 2 and design a multi-phase training strategy Local Global Scheme, which is in Algorithm 3.

## F.4 VALIDATION-BASED LINKS RECONSTRUCTION

We provide the pseudo-code of Validation-based Links Reconstruction (VLR) in Algorithm 4.

---

**Algorithm 2** Global Phase - Interactive Learning

---

**Input**: Station $i \in V$, cross stations set $S := [1, \ldots, n]$, train set $\mathbf{X} := [X^i_{1:t,:}, X^{i,1}_{1:t,:}, \ldots, X^{i,n}_{1:t,:}]$, model list $M := [f_i, f_{i,1}, \ldots, f_{i,n}]$.
**Output**: model list $M$.

1: **for** each $model$ in $M$ **do**
2:      $X \leftarrow select(\mathbf{X})$            ▷ Select the corresponding set
3:      **for** each epoch **do**
4:          $x, y \leftarrow batch(X)$
5:          $\{\hat{y}^i, \hat{y}^{i,1}, \ldots, \hat{y}^{i,n}\} \leftarrow M(x)$
6:          $\hat{y}^i_{Global} = \sum_{j \in S} \hat{y}^{i,j}$
7:          $loss \leftarrow \alpha \cdot Loss(\hat{y}^i, y) + (1 - \alpha) \cdot Loss(\hat{y}^i, \hat{y}^i_{Global})$ ▷ Global loss from Eq.equation 2
8:          $loss.backward()$
9:          $update(model.params)$
10:     **end for**
11: **end for**
12: **return** $M$

---

**Algorithm 3** Local Global Training Scheme

---

**Input**: Global iterations $N$, model list $M := [f_i, f_{i,1}, \ldots, f_{i,n}]$, and all the other necessary inputs in Algorithms 1 and 2.
**Output**: model list $M$.

1: $M \leftarrow$ Algorithm 1                         ▷ Local Phase
2: **for** each iteration **do**
3:      $M \leftarrow$ Algorithm 2                 ▷ Global Phase
4: **end for**
5: **return** $M$

---

# G   LIMITATIONS

While our method FlowNet demonstrates promising results in streamflow forecasting, several limitations remains. First, our framework strongly relies on a relationship graph among hydrology stations (Section 2.2). In river networks exhibiting weak hydrological connectivity or fragmented monitoring systems, our method may revert to independent station-wise prediction. This could diminish the performance advantages observed. Future work could integrate physical hydrological models to enhance robustness under sparse correlation conditions. Second, we propose to evaluate our approach on a large-scale benchmark dataset: diverse climatic zones (tropical, temperate, polar), multi-scale gauge configurations (high-density vs. sparse networks), and multi-temporal resolutions (hourly to monthly scales). Notwithstanding these limitations, our experiments demonstrate FlowNet's superiority over conventional GNN/RNN/Transformer baselines across three quantitative metrics (cf. Appendix C). Its flexibility on data and learning methods also permits future integration with advanced methods and can work with irregular datasets that are common in practice.

# H   BROADER IMPACT

The proposed method FlowNet provides an effective system to forecasting streamflow. It will be very useful for local authorities to provide water resouce management and contingency plans for coping with climate change. In many vulnerable areas in developing countries, where the data collection system is not well-developed, the data is normally irregular with much missing data, different periods and different collected hydrology feature as in the case of the Mekong River Basin in our study. FlowNet, with its ability to deal with such kind of data effectively, will be extremely useful for these areas.

---

**Algorithm 4** Validation-based Links Reconstruction.

---

**Input**: Stations $i, j \in V$, directional link $A_{i,j} \in \mathbb{B}$, train set $X^i_{1:t,:}, X^{i,j}_{1:t,:}$ with all channels, validation set $X^i_{t:t+\tau,c}, X^{i,j}_{t:t+\tau,c}$ with water flow channel $c$, regulation factor $\gamma$, PearCorr correlation value $\lambda_{i,j}$ and the PearCorr threshold value $\phi$.

**Output**: Directional flow link $A_{i,j}$.

1: Initialize models $f_i, f_{i,j}$
2: **for** each epoch **do**                                     $\triangleright$ Train $f_i, f_{i,j}$ on $X^i_{1:t,:}$ and $X^{i,j}_{1:t,:}$
3:     $x^i, y^i \leftarrow batch(X^i); x^j, y^i \leftarrow batch(X^{i,j})$
4:     $loss_i \leftarrow \mathcal{L}(f_i(x^i), y^i); loss_{i,j} \leftarrow \mathcal{L}(f_{i,j}(x^j), y^i)$
5:     $loss_i.backward(); loss_{i,j}.backward()$
6:     $update(f_i.params); update(f_{i,j}.params)$
7: **end for**
8: $loss_i \leftarrow f_i(X^i_{t:t+\tau}, c); loss_{i,j} \leftarrow f_{i,j}(X^{i,j}_{t:t+\tau,c})$     $\triangleright$ Validate $f_i, f_{i,j}$ on $X^i_{t:t+\tau,c}$ and $X^{i,j}_{t:t+\tau,c}$
9: **if** $loss_{i,j} < \gamma loss_i$ and $\lambda_{i,j} > \phi$ **then**
10:     $A_{i,j} \leftarrow 1$
11: **else if** otherwise **then**
12:     $A_{i,j} \leftarrow 0$
13: **end if**
14: **return** $A_{i,j}$

---

# I   LLM USAGE

This manuscript was slightly edited using LLMs for language polishing and writing improvements. The authors retain full responsibility for the research content, including the concepts, analyses, and conclusions.