# OpenReview forum: "FlowNet: A Generic Independent and Interactive Model for Streamflow Forecasting"
_ICLR.cc/2026/Conference — Submitted to ICLR 2026_

### Official Review · Reviewer_n4Tx · 2025-10-29

**Soundness:** 1
**Presentation:** 2
**Contribution:** 2
**Rating:** 2
**Confidence:** 4

**Summary:**

FlowNet proposes a multi-scale framework for streamflow forecasting. The manuscript proposes to consider the spatial connection when forecasting streamflow. Rather than using a single model for all stations, first, local models are trained to each station separately. This stage is designed to handle varying available local training data. Second, using a predefined graph, the local predictions between the stations are adjusted globally and iteratively via training models. The original graph used in the global stage is refined by measuring correlations to correct the links between the stations. FlowNet is built using the proposed backbone: Disentangled Multiscale Cross-attention Transformer (DMCT), that is designed to capture the multi-scale seasonality-trend information. The manuscript evaluates the proposed framework on 3 benchmarks and compares it to other backbones.

**Strengths:**

- The framework is clearly described with mathematical formulation. hyperparameters details are described.
- Evaluation on 3 datasets is comprehensive.
- It is promising to consider spatial connections between points for streamflow forecasting.

**Weaknesses:**

- The method is highly dependent on the graph definition and it does not scale well with graph n x n (L136). In other words, the model can't work on large-scale data and it is limited to small scale dataset e.g., few hundreds of stations.
- What actually effect the streamflow is also meteorological data in the catchment area and near the station. This is ignored by the framework. Furthermore, what is the point of connecting channel like temperature in the streamflow direction? In reality this should be bidirectional.
- Wrong assumption about reality because streamflow cant be obtained in real time and it is used by the framework, since it relies on it heavily.
- Per-station and cross-station models are very expensive to build (L201-202). This is counter intuitive to what we do in ML. For example, "L210-211: we train two sets of cross-station models for each station Si, including the inflow and outflow models", so each station will have 3 models to be trained as I understood. What we do is usually we leverage a single generalized ML model that is trained with a lot of data and let the model learn the correlations by itself rather than building different local models. You might want to look at previous works [[1](https://hess.copernicus.org/articles/28/4187/2024/hess-28-4187-2024.html)].
- It is not clear how the baselines are trained. i.e., the description of the baselines and how they are trained and finetuned is missing.
- State-of-the-art baselines are missing e.g., [[2](https://hess.copernicus.org/articles/23/5089/2019/hess-23-5089-2019.html), [3](https://agupubs.onlinelibrary.wiley.com/doi/full/10.1029/2023WR036170), [4](https://arxiv.org/abs/2505.22535), [5](https://essopenarchive.org/users/810569/articles/1227435-high-resolution-national-scale-water-modeling-is-enhanced-by-multiscale-differentiable-physics-informed-machine-learning), [6](https://agupubs.onlinelibrary.wiley.com/doi/10.1029/2023WR035337)]. At least [[2](https://hess.copernicus.org/articles/23/5089/2019/hess-23-5089-2019.html)] should be included. The baselines in the manuscript should be works for streamflow forecasting following proposed methodologies in previous works rather than different ML building blocks.
- Table 6: looking at the standard deviation and numbers, the improvement compared to some baselines with random seed is negligible.
- Ablation study, Fig. 3: the improvement is negligible
- Effects of different components of DMCT. The paper has some unjustified claims (L428-431). Some proposed component are redundant e.g., multi-scale (M) in Table 2. In my view, there is also no reason why the model should not work without a specific type of normalization. Most models for streamflow do not necessary use InstanceNorm.

**Questions:**

- L45-49: this is not true, static features are available globally and are not hard to obtain.
- Line140: where does the streamflow value come from? We can get them from the dataset but in reality, these values are not available in real time.
- How does per-station forecast work? Is it in parallel? It is highly inefficient.
- L204-205: do you finetune the parameters for each station? If yes this is high inefficient, if no this will lead to sup-optimal results. This is why we use one model usually.
- L208: loss function like MSE is not optimal for streamflow i.e., the loss function should consider extremes to account for river flood forecasting.
- I struggle to understand the relation between Eq. 1 and 2? $y_{inflow}$ and $y_{outflow}$ can be zero and the loss will be perfectly fine?
- Missing literature and baselines (see weaknesses).

**Minor**:
- Line740-741: I thought the model should work with an inconsistent spatial representation.
- L166: better to use $\hat{y}$ as ground truth, it is more common to void confusion.
- Eq 7 is incorrect > new variables need to be renamed.
- L345: I see more than 8 baselines. I think you mean 18.

---

> ### Author Response · Authors · 2025-11-21
> **Response to the Reviewer n4Tx (Part 1)**
>
> Many thanks for your insightful comments. Please find our responses below. All changes were updated into our paper (highlighted in blue).
>
> **W1. The method is highly dependent on the graph definition and it does not scale well with graph n x n (L136). In other words, the model can't work on large-scale data and it is limited to small scale dataset e.g., few hundreds of stations.**
>
> One major drawback of FlowNet is its high runtimes due to its additional inflow and outflow models in the local phase and its iterative interaction scheme in the global phase. However, as we also pointed out in our responses to Reviewer wMmd (Q3), the flow graphs in an area normally follow forest styles (i.e. a collection of tree-based components) due to the flow natures of rivers (circle connections may still happen but not very common) as we can see from Figure 4 for LamaH and Figure 5 for MRB data. Hence, the numbers of edges are typically smaller than the numbers of stations. For example, in LamaH, we have 452 stations but there are only 372 edges. For MRB data, we have 26 stations and 24 edges. Even for CAMEL-US, due to the absence of flow graphs, we use KNN graphs as a replacement, the number of edges is bounded by $K \cdot$ 671 stations. Hence, in real-world settings, the number of edges is $O(N)$, where $N$ is the number of stations.  Hence the number of models is also $O(N)$ (actually around $3\cdot N$) and not $O(N^2)$ luckily.
>
> We also conducted experiments using CAMELS and LamaH, two of the largest streamflow datasets in the literature. As we discussed in our response to Reviewer LeBJ (W4), it takes FlowNet (DMCT) and FlowNet (RLinear) 3550.46 and 1505.65 seconds for overall training on 4 Nvidia L4 GPU, respectively. The overall inference time for FlowNet with DMCT and RLinear barebones are 7.2 and 1.7 seconds on single L4 GPUs, respectively. These times are affordable, especially when having much better prediction accuracy. We also discuss various ways to improve the runtimes of FlowNet in our responses to reviewers wMmd (Q3) and EzYW (Q2) and in Section C.11 in Appendix of the revised paper including splitting data into smaller areas and training/inference in parallel due to highly parallelable nature of FlowNet.
>
> **W2. What actually effect the streamflow is also meteorological data in the catchment area and near the station. This is ignored by the framework. Furthermore, what is the point of connecting channel like temperature in the streamflow direction? In reality this should be bidirectional.**
>
> For all 3 studied datasets LamaH, CAMELS and MRB, we used available meteorological data provided with them together with streamflow data as we described in Section A in the Appendix. For example, for the LamaH dataset, we used 4 meteorological data including precipitation, topsoil moisture, air temperature, and surface pressure. For the CAMELS dataset, we used features such as  precipitation, temperature, and potential evapotranspiration. For the MRB dataset, we used water discharge and rainfall.
>
> As we demonstrated in Definition 1 in Section 2 of the paper, the downstream flow graph is used to link two stations $S_i$ and $S_j$ if water flows from $S_i$ to $S_j$ directly. We did not connect channel like temperature in the streamflow direction as the reviewer thought. We are very sorry if our writing led to these confusions. We carefully proofread the paper to remove any possible misunderstanding problem.
>
> **W3. Wrong assumption about reality because streamflow cant be obtained in real time and it is used by the framework, since it relies on it heavily.**
>
> Streamflow can be calculated in many different ways from traditional methods like velocity-area method, real-time sensors or other technologies like Radar [A, B, C]. For example, the National Water Information System (NWIS) Web provides streamflow data online in near real time in the USA.
>
> In our paper, the streamflow data is provided in the CAMELS, LamaH and MRB datasets. These datasets are commonly used in streamflow predictions in the literature. The streamflow values are also provided in these datasets.
>
> We are very sorry that we did not understand the reviewer’s comment about “wrong assumption” here very well? We are also sorry if we did put the term real-time in the paper (that may cause misunderstanding). We carefully checked the paper and removed any potential misleading terms and concepts.
>
> References:
>
> [A] Measuring real-time streamflow using emerging technologies: Radar, hydroacoustics, and the probability concept, Journal of Hydrology, 2018.
>
> [B] Discharge Measurements at Gaging Stations, USGS Techniques and Methods 3-A8.
>
> [C] USGS Water Data for USA website.
>
> **(To be continued…)**

---

> > ### Author Response · Authors · 2025-11-21
> > **Response to the Reviewer n4Tx (Part 2)**
> >
> > **W4. Per-station and cross-station models are very expensive to build (L201-202). This is counter intuitive to what we do in ML. For example, "L210-211: we train two sets of cross-station models for each station Si, including the inflow and outflow models", so each station will have 3 models to be trained as I understood. What we do is usually we leverage a single generalized ML model that is trained with a lot of data and let the model learn the correlations by itself rather than building different local models. You might want to look at previous works [1].**
> >
> > Thank you very much for pointing out for us the previous work [1]. We fully agreed with the reviewer that FlowNet incurs more computational overhead due to its per-station and cross-station models and its interaction scheme as also confirmed in W1.
> >
> > We also discussed in the paper that existing works tend to aim at building a single model for each station independently or using a single large model for all stations such as the appointed reference [1] or (Kratzert et al., 2018; Kirschtein et al. 2024) in our paper. While using a single large model is a very elegant approach, it still suffers from some key drawbacks as follows:
> >
> > 1. It entirely relies on blackbox models such as LSTM [1] to discover relationships among stations and input features. With huge amounts of training data, it may be possible. But in many areas like MRB, the available data is scarce. Also, there is no guarantee that the model can actually discover the relationship and make the prediction effectively since everything is enclosed inside a fully black box. FlowNet explicitly employs flow physics via the flow graph and in/out water flow relationships at each station to guide the learning process more effectively. Thus, it acquires much better performance as we demonstrated in the paper.
> >
> > 2. This approach requires restrictive assumptions such as uniform input features and/or similar training data periods. In many vulnerable areas in developing countries, where the data collection system is not well-developed, the data is normally irregular with much missing data, different periods and different collected hydrology features as in the case of the Mekong River Basin in our study. FlowNet, with its independent model constructions and interaction scheme, can effectively exploit all possible available data for improving performance. Hence, it will be extremely useful for these areas.
> >
> > 3. By going into a different direction to all existing works, FlowNet has proven that its unique iterative interaction method can help to significantly improve the prediction accuracy. However, this come with a runtime trade-off as mentioned above.
> >
> > References:
> >
> > [1] Kratzert et al., HESS Opinions: Never train a Long Short-Term Memory (LSTM) network on a single basin, Hydrology and Earth System Sciences, 2024.
> >
> > **W5. It is not clear how the baselines are trained. i.e., the description of the baselines and how they are trained and finetuned is missing.**
> >
> > Thank you very much for pointing out this. We added into Section B.1 in the Appendix the hyper-parameter search spaces for all employed baselines. We also added into Section B.2 the detailed descriptions of all 18 studied baselines. We copied some example paragraphs below. For the full details, please refer to the revised version of our paper.
> >
> > *We conduct a systematic hyperparameter search over: hidden dimension in $\{32, 64, 128, 256\}$, number of layers in $\{1, 2, 3\}$, learning rate in $\{10^{-1}, 10^{-2}, 10^{-3}, 10^{-4}\}$, loss regularization factor $\alpha \in [0.1, 0.95]$, number of global iterations $\mathrm{Iter} \in [1, 5]$, and global learning rate factor $\beta \in \{0.01, 0.1, 0.2\}$. The best configuration is selected via validation performance. We use the similar search strategies for baselines, number of heads in $\{2, 4, 8\}$, patch and stride length in $\{16, 48, 96, 160, 320\}$, forward function dimension $\{128, 256, 512, 1024\}$, and cycle length in $\{12, 36, 60, 120\}$.*
> >
> > *Transformer-based Models (PatchTST, iTransformer, CATS). These models represent the most influential architectural developments for long-sequence forecasting. PatchTST introduced patching and channel-independent attention, enabling efficient modeling of long temporal contexts. iTransformer inverted the attention dimension to emphasize cross-variable dependencies, advancing multivariate modeling. CATS further demonstrated that cross-attention only architecture for Transformer enhances inter-series interactions. Collectively, these models define the current SOTA direction for scalable Transformer-based forecasting.*
> >
> > **(To be continued…)**

---

> > > ### Author Response · Authors · 2025-11-21
> > > **Response to the Reviewer n4Tx (Part 3)**
> > >
> > > **W6. State-of-the-art baselines are missing e.g., [2, 3, 4, 5, 6]. At least [2] should be included.**
> > >
> > > *Table 1. Compare with EALSTM [2] on CAMELS.*
> > >
> > > | CAMELS  | H=1        |           |           | H=3        |           |       | H=5       |           |        |
> > > | ------- | ---------- | --------- | --------- | ---------- | --------- | ----- | --------- | --------- | ------ |
> > > | Metrics | NSE        | RMSE      | MAE       | NSE        | RMSE      | MAE   | NSE       | RMSE      | MAE    |
> > > | EALSTM  | 0.564      | 256.6     | 64.1803   | 0.4199     | 316.6     | **93.2**  | 0.3531    | **342.8**     | **109.14** |
> > > | **FlowNet** | **0.5784** | **250.7** | **64.01** | **0.4228** | **316.2** | 93.97 | **0.354** | **342.8** | 109.9  |
> > >
> > > Thank you for pointing out these interesting works. Table 1 above reports the forecasting results of EALSTM [2] and FlowNet on CAMELS dataset. The results show the superior performance of FlowNet compared to EALSTM over 7/9 cases. At the horizon 1, FlowNet achieves better results over all the metrics of NSE, RMSE and MAE. At horizon 3 and horizon 5, FlowNet attains best performance of NSE and RMSE.
> > >
> > > We also included all missing references into the Related Work section of the paper as follows:
> > >
> > > *River flow forecasting plays a critical role in flood management, water resource optimization, infrastructure protection, and climate resilience and has attracted many research efforts such as (Feng et al., 2021; Kratzert et al., 2019; Zhou et al., 2025; Bindas et al., 2024; Song et al., 2025; Eddin et al., 2025; Wang et al., 2024a; Kratzert et al., 2024).*
> > >
> > > References:
> > >
> > > [2] Kratzert et al., Towards learning universal, regional, and local hydrological behaviors via machine learning applied to large-sample datasets, 2019.
> > >
> > > **W7. Table 6: looking at the standard deviation and numbers, the improvement compared to some baselines with random seed is negligible.**
> > >
> > > Thank you. In Table 6 (now Table 7 in the revised paper), we study the stability of different methods under different random seeds for the MRB data with H=1. Please note that the results are averaged for all 26 stations. This can cause a feeling that the difference is negligible. However, taking a deeper look at FlowNet and DMCT as an example, the averaged NSE difference between them is 0.0014. However, FlowNet has better NSEs than DMCT on 21 over 26 stations with max NSE differences of 0.022 for Can Tho station. Hence, the performance differences are much larger than appears through the averaged values.
> > >
> > > Table 2 in the paper also reported the averaged performances of all studied methods for all three datasets MRB, LamaH and CAMELS on all prediction horizons $H=(1,3,5)$. FlowNet significantly outperforms all others in all the cases, followed by DMCT.
> > >
> > > For further assessing performances of FlowNet and other methods, we additionally conducted the Wilcoxon signed rank tests among all methods on our largest dataset CAMELS as an example. The results are shown in Figure 6 in Section C.12 in Appendix with H=1 and significant threshold of 0.05 for NSE values. FlowNet is statistically better than all other competitors. Taking the comparison of FlowNet and PatchTST as an example, the $p-value$ is $1.086e^{-25} \ll 0.05$, and looking deeper to each station, FlowNet has better NSE in $468/671$ cases(69.7%) compared to PatchTST with max NSE differences of $0.5556$, while PatchTST only has better NSE in $203/671$ cases (30.2%). And taking the comparison of FlowNet and SOFTS as an another example, the $p-value$ is $3.132e^{-51} \ll 0.05$, and looking deeper to each station, FlowNet has better NSE in $533/671$ cases(79.4%) compared to SOFTS with max NSE differences of $0.6618$, while SOFTS only has better NSE in $138/671$ cases (20.5%).
> > >
> > > **(To be continued…)**

---

> > > > ### Author Response · Authors · 2025-11-21
> > > > **Response to the Reviewer n4Tx (Part 4)**
> > > >
> > > > **W8. Ablation study, Fig. 3: the improvement is negligible.**
> > > >
> > > > *Table 2. Mean ablation results of local and global phases compare to FlowNet on 3 datasets LamaH, CAMELS and MRB with prediction horizon setting H = 1.*
> > > >
> > > > | Datasets    | CAMELS     |           |           | LamaH      |           |           | Mekong     |            |            |
> > > > | ----------- | ---------- | --------- | --------- | ---------- | --------- | --------- | ---------- | ---------- | ---------- |
> > > > | Metrics     | NSE        | RMSE      | MAE       | NSE        | RMSE      | MAE       | NSE        | RMSE       | MAE        |
> > > > | local       | 0.5631     | 257.8     | 68.6      | 0.6503     | 8.002     | 4.389     | 0.9309     | 0.1853     | 0.1198     |
> > > > | global      | 0.517      | 285.8     | 86.68     | 0.5963     | 8.701     | 5.259     | 0.9296     | 0.1875     | 0.1214     |
> > > > | **FlowNet** | **0.5784** | **250.7** | **64.01** | **0.6598** | **7.792** | **4.385** | **0.9323** | **0.1796** | **0.1144** |
> > > >
> > > > Since the prediction accuracies of DMCT and FlowNet on the MRB data are already quite high (c.f. Table 2 on the revised paper), the improvement will look small when using DMCT as a base model for FlowNet as we have seen in Figure 3A as an example. Hence, we conducted further studies on both LamaH and CAMELS. The results are shown in Table 2 above (also in Table 14 of the revised paper). The global interaction scheme significantly improves the prediction accuracy on all datasets and evaluation metrics. Concretely, on the CAMELS dataset, 571 over 671 (85.1%) stations have their NSEs improved after the global interaction phase with the maximum improvement of 0.7893. For LamaH, 399 over 425 stations (93.8%) show improvement in NSEs with max improvement of NSE value of 0.6374203. These have proven the design rationale of FlowNet.
> > > >
> > > > **W9. Effects of different components of DMCT. The paper has some unjustified claims (L428-431). Some proposed component are redundant e.g., multi-scale (M) in Table 2. In my view, there is also no reason why the model should not work without a specific type of normalization. Most models for streamflow do not necessary use InstanceNorm.**
> > > >
> > > > *Table 3: Effects of different components of DMCT on the Mekong dataset, where w/o D, w/o M, w/o IN denote without Disentangled, Multiscale and Instance Normalization modules, respectively.*
> > > >
> > > > |     | DMCT  | w/o D | w/o M | w/o IN |
> > > > | --- | ----- | ----- | ----- | ------ |
> > > > | H=1 | **0.930** | 0.920 | 0.927 | -0.18  |
> > > > | H=3 | **0.888** | 0.877 | 0.887 | -1.81  |
> > > > | H=5 | **0.853** | 0.843 | 0.852 | -2.45  |
> > > >
> > > > Thank you for your comment. Each specific method may require a different data normalization technique. In our case, we did try some different ways and DMCT works well with InstanceNorm. In Table 3 of our revised paper (copied here as Table 3), we performed an ablation study on performances of different components of DMCT including Disentangled, Multiscale and InstanceNorm. The results show that each component contributes to the overall performance of DMCT, particularly the Instance Norm.
> > > >
> > > > **Q1. L45-49: this is not true, static features are available globally and are not hard to obtain.**
> > > >
> > > > We agree with the reviewer, some static features are not hard to obtain. However, many are hard to collect due to many factors such as restricted access or even unavailable (please refer to Section 2.1.1 in (Brunner et al., 2021) for a nice review).
> > > >
> > > > **Q2. Line140: where does the streamflow value come from? We can get them from the dataset but in reality, these values are not available in real time.**
> > > >
> > > > Streamflow can be calculated in many different ways from traditional methods like velocity-area method, real-time sensors or other technologies like Radar [A, B, C]. For example, the National Water Information System (NWIS) Web provides streamflow data online in near real time in the USA.
> > > >
> > > > In our paper, the streamflow data is provided in the CAMELS, LamaH and MRB datasets. These datasets are commonly used in streamflow predictions in the literature. We also do not deal with real-time prediction or real-time data in our work.
> > > >
> > > > References:
> > > >
> > > > [A] Measuring real-time streamflow using emerging technologies: Radar, hydroacoustics, and the probability concept, Journal of Hydrology, 2018.
> > > >
> > > > [B] Discharge Measurements at Gaging Stations, USGS Techniques and Methods 3-A8.
> > > >
> > > > [C] USGS Water Data for USA website.
> > > >
> > > > **(To be continued…)**

---

> > > > > ### Author Response · Authors · 2025-11-21
> > > > > **Response to the Reviewer n4Tx (Part 5)**
> > > > >
> > > > > **Q3. How does per-station forecast work? Is it in parallel? It is highly inefficient.**
> > > > >
> > > > > For each station $S_i$, the per-station forecasting model uses the local historical climate and streamflow data at $S_i$ to predict future streamflow values. Since the training processes of all per-station models are entirely independent. They can be trained in parallel straightforwardly to reduce overall training times.
> > > > > In W1 above, we showed that, on on CAMELS, one of the largest streamflow datasets with 671 stations in the whole USA, it takes FlowNet (with DMCT) and FlowNet (with RLinear) 3550.46 and 1505.65 seconds for overall training on 4 Nvidia L4 GPU, respectively. The overall inference time for FlowNet with DMCT and RLinear barebones are 7.2 and 1.7 seconds on a single L4 GPUs, respectively. Overall, the runtimes of FlowNet is larger than others as the reviewer also pointed out but it returns in much better prediction accuracy. Moreover, for that huge dataset, 1h GPU for training is not something unaffordable.
> > > > >
> > > > > **Q4. L204-205: do you finetune the parameters for each station? If yes this is high inefficient, if no this will lead to sup-optimal results. This is why we use one model usually.**
> > > > >
> > > > > That is a great question. We indeed used the same parameters for all barebone models of FlowNet for reducing runtimes. As the reviewer said, that will lead to sub-optimal results for FlowNet. However, even though FlowNet already outperforms all baselines. This further proves the effectiveness of the independent local-global interactive scheme of FlowNet. When doing independent hyper-parameter turning for each model, the performance is expected to be further improved. We also used the same barebone for all stations, while choosing different suitable barebones for different stations would also help to further enhance the prediction accuracy.
> > > > >
> > > > > **Q5. L208: loss function like MSE is not optimal for streamflow i.e., the loss function should consider extremes to account for river flood forecasting.**
> > > > >
> > > > > The MSE loss is a very commonly used loss function or streamflow forecasting. That is why we used it in the paper. However, we fully agree with the reviewer that it may not be optimal when detecting extremes like river flood. Currently, we are focusing on designing a new prediction model that can effectively capture extremes in streamflow. We believe that an effective approach should come from both model architecture and learning algorithm design rather than relying on specific loss metrics.
> > > > >
> > > > > **Q6. I struggle to understand the relation between Eq. 1 and 2?  $y_{inflow}$ and $y_{outflow}$  can be zero and the loss will be perfectly fine?**
> > > > >
> > > > > The key idea of the global interaction phase of FlowNet is that we keep the prediction outcomes of each station $S_i$ not only close to its ground truth but also not to deviate too much from other two perspectives of inflow and outflow physics. We discovered that we made a typo in Eq.1 with a redundant $\hat{y}^i$. We fully understand why the reviewer is confused. We are really sorry for it. We fixed the Eq. 1 as follows:
> > > > >
> > > > > $\hat{y}^i_{Global}=mean(\hat{y}^i_{inflow},\hat{y}^i_{outflow})$
> > > > >
> > > > > we also added a note that:
> > > > >
> > > > > *if both inflow and outflow models do not exist, $\mathcal{L}^i_{Global}$ will be equivalent to the per-station loss $\mathcal{L}^i$*
> > > > >
> > > > > **Q7. Missing literature and baselines (see weaknesses).**
> > > > >
> > > > > Many thanks again. We added missing literatures and baselines in **W6** above.
> > > > >
> > > > > **Q8. Minor.**
> > > > >
> > > > > *Line740-741: I thought the model should work with an inconsistent spatial representation.*
> > > > >
> > > > > Yes, FlowNet can work with inconsistent spatial representation. However, for LamaH, we followed the data selection process of (Kirschtein et al., 2024) for simplicity.
> > > > >
> > > > > *L166: better to use as ground truth, it is more common to void confusion.*
> > > > >
> > > > > We figured out a typo in the paper saying that $\hat{y}^i$ is the ground truth and it lead to misunderstanding. We fixed the typo. Now in the paper $y^i$ denote ground truth and $\hat{y}^i$ denote predicted results for station $S^i$ consistently.
> > > > >
> > > > > *Eq 7 is incorrect > new variables need to be renamed.*
> > > > >
> > > > > Thank you. We fixed the paper adding:
> > > > >
> > > > > *Concretely, we use the mean and standard deviation values from the target station $S_i$ to replace the ones of the input time series from the cross-station series $x^{i,j}$ and predicted output $\hat{y}^{i,j}$ as follows:*
> > > > >
> > > > > *L345: I see more than 8 baselines. I think you mean 18.*
> > > > >
> > > > > Many thanks again. We fixed the typo as follows:
> > > > >
> > > > > *Mean NSE results of 18 selected SOTA baselines compared to our methods DMCT and FlowNet on 3 datasets with 3 different prediction horizon settings $H\in\{1, 3, 5\}$. Best results are highlighted in bold and second best results are underlined.*
> > > > >
> > > > > **Summarize:** We kindly hope that we can address all of your concerns effectively? Many thanks again for your great help on our paper.

---

> > > > > > ### Comment · Reviewer_n4Tx · 2025-11-27
> > > > > >
> > > > > > **W.1.** Actually, CAMELS and LamaH are not really large-scale datasets. There are e.g., CARAVAN (>22K points) and GRDC (> 11K points).
> > > > > >
> > > > > > **W.2.** Thank you for clarifying. Yes, I though you also connect other channels. The model acts like a time-series analysis where the output is post-processed with a predefined graph. Connecting meteorological conditions should be bidirectional which is currently ignored in the framework. Modelling meteorological conditions in the neighborhood is still ignored too. But this is not a concern anymore.
> > > > > >
> > > > > > **W.3.** Sorry for the misunderstanding. Maybe, I had to clarify more. Streamflow is for sure available in the dataset! The point is that we don’t use real-time streamflow for forecast as this information is not available due to data acquisition and data assimilation. In addition, observational data are not available everywhere (ungauged locations). The mentioned references are about how to measure streamflow not about operational systems which use these values in real-time. You would refer to [[HESS]](https://hess.copernicus.org/articles/26/5493/2022/).
> > > > > >
> > > > > > **W4.1.** FlowNet is also a black box model. GNNs are not interpretable. In my view, adding the flow direction does not make the model interpretable, however, it simply adds more inductive bias to the model. **W4.2.** FlowNet also relies on these assumptions, so I think the same argument can be used against FlowNet, since we can’t build the graph or use the streamflow (argued in W3) in developing countries too. Just as a side note, Google's model is running globally in regions with less data as well. **W4.3.** FlowNet does not go into different direction from previous works. There are many works about distributed modelling in hydrology (see main review).
> > > > > >
> > > > > > **W6.** I am not sure how the baseline was trained. The result of EDLSTM is available in both [[Nature]](https://www.nature.com/articles/s41586-024-07145-1) and [[HESS]](https://hess.copernicus.org/articles/23/5089/2019/) where benchmark datasets are available. FlowNet can be compared to the reported number in the previous works on the benchmark.
> > > > > >
> > > > > > **W.9.** The table did not show a significant improvement with multi-scale. The improvement is +0.003, 0.001, and 0.001 for H=1, H=3, H=5, respectively.  There is still no reason why the model doesn’t work without instance normalization.
> > > > > >
> > > > > > I would like to thank the authors for their detailed response. While I appreciate these efforts, I believe that **the limited novelty, missing a proper comparison to baselines, and scalability of the framework are still major issues in this manuscript**. Given these, I keep my rating for the current state of the manuscript.

---

> ### Author Response · Authors · 2025-12-02
> **Response to the Reviewer n4Tx follow-up comments (Part 1)**
>
> We sincerely thank the reviewer  for further discussions and comments on our work. Please find our response below.
>
> **W.1.** *Actually, CAMELS and LamaH are not really large-scale datasets. There are e.g., CARAVAN (>22K points) and GRDC (> 11K points).*
>
> Many thanks for pointing out larger datasets than LamaH and CAMELS. CARAVAN with thousands of catchments around the whole world  is clearly an enormous dataset and collection effort. Moreover, the CARAVAN dataset is a collection of multiple datasets, including CAMELS and LamaH, which overlaps to some extent with the dataset we have chosen. Although data from 10,000+ stations are available through GRDC, both the quality of the available records and the period of record for individual basins varies significantly and are not coupled with catchment attributes or meteorological forcing data.
>
> The datasets used in this paper are not intended to maximize the number of data points, but rather to evaluate our model and other methods on the same benchmark. Although CAMELS and LamaH-CE have relatively smaller basin counts than CARAVAN, they are already large-scale ones and they both provide high-quality, and consistently standardized time series data, ensuring reliable inputs for model training. In particular, **CAMELS and LamaH are two golden standard benchmark datasets within the hydrological machine learning community.**
>
> Nevertheless, as we mentioned,  all stations in an enormous area such as  the whole world in CARAVAN, where the geographics, topography, and climate conditions  can be entirely different for different locations,  will have significantly different flow characteristics  depending on their locations. For example, stations in mountain areas tend to have faster and unstable flows than in plain areas. Trying to jointly predict all these stations in a single model can be ineffective as pointed out in works like [3]. Hence, people tend to focus on particular areas in water research like the Central EU of LamaH and the USA of CAMELS.
>
> Reference:
>
> [1] Caravan - A global community dataset for large-sample hydrology
>
> [2] https://www.bafg.de/GRDC
>
> [3] Khand, Kul, and Gabriel B. Senay. "Evaluation of streamflow predictions from LSTM models in water-and energy-limited regions in the United States." Machine Learning with Applications 16 (2024): 100551.
>
> **W.2.** *Thank you for clarifying. Yes, I though you also connect other channels. The model acts like a time-series analysis where the output is post-processed with a predefined graph. Connecting meteorological conditions should be bidirectional which is currently ignored in the framework. Modelling meteorological conditions in the neighborhood is still ignored too. But this is not a concern anymore.*
>
> We thank the reviewer for the agreement. Please note that FlowNet actually does not ignore meteorological conditions in the neighborhood. It indeed considers them but in a very unique way as we explain below.
>
> Many existing works also consider meteorological conditions in the neighbors by two ways: (i) push everything into a single large model and let it try to identify relationships itself like LSTM (Kratzert et al., 2018); and (ii) use GNNs to encode all inputs (including meteorological data) of neighbors to do prediction.
>
> However, FlowNet considers meteorological data from neighbors into inputs of cross-model stations. The outputs of cross-model stations will then be used to adjust the outputs of per-station models. This interaction is a very unique way in FlowNet. It helps the output of each station to be determined also from meteorological data of its neighbors but from the output of the models rather than from the input sides as all existing works.
>
> **(To be continued…)**

---

> ### Author Response · Authors · 2025-12-02
> **Response to the Reviewer n4Tx follow-up comments (Part 2)**
>
> **W.3.** *Sorry for the misunderstanding. Maybe, I had to clarify more. Streamflow is for sure available in the dataset! The point is that we don’t use real-time streamflow for forecast as this information is not available due to data acquisition and data assimilation. In addition, observational data are not available everywhere (ungauged locations). The mentioned references are about how to measure streamflow not about operational systems which use these values in real-time. You would refer to [HESS].*
>
> Thank you for your valuable comment. The paper you mentioned highlights techniques for integrating near-real-time data to improve forecast accuracy [4]. We fully acknowledge the practical constraints in real-time hydrological forecasting, particularly concerning the data acquisition and assimilation lag associated with streamflow observations.
> In the literature, we can see two different design ways for forecasting models, one is using both streamflow and climate data as inputs. Others only employ climate data as inputs. Both have advantages and disadvantages. The former ones can employ historical streamflow data to have richer feature sets, thus hopefully can improve the performance such as AR LSTM in [4]. The latter one is more flexible since after training, the model can be used to predict other stations without historical streamflow data (ungauged stations), assuming that these ungauged stations are closely similar to the trained ones, either directly or using adaptation techniques.
>
> In our paper, we chose the former way due to data availability in CAMELS and LamaH. However, straightforwardly, FlowNet can be adapted to use climate data only (just removing the streamflow from the input). After we train the models, the ungauged stations will be treated in the same way with the latter case.
>
> Reference:
>
> [4] Nearing, Grey S., et al. "Data assimilation and autoregression for using near-real-time streamflow observations in long short-term memory networks." Hydrology and earth system sciences discussions 2021 (2021): 1-25.
>
> **(To be continued…)**

---

> ### Author Response · Authors · 2025-12-02
> **Response to the Reviewer n4Tx follow-up comments (Part 3)**
>
> **W4.1.** *FlowNet is also a black box model. GNNs are not interpretable. In my view, adding the flow direction does not make the model interpretable, however, it simply adds more inductive bias to the model.* **W4.2.** *FlowNet also relies on these assumptions, so I think the same argument can be used against FlowNet, since we can’t build the graph or use the streamflow (argued in W3) in developing countries too. Just as a side note, Google's model is running globally in regions with less data as well.* **W4.3.** *FlowNet does not go into different direction from previous works. There are many works about distributed modelling in hydrology (see main review).*
>
> Regarding W4.1., most DL models are blackboxes and so is FlowNet and GNNs.*We did not mention in our paper anything about interpretability? And it is also not in the scope of our paper*. We carefully checked the paper to see if there are some points that led to that misunderstanding and removed them if we found them. But since the reviewer raised this, we would like to highlight an additional point here, the interaction scheme of FlowNet allows us to understand which stations will contribute more to the flow at a station via the cross-station models. Hence, we can know, for example:  station A flows more water to station B than station C. Hence if an incident happens in A, it can be more affected by B than those from  C. This can somehow help the management more than fully blackboxes such as GNNs. However, it is only a minor point in our opinion. We agree with the reviewer that designing a fully interpretable forecasting model is an interesting direction. *We already proven in the paper that our interaction mechanism via the flow graph helps to significantly improve the prediction accuracy compared to all SOTA methods as presented in Section 3 and Appendix*. And one reason is that we have a physical-guided interaction mechanism to drive these blackbox models to reach better learning states. Biases may happen, but the benefits we have overcome them already.
>
> Regarding 4.2., that is the reason why we introduce FlowNet since it will be particularly suitable for data unavailability. The streamflow graph is very easy to build even by hand since rivers constantly flow into the seas. It is the easiest data to have. The harder cases are climate and streamflow data. We demonstrated in Table 6 in the Appendix of the revised paper that the data in 26 stations of the MRB varied significantly with different data collection periods and climate features for different stations. And FlowNet can employ as much data as possible which other methods cannot and acquires significantly better performances.
>
> Regarding 4.3., first, we emphasize that **we did not do any distributed method as the reviewer claimed**. All of the existing works the reviewer pointed out either focusing on single large models that jointly predict multiple stations at the same time or all stations independently. Some of them such as [5] also exploit physical models like FlowNet. **However, none of the appointed existing works by the reviewer has the independent interaction global-local scheme via the flow graph like FlowNet**.
>
> References:
>
> [5] Bindas, Tadd, et al. "Improving river routing using a differentiable Muskingum‐Cunge model and physics‐informed machine learning." Water Resources Research 60.1 (2024): e2023WR035337.
>
> **(To be continued…)**

---

> ### Author Response · Authors · 2025-12-02
> **Response to the Reviewer n4Tx follow-up comments (Part 4)**
>
> **W6.** *I am not sure how the baseline was trained. The result of EDLSTM is available in both [Nature] and [HESS] where benchmark datasets are available. FlowNet can be compared to the reported number in the previous works on the benchmark.*
>
> In our paper, we use all 671 stations of the CAMELS dataset. However, in the appointed papers, the authors only choose 531 stations among 671 stations. Hence,  we retrained EA-LSTM using Python codes provided by the authors to report the results on 671 stations with the same experimental settings to be consistent with our results on FlowNet and other SOTA models we studied. We demonstrated in our previous response (and in Table 11 in the Appendix of our paper) that FlowNet acquires better prediction accuracy than EA-LSTM. The detailed training procedure was provided in the revised version of the paper in Section B.2 in the Appendix.
>
> **W.9.** *The table did not show a significant improvement with multi-scale. The improvement is +0.003, 0.001, and 0.001 for H=1, H=3, H=5, respectively. There is still no reason why the model doesn’t work without instance normalization.*
>
> Please note that the results are averaged for all 26 stations. This can cause a feeling that the difference is negligible. However, taking a deeper look at DMCT with and without multi-scale, the averaged NSE difference between them is 0.001 in H=3 and H=5. However, DMCT with multi-scale has better NSEs than without multi-scale on 17 over 26 stations with max NSE differences of 0.04 for Phung Hiep station. **Hence, the performance differences are much larger than appears through the averaged values.**
>
> *Table 5. Effects of different components of DMCT on the LamaH dataset, where w/o D, w/o M, w/o IN denote without Disentangled, Multiscale and Instance Normalization modules, respectively.*
>
> |     | DMCT       | w/o D  | w/o M  | w/o IN |
> | --- | ---------- | ------ | ------ | ------ |
> | H=1 | **0.6598** | 0.6481 | 0.6249 | 0.6200 |
> | H=3 | **0.4928** | 0.4836 | 0.4604 | 0.4236 |
> | H=5 | **0.4067** | 0.3970 | 0.3966 | 0.3193 |
>
> Additionally, in Table 20 of our revised paper (copied here as Table 5), we performed an ablation study on performances of different components of DMCT including Disentangled, Multiscale and InstanceNorm on LamaH dataset. The results show that each component contributes to the overall performance of DMCT, particularly, without multi-scale modules the performance decreased 5.28%, 6.57% and 2.48% in H=1,3 and 5, respectively. Each specific method may require a different data normalization technique. InstanceNorm is a well-proven method to improve prediction accuracy in many model general forecasting systems. In our case, we did try some different ways and DMCT works well with InstanceNorm.
>
> **(To be continued…)**

---

> ### Author Response · Authors · 2025-12-02
> **Response to the Reviewer n4Tx follow-up comments (Part 5)**
>
> **RE:** I would like to thank the authors for their detailed response. While I appreciate these efforts, I believe that the limited novelty, missing a proper comparison to baselines, and scalability of the framework are still major issues in this manuscript. Given these, I keep my rating for the current state of the manuscript.*
>
> We sincerely thank the reviewer for spending their precious time on our work. Throughout our discussions, we would like to clarify the remaining concerns of the reviewer below:
>
> **Novelty:** The reviewer said that *“FlowNet does not go into different direction from previous works. There are many works about distributed modelling in hydrology (see main review).”* However, none of the existing works pointed out by the reviewer has a similar or close approach to the iterative interaction global-local scheme of FlowNet. We also would hope to clarify misunderstandings from the Reviewer that we did not aim at interpretable models, real-time forecasting or distributed models, these are out-of-scope of our paper. In the response to **Reviewer wMmd (Q2)**, we made an extensive explanation how FlowNet differs from all existing works. We kindly hope the above points can clarify all of the concerns on novelty of the Reviewer?
>
> **Proper comparison to baselines:** **We extensively compared FlowNet to 20 different baselines including SOTA time series forecasting methods such as PatchTST, SOTA streamflow forecasting methods such as GNNs and AGCLSTM, time series forecasting foundation models like TimeFM and EA-LSTM as requested by the reviewer.** We explained that the original EA-LSTM is run only on 531 stations rather than 671 stations of the CAMELS dataset like FlowNet and all other studied models. Hence, we rerun EA-LSTM with the same setting so that the results are consistent. We kindly hope that these clear the concern on the comparisons? With 20 different wide-variety models, we believe that we are among the most comprehensive evaluations in the literature (most existing works including those pointed out by the Reviewer only compare with a few common methods like LSTM). **We also studied three datasets including LamaH and CAMELS, two most common datasets in the field, together with the MRB, a special dataset with irregular data.** We kindly hope that these can clear the concern of the comparisons to the Reviewer?
>
> **Scalability of the framework:** We explained clearly that the $O(N^2)$ edge case is not a realistic setting in streamflow forecasting and other relevant problem like traffic forecasting where we can apply FlowNet directly. In both of these applications, the number of edges is O(N)  (please refer to **Reviewer wMmd (W1) in Round 2**). We also demonstrated that the runtimes of FlowNet is not much higher than others (please also refer to **Reviewer wMmd (W1) in Round 2**). For example, for inference times, with 7.2s for FlowNet+DMCT (or 1.7s for FlowNet+RLinear), compared to 3.2s of PatchTST, 4.1s of MICN or 3.6s of TQNet on CAMELS, the runtime of FlowNet should not be a significant concern (FlowNet+RLinear is even faster than these methods with better prediction accuracy). For traffic forecasting, FlowNet is also slower than others but not so much, for example, 2.4s of FlowNet vs. 1.1s of PatchTST and 3.92s of ASTGNN, the SOTA traffic forecasting method. Similarly, in term of training time as we discussed before, FlowNet (DMCT) and FlowNet (RLinear) 3550.46 and 1505.65 seconds on the large CAMELS dataset, compared to 296.7s of AGCLSTM and 141.8s of ResGCN. *Hence the runtime of FlowNet is not a significant drawback, especially when it has much better prediction accuracy, which is more important*. We also discussed that if we have much larger numbers of stations, we can always split them into smaller areas and run FlowNet and other methods since it is a much more efficient way that people commonly do in the field. We kindly hope that these clear the concern of the Reviewer?
>
> **Summarize:** We kindly hope that we have addressed all of the remaining concerns from the reviewer effectively? We special thanks to the reviewer again for spending time reviewing our paper and helping us to improve our works.

---

### Official Review · Reviewer_LeBJ · 2025-10-30

**Soundness:** 3
**Presentation:** 3
**Contribution:** 3
**Rating:** 6
**Confidence:** 4

**Summary:**

This paper proposes FlowNet, the first independent and interactive modeling framework for streamflow prediction. By employing a well-designed local-global interaction scheme and a Disentangled Multiscale Cross-attention Transformer (DMCT), the method achieves advanced performance across three large benchmark datasets, demonstrating its strong potential for real-world hydrological forecasting.

**Strengths:**

1. The topic of flow forecasting is highly practical and relevant, as it can help mitigate uncertainty caused by climate change.
2. The proposed framework is flexible and can be adapted to various data sources and model types.
3. The experiments are thorough and extensive, convincingly demonstrating the effectiveness of the proposed approach.

**Weaknesses:**

1. Although FlowNet is novel as a framework, the proposed DMCT appears to mainly reuse existing attention mechanisms with relatively simple adaptation and normalization.
2. The Graph Links Reconstruction Module seems could be simplified by using weighted relationships between stations (e.g., distance-based weights) rather than binary adjacency in the downstream flow graph.
3. The ablation study on the local-global interaction scheme shows limited improvement. For instance, most NSE gains are below 0.01, and FlowNet even underperforms the global-only setting on Phung H. and Ban D. stations. More analysis is needed to explain these results.
4. The design of training multiple independent models for each station increases computational and memory costs, which may limit scalability.

Minor Issue:
- In line 367, the reference to “Figure 1” should be corrected to “Figure 2.”

**Questions:**

In Section 2.2, the paper defines a global loss to minimize differences between the ground truth $y^i$ and local prediction $\hat{y}^i$, as well as between global prediction $\hat{y}^i_{Global}$ and local prediction $\hat{y}^i$. Why not directly minimize the difference between the global prediction $\hat{y}^i_{Global}$ and the ground truth $y^i$? This seems more straightforward.

---

> ### Author Response · Authors · 2025-11-21
> **Response to the Reviewer LeBJ (Part 1)**
>
> Many thanks for your insightful comments. Please find our responses below. All changes were updated into our paper (highlighted in blue).
>
> **W1. Although FlowNet is novel as a framework, the proposed DMCT appears to mainly reuse existing attention mechanisms...**
>
> We fully agree with the reviewer. In FlowNet, the more effective the model in the local stage, the more effective the interaction process. This motivates us to study different existing forecasting models and choose the most effective components to build DMCT. Compared to FlowNet, which is a unique framework, DMCT is merely an engineering effort in our opinion. However, it acquired very good performance compared to other 18 existing models as we showed in Table 2 of the revised paper. When being used inside FlowNet, the overall performance is further boosted as also shown in Table 2.  We kindly hope that DMCT, as an additional side contribution, will not distract the reviewers from the novelty of our FlowNet framework.
>
> **W2. The Graph Links Reconstruction Module seems could be simplified by using weighted relationships between stations (e.g., distance-based weights) rather than binary adjacency in the downstream flow graph.**
>
> This is a very great suggestion. We also aim at studying different kinds of weighted streamflow graphs in our FlowNet framework in the near future. And the distance-based graph will be one of first most targets due to its simplicity.
>
> **W3. The ablation study on the local-global interaction scheme shows limited improvement. For instance, most NSE gains are below 0.01, and FlowNet even underperforms the global-only setting on Phung H. and Ban D. stations. More analysis is needed to explain these results.**
>
> In Figure 3, we randomly choose 6 stations to perform the visualizations in the local-global interaction study. For more detailed results, after the global interaction, 21 over 26 (80.7%) stations have improved with a max improvement of 0.022 in NSE for Can Tho station. Only 5 over 26 stations decreased their performance including Phung H. and Ban D. These demonstrated the effectiveness of the global interaction scheme of FlowNet. Obviously, we cannot expect that all stations will be improved but 80.7% improvements are a significant and convincing number.
>
> *Table 1. Mean ablation results of local and global phases compare to FlowNet on 3 datasets LamaH, CAMELS and MRB with prediction horizon setting H = 1.*
>
> |Datasets|CAMELS| | |LamaH| | |Mekong| | |
> |---|---|---|---|---|---|---|---|---|---|
> |Metrics|NSE|RMSE|MAE|NSE|RMSE|MAE| NSE| RMSE |MAE|
> |local| 0.5631| 257.8| 68.6      | 0.6503     | 8.002     | 4.389     | 0.9309     | 0.1853     | 0.1198     |
> |global|0.517| 285.8| 86.68     | 0.5963     | 8.701     | 5.259     | 0.9296     | 0.1875     | 0.1214     |
> | **FlowNet** | **0.5784** | **250.7** | **64.01** | **0.6598** | **7.792** | **4.385** | **0.9323** | **0.1796** | **0.1144** |
>
>
> Since the prediction accuracy of DMCT on the MRB data is already quite high (c.f. Table 2 on the revised paper), the improvement will be small when using it as a base model for FlowNet as we have seen in Figure 3. Hence, we conducted further studies on both LamaH and CAMELS. The results are shown in Table 1 above (also in Table 14 of the revised paper). The global interaction scheme significantly improves the prediction accuracy on all datasets and evaluation metrics. Concretely, on the CAMELS dataset, 571 over 671 (85.1%) stations have their NSEs improved after the global interaction phase with the maximum improvement of 0.7893. For LamaH, 399 over 425 stations (93.8%) show improvement in NSEs with max improvement of NSE value of 0.6374203.
>
> **W4. The design of training multiple independent models for each station increases computational and memory costs, which may limit scalability.** We fully agree with the reviewer. A drawback of the independent interactive scheme of FlowNet is that it leads to higher computational overhead in exchange for better prediction accuracy as also pointed out by other reviewers.
>
> However, in our responses to Reviewers wMmd (Q3) and EzYW (Q2), we also pointed out that for CAMELS, a very large dataset with 671 stations in the whole USA, it takes FlowNet (DMCT) and FlowNet (RLinear) 3550.46 and 1505.65 seconds for overall training on a Nvidia L4 GPU, respectively. The overall inference time for FlowNet with DMCT and RLinear barebones are 7.2 and 1.7 seconds on the same L4 GPUs, respectively. These times are affordable, especially when having much better prediction accuracy. We also discuss various ways to improve the runtimes of FlowNet in our responses to reviewers wMmd (Q3) and EzYW (Q2) and in Section C.11 in Appendix of the revised paper including splitting data into smaller areas and training/inference in parallel due to highly parallelable nature of FlowNet.
>
> **(To be continued…)**

---

> > ### Author Response · Authors · 2025-11-21
> > **Response to the Reviewer LeBJ (Part 2)**
> >
> > **Q1. In Section 2.2, the paper defines a global loss to minimize differences between the ground truth  and local prediction, as well as between global prediction and local prediction. Why not directly minimize the difference between the global prediction  and the ground truth? This seems more straightforward.**
> >
> > When updating the model in the global phase, we need to keep the prediction of a model to be close to its ground truth (i.e. $Loss(\hat{y}^i,y^i)$). At the same time, we do not want the prediction result of the model to be deviated too much from the ensembled prediction $\hat{y}^i_{Global}$ of 3 different prediction viewpoints (per-station, inflow stations and outflow stations) (i.e. $Loss(\hat{y}^i,\hat{y}^i_{Global}$)).
> >
> >
> > **Summarize:** We kindly hope that we can address all of your concerns effectively? Many thanks again for your great help on our paper.

---

> > > ### Comment · Reviewer_LeBJ · 2025-11-25
> > > **Follow-up Comment on Q1**
> > >
> > > Thank you for your response. Most of my concerns have been addressed, except for **Q1**. As you explained, the two distance terms serve different purposes. However, Eq. (2) introduces a balancing hyperparameter $\alpha$ between these two losses, and the ablation study in Appendix D (Fig. 6(A)) shows that performance generally improves as $\alpha$ approaches 1.
> > >
> > > This trend suggests that the model benefits more from the supervised term $\mathcal{L}(\hat{y}^i, y^i)$, while the contribution of the consistency term $\mathcal{L}(\hat{y}^i, \hat{y}^i_{\text{Global}})$ appears marginal or possibly unnecessary. Could the authors clarify whether this consistency loss is truly beneficial? If $\alpha \to 1$ consistently yields the best performance, does this imply that the global-local consistency term does not play an essential role in training?

---

> > > > ### Author Response · Authors · 2025-11-26
> > > > **Response to the Reviewer LeBJ Follow-up Comment on Q1**
> > > >
> > > > Many thanks for your comments. It is a very great point. In Figure 6A (now Figure 7A in our revised paper), we only showed $\alpha$ to 0.9. This leads to miss-understanding that the performance increases with $\alpha$. We updated Figure 7A adding $\alpha=1$ cases. The results show that for the prediction horizon H=3, and H = 5, the best values for $\alpha$ are both 0.9, rather than 1 when $H=1$. In addition, we performed another study on LamaH, which is much larger than MRB. The results are shown in Figure 8 in the revised paper. For $H=1$, $H=3$ and $H=5$, the best values for $\alpha$ are 0.9, 0.5 and 0.3, respectively. These show the effectiveness of the global interaction phase and the consistency loss.
> > > >
> > > > We kindly hope that we can address all of your concerns effectively? Many thanks again for your great help on our paper.

---

### Official Review · Reviewer_EzYW · 2025-11-05

**Soundness:** 3
**Presentation:** 3
**Contribution:** 3
**Rating:** 6
**Confidence:** 3

**Summary:**

FlowNet is a general framework designed for multivariate spatiotemporal runoff prediction. It introduces a local-global interactive modeling strategy, which allows individual station models to maintain their independence while mutually correcting predictions through iterative integration (Global Consensus). This approach enhances the consistency and robustness of predictions, making it particularly suitable for sets of stations with irregular input features and data lengths. Furthermore, FlowNet incorporates a directional and delay-aware graph reconstruction method to optimize the modeling of spatial relationships. Additionally, it proposes a decoupled multi-scale cross-attention Transformer (DMCT) as a backbone network for efficiently capturing temporal features.

**Strengths:**

1. FlowNet natively supports the processing of irregular datasets across monitoring stations — for example, when the length of historical records or the dimensionality of input features is inconsistent. This flexibility is of considerable practical importance in hydrological applications.

2. Comprehensive Baselines and Experiments: The paper employs three challenging large-scale hydrological datasets and conducts comparisons with up to 18 state-of-the-art methods of different architectures (including statistical, MLP, RNN, CNN, Transformer, and GNN approaches), demonstrating extensive empirical advantages.

**Weaknesses:**

1. **Relevance and Citation of Baseline Selection:** While a substantial number of baseline models are included, some—such as Informer, Autoformer, and FEDformer—are not directly discussed in the Related Work section regarding their application or limitations in hydrological contexts. It is recommended to briefly explain the rationale for including these models in the Related Work or experimental setup, ensuring that all baselines have been referenced in hydrology or time series forecasting.

2. **Clarity of the Ablation Study:** The current ablation results are primarily presented in graphical form (e.g., Figure 3). However, due to the relatively small performance differences, visual representations may not precisely reflect the quantitative contribution of each module. To enhance rigor, it is strongly recommended to supplement with detailed ablation performance tables for the core components of FlowNet—local learning, graph reconstruction, and global interaction—across all major datasets (LamaH-CE, CAMELS, MRB). Tabular data would allow clearer and more accurate assessment of the gain from each module.

**Questions:**

1. **Comparison with Large Time Series Models (TS-LLMs):** Given rapid advances in time series forecasting, large pre-trained models such as TimesFM, TimeGPT, and TabPFN have emerged as new state-of-the-art baselines. Have the authors considered including these as additional and more challenging baselines for comparison? This would further strengthen the demonstrated competitiveness of FlowNet.

2. **Computational Complexity and Efficiency:** Although FlowNet employs independent lightweight models, the global interaction phase is iterative. How does FlowNet’s actual overhead in inference time and training time compare to that of single end-to-end GNN models, such as ResGAT or AGCLSTM?

---

> ### Author Response · Authors · 2025-11-21
> **Response to the Reviewer EzYW (Part 1)**
>
> Many thanks for your insightful comments. Please find our responses below. All changes were updated into our paper (highlighted in blue).
>
> **W1. Relevance and Citation of Baseline Selection.**
>
> Thank you for pointing out this excellent point. We added the missing citations for these models and added a paragraph to our Related Work section explaining that we included these general time-series models (such as  iTransformer, FilterNet, TQNet, CycleNet, Informer, AutoFormer, FEDFormer, PatchTST, CAT, DLinear, RLinear, etc.) to ensure our comparison was not limited only to models specifically designed for hydrology as follows:
>
> *Recently, many SOTA methods in general time series forecasting have been introduced in the literature and archived SOTA performances on various time series benchmarks such as iTransformer (Liu et al., 2023), FilterNet (Yi et al., 2024), TQNet (Lin et al., 2025), CycleNet (Lin et al., 2024), Informer (Zhou et al., 2021), AutoFormer (Wu et al., 2021), FEDFormer (Zhou et al., 2022), PatchTST (Nie et al., 2023), CAT (Kim et al., 2024), DLinear (Zeng et al., 2023) or RLinear (Li et al., 2023). These methods can also be applied for streamflow forecasting. Hence, we include them into our study to
> ensure our comparison is not limited only to models specifically designed for hydrology.*
>
> We also added Section B.2 in Appendix to explain the rationale on choosing these models as our competitors in the paper. An example can be found below. Since the content is quite long, please refer to Section B.2 for full details.
>
> *MLP-based Models (DLinear, RLinear, CycleNet, TQNet, FilterNet, SOFTS, TSMixer). Recent work has shown that well-designed MLP architectures can match or surpass complex Transformer models while being highly efficient. DLinear, RLinear, Cycle, TQNet and FilterNet revealed the surprising strength of decomposition-based linear layers, reshaping the community’s understanding of linear models. Mixer-style models such as TSMixer and SOFTS introduced mixing for channel–temporal interactions, providing strong MLP-based baselines. These models collectively represent the competitive methods with lightweight design in modern time-series forecasting.*
>
> **W2. Clarity of the Ablation Study.**
>
> We completely agree. A table is much clearer for this analysis. We added detailed ablation study tables that present the quantitative results in Figure 3 as suggested in Section C.7, C.8, and C.9. These Tables are  also copied below.
>
> *Table 1. Mean ablation results of local and global phases compare to FlowNet on 3 datasets LamaH, CAMELS and MRB with prediction horizon setting H = 1.*
>
> | Datasets    | CAMELS     |           |           | LamaH      |           |           | Mekong     |            |            |
> | ----------- | ---------- | --------- | --------- | ---------- | --------- | --------- | ---------- | ---------- | ---------- |
> | Metrics     | NSE        | RMSE      | MAE       | NSE        | RMSE      | MAE       | NSE        | RMSE       | MAE        |
> | local       | 0.5631     | 257.8     | 68.6      | 0.6503     | 8.002     | 4.389     | 0.9309     | 0.1853     | 0.1198     |
> | global      | 0.517      | 285.8     | 86.68     | 0.5963     | 8.701     | 5.259     | 0.9296     | 0.1875     | 0.1214     |
> | **FlowNet** | **0.5784** | **250.7** | **64.01** | **0.6598** | **7.792** | **4.385** | **0.9323** | **0.1796** | **0.1144** |
>
>
> *Table 2. Mean ablation results of 3 different backbone models LSTM, GRU and DMCT with and without FlowNet on Mekong dataset with prediction horizon setting H = 1.*
>
> | Datasets       | LSTM   |        |        | GRU    |        |        | DMCT   |        |        |
> | -------------- | ------ | ------ | ------ | ------ | ------ | ------ | ------ | ------ | ------ |
> | Metrics        | NSE    | RMSE   | MAE    | NSE    | RMSE   | MAE    | NSE    | RMSE   | MAE    |
> | w/o FlowNet    | 0.8708 | 0.2612 | 0.1775 | 0.8862 | 0.2416 | 0.1631 | 0.9309 | 0.1853 | 0.1198 |
> | **w/ FlowNet** | **0.899**  | **0.2017** | **0.1344** | **0.894**  | **0.1969** | **0.1285** | **0.9323** | **0.1796** | **0.1144** |
>
>
> *Table 3. Mean ablation results of DMCT, with Original graph and with PearCorr VLR on Mekong
> dataset with prediction horizon $H \in \{1, 3, 5\}$.*
>
> | Horizons        | Horizon 1 |        |        | Horizon 3 |        |        | Horizon 5 |        |        |
> | --------------- | --------- | ------ | ------ | --------- | ------ | ------ | --------- | ------ | ------ |
> | Metric          | NSE       | RMSE   | MAE    | NSE       | RMSE   | MAE    | NSE       | RMSE   | MAE    |
> | DMCT            | 0.9309    | 0.1853 | 0.1198 | 0.8889    | 0.2985 | 0.1868 | 0.853     | 0.3802 | 0.2391 |
> | w/ Original     | 0.8583    | 0.1918 | 0.2772 | 0.8012    | 0.3966 | 0.2673 | 0.7607    | 0.4762 | 0.3174 |
> | w/ PearCorr VLR | **0.9323**    | **0.1796** | **0.1144** | **0.8908**    | **0.2945** | **0.1835** | **0.8555**    | **0.3757** | **0.2354** |
>
>
> **(To be continued…)**

---

> ### Author Response · Authors · 2025-11-21
> **Response to the Reviewer EzYW (Part 2)**
>
> We also added Section C.10 in Appendix, studying the effect of KNN graphs on the CAMELS dataset. The results are also copied below.
>
> *Table 4. The results of different KNN settings on CAMELS of FlowNet with different horizon length $H\in\{1, 3, 5\}$.*
>
> |Horizons|1| | |3| | |5| | |
> |---|---|---|---|---|---|---|---|---|---|
> |Metrics|NSE|RMSE|MAE|NSE|RMSE|MAE|NSE|RMSE|MAE|
> |knn=2|**0.5784**|**250.7**|**64.01**|**0.4228**|**316.2**|**93.97**|**0.354**|**342.8**|**109.9**|
> |knn=5|0.5442|267.3|75.76|0.4106|323.2|101.2|0.3427|349|117.1|
> |knn=10|0.3784|323|102.4|0.3906|328.6|105.6|0.3333|352.2|121.5|
>
>
> **Q1. Comparison with Large Time Series Models (TS-LLMs).**
>
> We thank the reviewer for this forward-looking suggestion. These large pre-trained models (e.g., TimesFM) are a very recent and exciting paradigm. Hence understanding performances between them and FlowNet is interesting.
>
>
> Tables 5 and 6 below report the forecasting performance of TimesFM and FlowNet on CAMELS and LamaH datasets. These tables show that FlowNet achieves better performance across the majority of metrics on NSE and RMSE. Moreover, it takes TimesFM more than 18 hours for inferences while FlowNet takes only 7.2 seconds on CAMELS. We also included these comparisons into Section C.6 in the Appendix of the revised paper.
>
> *Table 5. Mean results of FlowNet compared to TimesFM on CAMELS with 3 different prediction
> horizon settings $H\in\{1, 3, 5\}$. Best results are highlighted in bold.*
>
> | CAMELS  | H=1        |           |           | H=3        |           |       | H=5       |           |        |
> | ------- | ---------- | --------- | --------- | ---------- | --------- | ----- | --------- | --------- | ------ |
> | Metrics | NSE        | RMSE      | MAE       | NSE        | RMSE      | MAE   | NSE       | RMSE      | MAE    |
> | TimesFM  |  0.5577 | 255.7 | **61.77** | 0.3746 | 327.2 | **91.99** | 0.2952 | 358.5 | **109.5** |
> | **FlowNet** | **0.5784** | **250.7** | 64.01 | **0.4228** | **316.2** | 93.97 | **0.354** | **342.8** | 109.9  |
>
>
> *Table 6. Mean results of FlowNet compared to TimesFM on LamaH with 3 different prediction
> horizon settings $H\in\{1, 3, 5\}$. Best results are highlighted in bold.*
>
> |LamaH|H=1| | |H=3| | |H=5| | |
> |---|---|---|---|---|---|---|---|---|---|
> |Metrics|NSE|RMSE|MAE|NSE|RMSE|MAE|NSE|RMSE|MAE|
> |TimesFM|0.63|8.304|**4.002**|0.4376|11.97|**6.266**|0.3517|13.5|**7.491**|
> |**FlowNet**|**0.6598**|**7.792**|4.385|**0.4928**|**11.25**|6.514|**0.4067**|**12.7**|7.536|
>
>
> References:
>
> [1] A decoder-only foundation model for time-series forecasting, ICML, 2024.
>
>
> **Q2. Computational Complexity and Efficiency.**
>
> Let us take CAMELS dataset with prediction horizon $H=1$ as an example, FlowNet with DMCT barebone requires 3550.46 seconds for training, while ResGAT and AGCLSTM consumes 110.81 and 296.70 seconds for training, respectively, which are much faster than FlowNet. It is not a surprise since these models can exploit the processing power of GPUs when calculating large tensor data. Moreover, FlowNet must incur significant overheads for its iterative interaction scheme. In terms of inference time, FlowNet with DMCT barebone, ResGAT and AGCLSTM consumes 7.2, 0.39, and 0.69 seconds, respectively. However, the slow runtimes of FlowNet return in much better prediction accuracy of mean NSEs on 671 stations of 0.5784 compared to 0.5414 and 0.1956 of ResGAT and AGCLSTM, respectively, a considerably runtime accuracy trade-off. When using RLinear barebone, which is faster than DMCT, the training and inference times for FlowNet reduce to 1505.65 and 1.71 seconds, respectively.
>
> The runtime of FlowNet, however, can be significantly reduced via some training strategies such as predicting smaller groups of stations in smaller areas rather than predicting all stations in an enormous area and employing parallel processing since all models in FlowNet can be trained independently in both the local and global phases.
>
> We further discussed the runtime problem of FlowNet and potential solutions in Section C.11 in the Appendix.
>
> **Summarize:** We kindly hope that we can address all of your concerns effectively? Many thanks again for your great help on our paper.

---

### Official Review · Reviewer_wMmd · 2025-11-06

**Soundness:** 2
**Presentation:** 3
**Contribution:** 2
**Rating:** 4
**Confidence:** 4

**Summary:**

**Summary**
This paper proposes **FlowNet**, a flexible and generalizable framework for multivariate spatio-temporal streamflow forecasting across multiple gauge stations. It introduces an **interactive local-global modeling strategy**, where each station is modeled independently in a local phase and then iteratively refined via cross-station interactions in a global phase. A novel **DMCT (Disentangled Multiscale Cross-attention Transformer)** is also proposed as a backbone model to capture multiscale temporal patterns. The method is evaluated on three hydrological datasets and shows superior performance over 18 SOTA baselines.

**Strengths:**

**Strengths**
- **Extensive and convincing experiments**: The paper compares FlowNet with a wide range of strong baselines (e.g., Transformers, GNNs, LSTMs) across three datasets (LamaH, CAMELS, Mekong) and multiple horizons. Results consistently show FlowNet outperforms others in NSE, RMSE, and MAE.
- **Ablation studies and robustness checks**: Ablations validate the contribution of each component (local/global phases, graph reconstruction, DMCT modules), and multiple random seeds are used to ensure stability.
- **Handles heterogeneous data**: FlowNet accommodates varying input lengths and feature sets across stations, which is a practical advantage in real-world hydrological systems.

**Weaknesses:**

**Weaknesses**
- **Poor readability in methodology**: The method description is overly technical and difficult to follow, with inconsistent notation and a lack of high-level intuition. Variable definitions are scattered and not unified, hindering clarity.
- **Limited novelty in integration**: The local/global phase learning and DMCT module feel like a straightforward combination of existing ideas (interactive learning + multiscale decomposition) rather than a deeply integrated innovation. The novelty appears incremental.
- **Scalability concerns**: FlowNet requires training separate per-station and cross-station models for each link in the graph. With hundreds or thousands of stations, the computational cost grows sharply, severely limiting its applicability to large-scale river networks. This is only briefly mentioned but not adequately addressed.

**Questions:**

1.  **Readability:** The methodology is dense and hard to follow due to inconsistent notation. Can the authors provide a clearer, more unified presentation of variables and core concepts?

2.  **Novelty & Integration:** The framework feels like a composition of a local-global scheme and a DMCT backbone. What is the key synergistic novelty beyond this combination?

3.  **Scalability:** The requirement for O(N²) models seems prohibitive for large networks. What is the formal computational complexity, and what strategies are proposed for scaling to real-world, large-scale basins?

---

> ### Author Response · Authors · 2025-11-21
> **Response to the Reviewer wMmd (Part 1)**
>
> Many thanks for your insightful comments. Please find our responses below. All changes were updated into our paper (highlighted in blue).
>
> **Q1. Readability: The methodology is dense and hard to follow…**
>
> First, we added Table 1 into the beginning of Section 2 to summarize all key notations in our paper. We hope that it will help to enhance readability of the paper. We also copy the Table 1 below.
>
> *Table1. Summary of notations in the paper.*
>
> | Notation                                             | Definition                                                               |
> | ---------------------------------------------------- | ------------------------------------------------------------------------ |
> | $X\in\mathbb{R}^{T\times C}$                         | multivariate time series with time steps $T$ and channel $C$             |
> | $S=\{S_1,\dots,S_N\}$                                | the set of $N$ hydrology stations in river networks                      |
> | $A\in\mathbb{B}^{N\times N}$                         | the adjacent matrix of stations                                          |
> | $\mathcal{G} = (S, A)$                               | a directed graph of stations                                             |
> | $x^i_t=X^i_{t-L:t, :} \in \mathbb{R}^{L\times C_i}$  | the historical data at station $S_i$ from $t-L$ to $t$                   |
> | $x^{i,j}=X^j_{t-L:t,:}\oplus X^i_{t-L:t,c}$          | a concatenated historical data of $X^j$ and historical flow of $X^i$     |
> | $F=\{f_k\|k=1..M\}$                                  | a set of $M$ small independent models                                    |
> | $f_i$                                                | per-station model of station $S_i$                                       |
> | $f_{i,j}$                                            | cross-station model of stations $S_i,S_j$                                |
> | $y_t^i = X_{t+1:t+H, c}^i \in \mathbb{R}^{H \times 1}$ | the future data of $c$-th channel associate from historical data $x^i_t$ |
> | $\hat{y}^{i}=f_{i}(x^{i})$                           | a prediction outcome of station $S_i$                                    |
> | $\hat{y}^{i,j}=f_{i,j}(x^{i,j})$                     | the prediction outcome of $f_{i,j}$                                      |
> | $\hat{y}^{i}_{inflow}$                               | the predicted inflow of $S_i$                                            |
> | $\hat{y}^{i}_{outflow}$                              | the predicted outflow of $S_i$                                           |
> | $\mathbf{s},\mathbf{t}$                              | the seasonality/trend components of time series                          |
> | $\mathbf{h}$                                         | the multiscale latent sequences of Transformer block                     |
> | $\Omega$                                             | the maximum lag threshold for Pearson Correlation Analysis               |
> | $\lambda$                                            | the maximum correlations under $\Omega$ lags                             |
>
>
> Second, we carefully scanned the paper to find any source of inconsistency or misleading and fixed them. Particularly,
>
> * We updated Definition 2, changing $x^i_t=X^i_{t:t-L,:}$ to $x^i_t=X^i_{t-L:t,:}$, to make the time dimension more consistent.
> * We updated Definition 3, adding the time $t$ back rather than omitting it so that it is consistent with Definition 2. However, for simplicity, we still omit the time $t$ from the models in Section 2.2 to avoid over complicated formulas. We also fixed a typo that $\hat{y}^i$ is the prediction ground truth into prediction outcome, which may be the main source of confusions.
> * In Section 2.3, we make a note that the station index $i$ will be omitted since it is clear from the context. This helps to simplify the formulas in Section 2.3, thus enhancing readability.
> * In Section 2.4, Equation 8 was updated, changing $X_{i,c}$ to $X^i_c$ for consistency with the station data definition in Section 2.1.
>
> **(To be continued…)**

---

> > ### Author Response · Authors · 2025-11-21
> > **Response to the Reviewer wMmd (Part 2)**
> >
> > Third, we carefully revised the paper, aiming at enhancing high-level intuition and clearer technical justifications as suggested. Particularly,
> >
> > * We moved the problem formulation into the beginning of Section 2, right after the definition notion of the multivariate time series as follows: *“Let $S=\{S_1,\dots,S_N\}$ be the set of $N$ hydrology stations in river networks. Let $\mathcal{G}$ be a directed graph that connects these stations via their direct flow relationships as follows. For each station $S_i$, let $X^i\in\mathbb{R}^{T_i\times C_i}$ be the multivariate time series data associated with it, where the $c$-th channel contains streamflow values and the remaining channels are exogenous variables such as climate. Notably, the number of time steps $T_i$ and the number of channels $C_i$ are station-specific. We aim to predict future flow values (i.e. the $c$-th channel) of all stations jointly.”*
> > * We revised and highlighted the high-level key concepts of FlowNet together with their justifications after the problem formulation as follows: *”**Key concepts of FlowNet.**
> > Though all existing works, such as (Zhao et al., 2020), focus on a single large joint prediction model $f$ for all stations, we follow an entirely different approach that constructs a set of $M$ small independent models $F=\{f_k|k=1..M\}$ and train them in two different phases: the **local** and **global** ones. Initially, in the *local* phase, each $f_k$ predicts the flow value of a single station independently in the beginning and belongs to one of the 3 categories including **per-station** (i.e. measured flow at a station), **inflow** (i.e. water flow into a station) and **outflow** (i.e. water flow out a station) forecasting as described in Section 2.1. After that, in the **global** phase, these models *iteratively interact* with others to adjust themselves via the flow graph $\mathcal{G}$ to maximize their prediction agreements, thus reducing uncertainty and increasing their accuracy (cf. Section 2.2). This setting is **data flexible**, since in many river basins like the MRB, the time series data lengths and collected hydrology features can be very different at different stations. Rather than choosing only a uniform subset of data for all stations like most existing works such as (Kirschtein et al., 2024), FlowNet can effectively utilize all of them due to its **independent** learning model to maximize learning generability. Concretely, each model can be trained in the local phase using all available data locally. During the global phase, the data is limited to common data of participated models. Note that some recent HGNN-based methods such as (Jiang et al., 2024) can deal with data heterogeneously but still require uniform data for nodes with the same type.  Moreover, FlowNet is also **model flexible**, any existing deep learning model can be independently used as $f_i$, including lightweight models that are computationally efficient and less prone to overfitting, especially when training data are limited. The interaction and ensemble fusion concepts of FlowNet among **per-station**, **inflow**, and **outflow** models also help it to produce more stable results than existing works.”*
> >
> > We kindly hope that all of these modifications make the paper easier to read and understand. Please do not hesitate to let us know if there are other typos or you would like to suggest some changes. We will be very happy to revise our paper toward a better version.
> >
> > **Q2. Novelty & Integration: The framework feels like a composition of a local-global scheme and a DMCT backbone. What is the key synergistic novelty beyond this combination?**
> >
> > As we mentioned in the paper, all existing works in streamflow forecasting follow two different approaches. Most of them treat studied stations independently, thus ignoring their spatial relationships and water flow direction dependencies among stations. Some other approach constructs a single model to jointly all stations. The relationships among stations are either (i) implicitly embedded inside the models by putting all data into models such as LSTM (Kratzert et al., 2018) and letting the models to discover relationships themselves (hopefully) or (ii) explicitly stated via feature embedding mechanisms of GNNs such as (Kirschtein et al., 2024) before predicting streamflows.
> >
> > Compared to existing works, FlowNet follows an entirely different concept. **To the best of our knowledge, there exists no other methods with the iterative and interactive scheme like FlowNet for streamflow forecasting**. The key synergistic innovations and concepts of FlowNet are as follows:
> >
> > **(To be continued…)**

---

> ### Author Response · Authors · 2025-11-21
> **Response to the Reviewer wMmd (Part 3)**
>
> **K1. Physic-informed interactions considering both inflow and outflow aspects:** Constructing independent models for predicting streamflow at each station $S_i$ via 3 different perspectives: (i) per-station model - predicting using local station data; (ii) inflow model - predicting from water flows in $S_i$ from upstream stations; and (iii) outflow model - predicting water flow out from $S_i$ to downstream stations. These models are inspired by streamflow physics, particularly in and out flows at stations. Some existing works like GNN approaches also exploit these flow physics. However, they are limited by the way **input** data from nearby stations of $S_i$ are jointly embedded by feature propagation schemes of GNNs before feeding to models to predict $S_i$. FlowNet, however, targets the relationships via model **outcomes** rather than inputs like GNNs and focuses on enhancing consistency and accuracy among 3 different perspectives. Moreover, since each type of model uses different input data (from $S_i$ and their parents/childs stations), forecasting results of each station are also determined via different input aspects like GNNs. So, FlowNet in general consists of diverse input data powers like GNNs and its own interactive interaction scheme for model outputs.
>
> **K2. Local and global interaction scheme.** The interaction scheme of FlowNet is also specifically designed into two phases. The key intuition behind that design is that to the model interaction and adjustment to be effective, each model needs to be good enough. Hence, the local phase aims at training each model to reach a good state before the interaction. The global phase is also designed so that the interaction scheme follows the flow physics with **learnable deep network for inflow and outflow contributions** under an ensemble fashion.
>
> **K3. Ensemble outcome from diverse inputs and perspectives.** Normally, we forecast streamflow at each station via its own local data (e.g. flow and climate). In FlowNet, per-station models also use the local data. However, inflow models use data from parent stations and outflow models use data from child stations. These models also capture two important aspects of streamflow in each station: water flow in and water flow out. These made the final prediction of each station to be determined via diverse input data and flow perspectives, especially under our ensemble strategy. These make the results of FlowNet not only more accurate (as shown in Table 2 in the revised paper) but also more stable (as shown in Table 7 in the revised paper).
>
> **K4. Data and model flexibility.** In many areas like the Mekong River Basin (MRB), the collected data for each station can be very different (in terms of data collection periods and relevant climate features as we demonstrated in Table 6 in Appendix for MRB). FlowNet with its independent model training and interaction is also designed to cope with this problem. Particularly, in the local phase, each model can be trained using all possible local data. And during the global interaction phase, the interaction can be done on all common available data for each group of stations. These ensure that we will exploit as much data as possible to further enhance the performance. Other existing works do not have that much data flexibility like FlowNet. Moreover, as a generic framework, FlowNet is entirely model flexible, all possible models can be employed. Even each station can employ different kinds of models that fit with its most. We showed in Figure 3 that FlowNet can help to boost performances of studied models such as LSTM, GRU and DMCT.
>
> **K5. Interaction filtering.** As we mentioned in the paper, if two stations are staying very far away from each other or are affected severely by human intervention like dam operations or water irrigation, the interaction may not bring up positive effects. Hence we introduced a dynamic graph link reconstruction component (VLR) aiming at reducing ineffective interactions during the global interaction phase.
>
> **K6. The Disentangled Multiscale Cross-attention Transformer (DMCT).** As we mentioned in K2, the more effective the model in the local stage, the more effective the interaction process. This motivates us to study different existing forecasting models and choose the most effective components to build DMCT. Compared to FlowNet, which is a highly unique framework, DMCT is merely an engineering effort in our opinion. However, it acquired very good performance compared to other 18 existing models as we showed in Table 2 of the revised paper. When being used inside FlowNet, the overall performance is further boosted as also shown in Table 2.  We kindly hope that DMCT, as an additional side contribution, will not distract the reviewers from the novelty of our FlowNet framework.
>
> **(To be continued…)**

---

> > ### Author Response · Authors · 2025-11-21
> > **Response to the Reviewer wMmd (Part 4)**
> >
> > Overall, we kindly hope that our discussions on the novelty of FlowNet from K1 to K6 above can help to clarify the synergistic innovations and concepts of FlowNet as requested. The real novelty of the paper is not lying on key words like interactive learning or seasonality decomposition but on the way the interaction process is actually designed taking into account different aspects including flow physics, input diversity and output interaction guided by the flow graph, different flow aspect ensemble, heterogeneous hydrological data and model flexibility, interaction filtering and advanced base model like DMCT as we clarified above from K1 to K6.
> >
> >
> > **Q3. Scalability: The requirement for O(N²) models seems prohibitive for large networks. What is the formal computational complexity, and what strategies are proposed for scaling to real-world, large-scale basins?**
> >
> > This is indeed a very good and important point. Please let us clarify some points below.
> >
> > First, the flow graphs in an area normally follow forest styles (i.e. a collection of tree-based components) due to the flow natures of rivers (circle connections may still happen but not very common) as we can see from Figure 4 for LamaH and Figure 5 for MRB data. Hence, the numbers of edges are typically smaller than the numbers of stations. For example, in LamaH, we have 452 stations but there are only 372 edges. For MRB data, we have 26 stations and 24 edges. Even for CAMEL-US, due to the absence of flow graphs, we use KNN graphs as a replacement, the number of edges is bounded by $K \cdot$ 671 stations. Hence, in real-world settings, the number of edges is $O(N)$, where $N$ is the number of stations.  Hence the number of models is also $O(N)$ (actually around $3\cdot N$) and not $O(N^2)$ luckily.
> >
> > Second, nevertheless, we fully agree with the reviewer that our FlowNet framework would be slower than ordinary approaches of treating stations independently or using a single large model for all stations during the training time due to higher numbers of models (including inflow and outflow ones). As also shown in Table 4, FlowNet (using DMCT backbone) is among the most expensive methods together with SOFT, iTransformer, PatchTST and MICN. *However, its slower training time results in significantly better prediction accuracy than other methods as demonstrated in Table 2 and Figure 2 in the revised paper.* *And in our paper, we aim at enhancing prediction accuracy rather than training speeds.* Moreover, in terms of inference times, FlowNet (with DMCT) is only slightly slower than ordinary approaches as we demonstrated in the below Table 2. FlowNet with DMCT takes 7.208s for inference while DMCT and MICN take 5.358s and 4.102s, respectively. FlowNet uses an acceptable extra inference time to get a significant performance boost. When using the much faster basebone like RLinear, the runtime of FlowNet will be much smaller (only 1.71 seconds). We also added Table 2 into the revised paper as Table 18 in the Appendix.
> >
> > *Table 2: Inference time for different methods.*
> >
> > |Methods|Inference Time(s)|
> > |---|---|
> > |FlowNet (DMCT)|7.208|
> > |FlowNet (RLinear)|1.717|
> > |DMCT|5.358|
> > |DLinear|1.144|
> > |RLinear|1.234|
> > |CycleNet|3.018|
> > |TQNet|3.611|
> > |FilterNet|1.649|
> > |iTransformer|2.999|
> > |SOFTS|2.577|
> > |CATS|3.147|
> > |TSMixer|1.891|
> > |LSTM|1.134|
> > |GRU|1.001|
> > |PatchTST|3.2|
> > |MICN|4.102|
> > |GCN|0.325|
> > |GCNII|0.35|
> > |ResGCN|0.37|
> > |ResGAT|0.394|
> > |AGCLSTM|0.681|
> >
> > Third, we demonstrated FlowNet on two very large datasets LamaH (with 452 stations in the Central EU) and CAMEL-US (with 671 stations in USA). They are among the largest available streamflow datasets in the literature already.
> >
> > Fourth, you have raised a very interesting question of how to reduce training times for FlowNet. Please find some of our strategies below:
> >
> > * **S1: Predicting sub-area**: Rather than predicting a large area, we can divide it into smaller catchment areas and perform FlowNet on these sub-areas instead. For example, CAMEL-US covers the whole USA. In such an enormous area, the geographics, topography, and climate conditions will be significantly different for different locations, leading to significantly different flow characteristics in different stations at different subareas. For example, stations in mountain areas tend to have faster and unstable flows than in plain areas. Trying to jointly predict these stations can be ineffective. Hence, by focusing on smaller areas, we can reduce the number of graph links and models, thus significantly enhancing the overall performance.
> >
> > * **S2: Parallel processing**: In terms of algorithmics, FlowNet is a highly parallelable method. For example, during the local phase, per-station, inflow and outflow models can all be trained in parallel due to their independences. Similarly, the interaction during the global phase can also be parallelized quite straightforwardly.
> >
> >  **(To be continued…)**

---

> > > ### Author Response · Authors · 2025-11-21
> > > **Response to the Reviewer wMmd (Part 5)**
> > >
> > > We also updated our paper, adding discussions on the scalability of FlowNet into the revised paper (Section C.11 in Appendix).
> > >
> > > **Summarize:** We kindly hope that we have addressed all of your concerns effectively. We also noticed that your rating score was adjusted (from 4 to 2) shortly after the review release. This makes us quite concerned that there may be critical shortcomings or misunderstandings we have missed. May we ask if there are specific reasons for this adjustment so that we can try our absolute best to address them during this discussion period?
> > >
> > > Many thanks again for your time and your help in improving our paper.

---

> > > > ### Comment · Reviewer_wMmd · 2025-11-25
> > > > **Official Comment by Reviewer wMmd**
> > > >
> > > > I appreciate the substantial effort the authors have put into the rebuttal. Specifically, the revisions regarding readability are effective, and the manuscript is now much clearer. I also acknowledge the extensive supplementary experiments included to addressthe reviewers' comments. Consequently, I have decided to raise my score from 2 to 4.
> > > >
> > > > However, I cannot give a higher score because the responses to my two main concerns remain somewhat unconvincing:
> > > >
> > > > 1.  **Regarding scalability:** The authors effectively admit that their method incurs high computational complexity. While this increase may be less pronounced in the specific context of tree-like river structures, it implies that the method's generalizability is limited.
> > > > 2.  **Regarding novelty:** I maintain the view that the novelty of this work stems primarily from the specific problem setting rather than the methodology. The technical innovation itself appears limited.

---

> ### Author Response · Authors · 2025-12-02
> **Response to the Reviewer wMmd follow-up comments (Part 1)**
>
> We appreciate the time and efforts of the reviewer very much. We also special thank the reviewer for raising their rating score. However, we would hope to have some further discussions on the remaining concerns.
>
> **W1:** *Regarding scalability. The authors effectively admit that their method incurs high computational complexity. While this increase may be less pronounced in the specific context of tree-like river structures, it implies that the method's generalizability is limited.*
>
> The main focus of our paper is on streamflow forecasting, an important research problem with enormous research efforts not only in water research and environment but also in AI communities recently. For that purpose, we demonstrated the runtime performance of FlowNet on a large scale dataset CAMEL with KNN simulated graph (rather than tree-based flow graph due to its unavailability) and 671 stations. For inference times, with 7.2s for FlowNet+DMCT (or 1.7s for FlowNet+RLinear) as shown in **Table 2 of Q3** above, compared to 3.2s of PatchTST, 4.1s of MICN or 3.6s of TQNet, the runtime of FlowNet should not be a significant concern (FlowNet+RLinear is even faster than these methods with better prediction accuracy). Moreover, for forecasting problems, people usually care more about prediction accuracy than about runtime (except real-time forecasting which is out-of-scope of this paper), especially when accurately forecasting can save many lives and properties as in the case of streamflow forecasting. That is also why Time Series Foundation Models, that are trained *“on massive diverse datasets to develop generalizable forecasting abilities for time series data”*, have been emerged recently despite their enormous runtime as also kindly pointed out by the **Reviewer EzYW** below such as TimesFM. In our response for the **Reviewer EzYW**, we showed that FlowNet acquires better accuracy than TimesFM while running much faster (7.2 seconds vs. 18 hours).
>
> If we understand the reviewer's comment correctly, we believe that the reviewer already agreed with our responses before and above on runtimes of FlowNet for streamflow forecasting. And *the scalability concern the review raised here is generalizability when using FlowNet for other forecasting problems rather than streamflow forecasting, where the representation graph can be $O(N^2)$ rather than $O(N)$ of the downstream flow graphs (N is the number of stations)*? If so, please find some of our thoughts below.
>
> * First, FlowNet is originally designed for streamflow forecasting where the number of edges among $N$ stations are typically smaller than $N$ as we demonstrated in **Q3** above. An $O(N^2)$ edge case simply does not exist in this streamflow setting in reality.
>
> * Second, though FlowNet is designed for streamflow forecasting, it can also be straightforwardly applied in other application domains such as traffic flow forecasting on road networks. Here, we can consider traffics as flows. The $\beta$ index (edge to node ratio) for planar road network is max 3.0 [A, B]. For example, the urban road network in Zurich contains 6,629 nodes and 9,883 edges ($\beta=1.49$). Hence the $O(N^2)$ edge case will not happen too.
>
> **Hence, though we understand the concern of the reviewer on $O(N^2) edges$, it simply does not apply for the studying streamflow problem and some other relevant tasks which FlowNet can also be used for, like the traffic forecasting problem.** We will also demonstrate performances of FlowNet on traffic prediction below including accuracy and runtimes.
>
> References:
>
> [A] Casali, Ylenia, and Hans R. Heinimann. "A topological analysis of growth in the Zurich road network." Computers, Environment and Urban Systems 75 (2019): 244-253.
>
> [B] Boeing, Geoff. "Planarity and street network representation in urban form analysis." Environment and Planning B: Urban Analytics and City Science 47.5 (2020): 855-869.
>
> **(To be continued…)**

---

> ### Author Response · Authors · 2025-12-02
> **Response to the Reviewer wMmd follow-up comments (Part 2)**
>
> **W2:** *Regarding novelty. I maintain the view that the novelty of this work stems primarily from the specific problem setting rather than the methodology. The technical innovation itself appears limited.*
>
> We may not fully understand the concern of the reviewer here. Hence, we will try as much as possible to cover all possible understanding ways.
>
> First, streamflow forecasting is a very important research problem with high impacts on reality. **FlowNet, based on the nature of flows, introduced a very unique iterative interaction approach, which has not been introduced before in the field**. Compared to existing works, beside its technical uniqueness, FlowNet has some advantages such as data and model flexibility (as we clearly presented in the paper and re-clarified in **Q2** above). It also acquires SOTA results on some main benchmarks such as LamaH and CAMELS. We agree with the reviewer that FlowNet is built upon the particular concept of downstream water  flows. However, we kindly disagree with the reviewer that this makes the technical innovation limited. Physic-informed ML has been an emerging topic to break the current performance boundary as we can see in [C,D,E]. These works usually introduce different innovative ways to incorporate domain knowledge into DL models and these are clearly non-trivial tasks. We can also take  image classification in Computer Vision as another context, where most methods rely on characteristics of images but  we cannot say that they are limited in novelty because of that.
>
> Second, if what the reviewer means is that FlowNet is only limited to streamflow forecasting, then it can straightforwardly apply to other domains such as traffic forecasting as mentioned above. Tables 1 and 2 below demonstrate prediction accuracy and runtimes of FlowNet on PEMS08 dataset [E], which contains 170 nodes and 295 edges of traffic data in San Bernardino from July to August 2016. FlowNet acquires much better prediction accuracy than others, including ASTGNN [E], which is specifically designed for traffic flow forecasting. This comes with a cost that FlowNet is slower than others but not so much, for example, 2.4s of FlowNet vs. 1.1s of PatchTST and 3.92s of ASTGNN.
>
> *Table 1. The results of traffic forecasting tasks on PEMS08 dataset with different forecasting horizons $H\in\{1,3,5\}$.*
>
> | Horizons     | 1     |       | 3     |       | 5     |       |
> | ------------ | ----- | ----- | ----- | ----- | ----- | ----- |
> | Metrics      | RMSE  | MAE   | RMSE  | MAE   | RMSE  | MAE   |
> | DLinear      | 19.16 | 13.66 | 20.8  | 14.7  | 22.18 | 15.58 |
> | iTransformer | 18.46 | 13.2  | 19.7  | 14.27 | 20.17 | 14.46 |
> | SOFTS        | 18.77 | 13.61 | 19.51 | 14    | 20.19 | 14.35 |
> | LSTM         | 18.29 | 13.06 | 19.43 | 13.8  | 20.32 | 14.42 |
> | PatchTST     | 19.16 | 14.02 | 20.16 | 14.87 | 20.83 | 15.32 |
> | ASTGNN       | 26.21 | 18.66 | 24.24 | 17.04 | 23.73 | 16.49 |
> | FlowNet      | **17.51** | **12.59** | **18.53** | **13.18** | **19.26** | **13.64** |
>
>
> *Table 2.  Inference time for different methods on PEMS08 dataset.*
>
> | Horizons     | 1      | 3      | 5      |
> | ------------ | ------ | ------ | ------ |
> | DLinear      | 0.3296 | 0.3735 | 0.2458 |
> | iTransformer | 0.9854 | 1.0972 | 0.9435 |
> | SOFTS        | 0.9400 | 0.9998 | 0.8402 |
> | LSTM         | 0.7482 | 0.7549 | 0.6965 |
> | PatchTST     | 1.1696 | 1.0636 | 1.0147 |
> | ASTGNN     | 3.92| 3.75 | 6.12 |
> | FlowNet      | 2.4343 | 2.5051 | 2.5921 |
>
>
> References:
>
> [C] The Merit of River Network Topology for Neural Flood Forecasting, ICML, 2024.
> [D] Spatial and Temporal Aware Graph Convolutional Network for Flood Forecasting, IJCNN, 2021.
> [E] Attention Based Spatial-Temporal Graph Convolutional Networks for Traffic Flow Forecasting, AAAI, 2019.
>
>
> **Summarize:** We kindly hope that we have addressed all of the remaining concerns from the reviewer effectively? We special thanks to the reviewer again for spending time reviewing our paper and helping us to improve our works.

---

### Author Response · Authors · 2025-12-02
**General responds summary**

**Dear Reviewers, Area Chairs and Senior Area Chairs,**

We value your time and thank you for your effort and hard work. We provide an official summary highlighting paper revision, main Reviewers' concerns, as well as our responses, which are available below.

---

> ### Author Response · Authors · 2025-12-02
> **Summary for Reviewer wMmd**
>
> **Comments:** The reviewer raised concerns regarding the readability of the methodology (dense notation), questioned the novelty of the framework (perceived as a combination of local-global schemes and DMCT), and expressed concern regarding scalability, specifically fearing $O(N^2)$ complexity for large networks.
>
> **Responses:**
>
> **1. Readability:** We significantly revised the paper, adding a notation table, correcting inconsistencies in definitions (specifically definitions 2 and 3), and refining the problem formulation to enhance clarity.
>
> **2. Novelty:** We clarified that FlowNet’s novelty lies in its **unique iterative interaction scheme based on flow physics (inflow/outflow)**, **Local and global interaction scheme among forecasting models**, **Ensemble outcome from diverse inputs**, **Data and model flexibility**, and **Interaction filtering**. To the best of our knowledge, FlowNet is the first independent iterative interaction approach for streamflow forecasting.
>
> **3. Scalability:** The $O(N^2)$ complexity is addressed by noting that river networks follow a **forest-style structure**, resulting in $O(N)$ edges, not $O(N^2)$. For example, LamaH (452 stations) has only 372 edges. We acknowledge a slower training time (e.g. FlowNet (DMCT) and FlowNet (RLinear) takes 3550.46 and 1505.65 seconds on the large CAMELS dataset, compared to 296.7s of AGCLSTM and 141.8s of ResGCN) but emphasize that it yields **significantly better prediction accuracy** and affordable training costs. **Inference time is acceptable**: FlowNet with DMCT takes 7.2s, and FlowNet with RLinear takes only 1.7s on large datasets, not the fastest but not the worst among all competitors (e.g. 3.2s of PatchTST, 4.1s of MICN or 3.6s of TQNet). We can improve the runtime of FlowNet straightforwardly via **Predicting sub-areas** and **Parallel processing**.
>
> **4. Generality on other tasks:** We also demonstrated the generality of FlowNet to traffic forecasting on road networks, a separate domain. FlowNet also acquires better accuracy than the SOTA traffic forecasting model ASTGNN in both runtimes and prediction accuracy (FlowNet 17.51 vs ASTGNN 26.21 on RMSE H=1 and FlowNet 2.43s vs ASTGNN 3.92s on runtime).
>
> **The reviewer kindly agreed with most of our answers and kindly raised their rating score.** The remaining concern on the generality and scalability on other forecasting tasks were carefully addressed in our last responses after the rebuttal closed notification.

---

> ### Author Response · Authors · 2025-12-02
> **Summary for Reviewer EzYW**
>
> **Comments:** The reviewer suggested comparing the method against Large Time Series Foundation Models (TS-LLMs) like TimesFM, questioned the computational complexity and some suggestions in the Related work section.
>
> **Responses:**
>
> **1. TS-LLMs:** We included a comparison with the large pre-trained foundation model for time series forecasting **TimesFM** in Appendix C.6. Results show FlowNet achieves **better performance** across most NSE (0.5784 vs 0.5577 on CAMELS H=1) and RMSE (250.7 vs 255.7on CAMELS H=1) metrics. Crucially, FlowNet is **significantly faster in inference** (7.2 seconds) compared to TimesFM (more than 18 hours) on the CAMELS dataset.
>
> **2. Clarity of Ablation Study:** We completely agree and have added **detailed ablation study tables** (Sections C.7, C.8, C.9) to provide clearer quantitative results for the analysis.
>
> **3. Baselines:** We added citations and rationale in the **Related Work** and **Appendix B.2** to explain that general SOTA time-series models were included to ensure a comprehensive comparison, not limited to hydrology-specific models.
>
> **4. Efficiency:** We acknowledge FlowNet's higher training cost (3550.46s with DMCT and 1505.6 with RLinear on CAMELS) due to its iterative scheme, compared to 296.7s of AGCLSTM and 141.8s of ResGCN. This complexity is justified by the **much better prediction accuracy** (mean NSE 0.5784 for FlowNet vs. 0.5414 for ResGAT), while still being affordable. Inference time is reasonable (7.2s for Flow with DMCT, 1.71s for FlowNet with RLinear on CAMELS, not the fastest but not the worst among all competitors, e.g. 3.2s of PatchTST, 4.1s of MICN or 3.6s of TQNet). Runtimes can be reduced via predicting sub-areas and parallel processing.
>
> **We did not hear back from the reviewer before the rebuttal closed. But we strongly believe that all of the concerns are fully addressed.**

---

> ### Author Response · Authors · 2025-12-02
> **Summary for Reviewer LeBJ**
>
> **Comments:** The reviewer felt the DMCT backbone lacked novelty (mainly engineering), suggested weighted graphs over binary connections, and noted that the ablation study showed limited quantitative gains from the interaction scheme.
>
> **Response:**
>
> **1. DMCT Novelty:** We agree that DMCT is an engineering effort and a side contribution of our paper, while the **FlowNet framework is the unique and main novelty**.
>
> **2. Graph:** We acknowledge this as a great suggestion and intend to study different kinds of weighted streamflow graphs, including distance-based weights, in future work.
>
> **3. Ablation Validity:** The seemingly small improvements were observed on the high-performing MRB dataset due to reported averaged results on 26 stations. New results on larger datasets CAMELS and LamaH prove the scheme's effectiveness: **85.1% of stations** improved on CAMELS, and **93.8% of stations** improved on LamaH after the global interaction phase, demonstrating a **significant and convincing** effects. We also provide statistical significance via Wilcoxon signed rank tests to strengthen the conclusion.
>
> **4. Global Loss Function:** We justify the global loss as ensuring the prediction ($\hat{y}^i$) stays close to both its **ground truth** ($y^i$) and the **ensembled prediction** ($\hat{y}^i_{Global}$) from three different viewpoints (per-station, inflow, outflow), which is effective, as demonstrated by the optimal $\alpha < 1$ values in the ablation study.
>
> **The Reviewer kindly agreed with the response. A final question on $\alpha=1$  was addressed after the rebuttal cutoff decision.**

---

> ### Author Response · Authors · 2025-12-02
> **Summary for Reviewer n4Tx (Part 1)**
>
> **Comments:** The reviewer raised a concern of the graph dependency (cannot work well on large-scale data), questioned the use of meteorological data and streamflow availability (real-time data concerns), and argued that training per-station models is inefficient compared to single generalized models.
>
> **Response:**
> **1. Scalability:** We clarify the argument that river networks have $O(N)$ edges, not $O(N^2)$. Experiments on the large **LamaH (452 stations) and CAMELS (671 stations)** datasets prove the framework scales. Runtimes are acceptable and affordable (as we also pointed out  in other responses above).
>
> **2. Missing Data:** We clarify that **meteorological data** is used for all three studied datasets. We also confirmed that we **do not connect channels like temperature** in the streamflow direction.
>
> **2. Modeling Approach:** We defend the per-station approach, arguing that single "blackbox" models often fail to capture specific physical relationships in data-scarce regions (like the Mekong River Basin) and that FlowNet explicitly utilizes flow physics to guide learning.
>
> **3. Baselines:** We add **hyper-parameter search spaces** for all baselines (Appendix B.1) and detail **descriptions of all 18 baselines** (Appendix B.2). We include the suggested SOTA baseline **EALSTM** and demonstrate FlowNet's superior performance.
>
> **4. Negligible improvement:** The small difference is due to averaging over all 26 stations. We performed **Wilcoxon signed rank tests** on the CAMELS dataset, which proves that **FlowNet is statistically better** than all other competitors (e.g., $p-value$ of $1.086e^{-25} \ll 0.05$ when compared to PatchTST).
>
> **5. Parameters:** We clarify that **per-station models can be trained in parallel** to reduce overall time due to their independence. We confirm that we use the same parameters for all stations (using the same barebone model for all) to reduce runtime,  despite that FlowNet still outperforms baselines, further proving the framework's effectiveness.
>
> **6. Eq. 1 and 2 Relation:** We apologize for a typo and fixe Eq. 1 to $\hat{y}^i_{Global}=mean(\hat{y}^i_{inflow},\hat{y}^i_{outflow})$, clarifying the calculation of the loss function.
>
> **7. General Readability:** We introduce **Table 1** (key notations), update **Definition 2 and 3** for time consistency, fix the typo concerning $\hat{y}^i$ as ground truth, and revise the problem formulation for clarity.
>
> **8. Larger Datasets:** We clarify that the datasets used in this paper are not intended to maximize the number of data points, but rather to evaluate our model and other methods on the same benchmark. In particular, **CAMELS and LamaH are two golden standard benchmark datasets within the hydrological machine learning community.**
>
> **9. Meteorological Channels:** We clarify that FlowNet actually **does not ignore meteorological conditions in the neighborhood.**  FlowNet considers meteorological data from neighbors into inputs of cross-model stations. The outputs of cross-model stations will then be used to adjust the outputs of per-station models. It helps the output of each station to be determined also from meteorological data of its neighbors from the output of the models.
>
> **10. Data Acquisition and Data Assimilation:** We clarify that we chose **using both streamflow and climate data as inputs due to data availability in CAMELS and LamaH**. However, **FlowNet can be adapted to use climate data only** (just removing the streamflow from the input). After we train the models, the model can be used to **predict other stations without historical streamflow data**, either directly or using techniques like transfer learning.
>
> **(to be continued...)**

---

> > ### Author Response · Authors · 2025-12-02
> > **Summary for Reviewer n4Tx (Part 2)**
> >
> > **11. Interpretability:** We clarify that most DL models are blackboxes and so are FlowNet and GNNs. *And interpretability is not in the scope of our paper*. We agree with the reviewer that designing a fully interpretable forecasting model is an interesting direction.
> >
> > **12. Data Unavailability:** We confirm that **FlowNet will be particularly suitable for data unavailability**. And FlowNet can employ as much data as possible which other methods cannot and acquires significantly better performances.
> >
> > **13. Distributed Method:** We clarify that **we did not do any distributed method as the reviewer claimed**.
> >
> > **14. Compared to EA-LSTM:** We clarify that we retrained EA-LSTM using Python codes provided by the authors to report the results on 671 stations with the same experimental settings to be consistent with our results on FlowNet and other SOTA models we studied. **FlowNet acquires better prediction accuracy than EA-LSTM (0.5784 vs 0.564 on CAMELS H=1)**.
> >
> > **15. Multi-scale Ablation:** Ablation studies confirm that **each component contributes** to DMCT's overall performance, including the Instance Normalization module. We add new ablation results on LamaH dataset, compared to the DMCT, **without multi-scale modules the performance decreased 5.28%, 6.57% and 2.48% in H=1,3 and 5, respectively.**
> >
> > **16. InstanceNorm:** InstanceNorm is a well-proven method to improve prediction accuracy in many model general forecasting systems. In our case, we did try some different ways and DMCT works well with InstanceNorm.
> >
> > **The discussions with the Reviewer were  ongoing before the rebuttal discussion stopped. However, we strongly believe that we addressed all the remaining concerns.**

---

> ### Author Response · Authors · 2025-12-02
>
> We have incorporated all the above additions into the revised manuscript. We believe these supplementary experiments and analyses effectively address the reviewers' concerns and solidify the paper's contribution.
>
> Thank you again for your time and dedication to improving our work.
>
> Sincerely,
>
> The Authors of FlowNet

---

### Meta-Review · Area_Chair_Fbm1 · 2026-01-09

**Summary:**

This paper presents FlowNet, a flexible and generalizable framework for multivariate spatio-temporal streamflow forecasting across multiple gauge stations.  A Disentangled Multiscale Cross-attention Transformer (DMCT) is further proposed as a backbone model to capture the multi-scale seasonality-trend information. Experiments on three hydrological datasets demonstrate the superior performance of proposed method over 18 SOTA baselines.

**Reviewer Concerns:**

All four reviewers provided insightful and constructive feedback. The major concern is the method’s high computational cost relative to its modest performance gains. Although the authors suggest that the runtime can be reduced via predicting sub-areas or using a lighter-weight RLinear variant, they did not provide any additional experimental results during the rebuttal to validate the effectiveness of the proposal. Unfortunately, this omission makes the paper fall below the acceptance threshold.

**Reviewer Scores:**

Reviewer wMmd: The method's scalability to large-scale datasets is limited and was not fully addressed. The novelty of this work stems primarily from the specific problem setting rather than the methodology.
Reviewer EzYW: The comparison with large time-series models is missing.
Reviewer LeBJ: The design of the global loss lacks explanation and was not fully addressed.
Reviewer n4Tx: Incorrect assumption about reality: streamflow cannot be obtained in real time, yet the framework relies heavily on it.

---

### Decision · Program_Chairs · 2026-01-26

Reject